# Vascular invasion-associated gene expression is detectable in pre-surgical biopsies of stage I lung adenocarcinoma

Dylan Steiner [1], Lila Sultan[2], Travis Sullivan[3], Hanqiao Liu[1], Xiaohui Xiao[1], Ashley LeClerc[4], Savannah Melvin[5], Yuriy O. Alekseyev [2], Gang Liu[1], Sarah A. Mazzilli [1], Jiarui Zhang[1], Kei Suzuki[5], Kimberly Rieger-Christ[3], Eric J. Burks [2], Jennifer Beane [1,6] & Marc E. Lenburg [1,2,6] ✉

Microscopic vascular invasion (VI) predicts recurrence and benefit from lobectomy in stage I lung adenocarcinoma (LUAD) but cannot be accurately predicted before surgery. Thus, biomarkers that identify this aggressive tumor subset are needed. Here, we show that VI in stage I LUAD is associated with reproducible gene expression programs detectable beyond the invasive focus. Using bulk RNA sequencing of 162 resected tumors and spatial transcriptomics in a subset, we identify and characterize a VI-associated gene signature. A predictor derived from this signature is associated with VI and recurrence in an independent validation cohort and is robust to intra-tumor heterogeneity in multiregional sampling data. In a cohort of pre-surgical biopsies, predictor scores correlate with matched resections and show promising discrimination of VI. These findings indicate that VI-associated transcriptional changes extend across the tumor and are detectable in limited biopsy material, supporting further validation for preoperative risk stratification in stage I LUAD.

Lung adenocarcinoma (LUAD) is the most common lung cancer subtype, and invasive LUAD represents 70–90% of surgically resected lung cancers[1]. Improved early detection through computed tomography (CT)-based screening programs is expected to increase the proportion of LUAD diagnosed at an early stage. Microscopic vascular invasion (VI), defined as tumor invasion within the lumen of veins or arteries, is a well-described route to metastatic dissemination and is consistently associated with higher rates of tumor recurrence among stage I LUAD[2–7]. VI is not included in the current World Health Organization (WHO) grading system for lung cancer, but may be a better predictor of recurrence than the most severe WHO-2021 grade, leading to our proposal for VI⁺ LUAD to be reclassified as distinct angioinvasive LUAD[8].

Patients with stage I VI⁺ LUAD may benefit from adjuvant therapy[9–12], but it is difficult to assess in resected tumor specimens. Elastic stains can be used to improve visualization of invaded blood vessels over hematoxylin and eosin (H&E), but comprehensive tumor histopathology review for small (<1 mm) VI foci is difficult and prone to false negatives. A biomarker of VI would improve the existing routine histopathologic review of high-risk cases at resection. Moreover, biopsy specimens do not provide enough tissue material for pathologists to identify VI⁺ LUAD prior to surgery, preventing the use of VI status to guide surgical or neoadjuvant treatment approaches. Retrospective analysis shows that VI⁺ patients receiving sublobar resection have increased rates of recurrence[13]. While recent evidence from large clinical trials, including JCOG0802/WJOG4607L and CALGB140503,

[1]Department of Medicine, Section of Computational Biomedicine, Boston University Chobanian and Avedisian School of Medicine, Boston, MA, USA. [2]Department of Pathology and Laboratory Medicine, Boston University Chobanian and Avedisian School of Medicine, Boston, MA, USA. [3]Department of Translational Research, Lahey Hospital and Medical Center, Burlington, MA, USA. [4]Boston University Microarray and Sequencing Resource Core Facility, Boston, MA, USA. [5]Thoracic Surgery, Inova Schar Cancer Institute, Fairfax, VA, USA. [6]These authors jointly supervised this work: Jennifer Beane, Marc E. Lenburg. ✉e-mail: mlenburg@bu.edu

suggests that lobectomy provides no clinical benefit over sublobar resection for patients with stage IA disease[14,15], post-hoc analyses of pathologic factors specifically associated with recurrence in patients who underwent sublobar resection have yet to be published[16,17]. Therefore, there is an emerging need to identify if stage I patients will benefit from precision surgery. In parallel, biomarker approaches could also improve patient selection for neoadjuvant or adjuvant therapy in early-stage LUAD[18,19].

Molecular profiling technologies are common in clinical pathology, but prior RNA-sequencing (RNA-seq) studies of LUAD have primarily focused on identifying signatures of outcome, which lack reproducibility[20,21]. Others have identified signatures associated with high-grade growth patterns or subtypes[22,23], such as solid LUAD, but they have not been clinically deployed. To our knowledge, VI-associated gene expression has not been studied in lung cancer. Pathologists infrequently document VI as a separate entity from lymphatic invasion (LI), preferring to designate the presence of either type as lymphovascular invasion (LVI). This obfuscates the independent prognostic value of VI[24] and complicates downstream molecular approaches seeking to disentangle these modes of tumor spread, the differences of which are still incompletely understood[25]. Spatial transcriptomics (stRNA-seq) allows for resolving transcriptomic changes associated with tumor invasion within the spatial context of the tumor and tumor microenvironment[26]. This represents an opportunity to elucidate VI+ LUAD biology with the aim to improve detection of these tumors.

In this study, we profile 163 tumors with detailed histopathologic annotations by RNA-seq, including 15 by stRNA-seq, from a diverse multi-institutional cohort of stage I LUAD patients to comprehensively define the transcriptional landscape of angioinvasive LUAD. We derive a molecular signature associated with VI+ LUAD using bulk RNA-seq, describe its association with histopathology in situ using stRNA-seq, develop and validate a machine-learning predictor of VI, evaluate its stability across intra-tumor heterogeneity (ITH), and demonstrate its proof-of-concept feasibility in a pilot cohort of pre-surgical tumor biopsies.

## Results

### VI is the invasion type most associated with stage I LUAD recurrence

We assembled a retrospective discovery cohort of stage I LUAD tumors ($n = 192$) from patients across two institutions, Boston Medical Center (BMC), a large urban safety-net hospital, and Lahey Hospital and Medical Center (LHMC), a suburban hospital, a subset of a previously published cohort with detailed clinicopathology annotations (Table S1)[8]. Tumors were graded using our novel grading system, whose reproducibility has been validated by us and others[8,13,27,28]. This novel grading system classifies tumors into low malignant potential (LMP), no special type (NST), or VI, and is more associated with outcome than the WHO grading system (Supplementary Fig. 1a, b). VI was the strongest predictor (hazard ratio (HR) = 7.62, 95% CI 3.70–15.8, Cox regression $p = 5.17 \times 10^{-8}$) of 7-year recurrence-free survival (RFS) among the four observed modes of tumor spread (VI, LI, visceral pleural invasion (VPI), and spread through air spaces (STAS)), in agreement with previous reports of its prognostic value in stage I LUAD[8,29–31] (Supplementary Fig. 1c–g). VI was present in 30% of tumors, and 84% of VI+ tumors contained multiple types of invasion (Supplementary Fig. 1h). Regardless, VI remained a significant predictor of recurrence when including clinical covariates and other invasion types (HR = 11.04, 95% CI 3.80–32.1, multivariate Cox regression $p = 1.01 \times 10^{-5}$; Supplementary Fig. 1i). Of the different invasion types, only LI was also an independent predictor of recurrence (HR = 2.73, 95% CI 1.15–6.44, multivariate Cox regression $p = 0.02$). When scoring tumors based on the proportion of LUAD growth patterns, VI was associated with solid (Bonferroni adjusted $P$ value (p.adj) = 0.004,

Wilcoxon test), but not micropapillary (p.adj = 0.08) or cribriform proportion (p.adj = 0.35) (Supplementary Fig. 1j). In contrast, STAS was most strongly positively associated with increased micropapillary proportion (p.adj = $5.3 \times 10^{-7}$), as previously reported[32]. VI+ tumors were more strongly associated with distant recurrence (sub-distribution HR = 19.1, 95% CI 4.27–85.0, multivariable Fine-Gray regression $p = 1.1 \times 10^{-4}$) than loco-regional recurrence only (sub-distribution HR = 3.86, 95% CI 1.22–12.2, multivariable Fine-Gray regression $p = 0.02$), consistent with tumor spread through the vascular system (Supplementary Fig. 1k–m).

### Gene expression changes in stage I LUAD with VI

To identify gene expression changes in tumors with VI, we profiled a subset of surgically resected stage I LUAD tumors from the discovery cohort ($n = 108$, $n = 103$, post-QC) by bulk RNA-sequencing (RNA-seq) (Fig. 1a, Table S2). Among tumors that passed quality control, 78 were VI- and 25 were VI+. We identified 474 genes differentially expressed between VI+ tumors and tumors of LMP using the three-level factor of tumor grade (LMP, NST, VI) as the independent variable at a significance threshold of FDR < 0.01, which we clustered into four gene expression clusters (Fig. 1b). We selected the optimal number of clusters as four via consensus clustering (Supplementary Fig. 2a). Genes in three of the clusters had increased expression in LUAD with VI (clusters 1, 2, and 3) and genes in the other cluster had decreased expression (cluster 4). Pathway enrichment analysis using EnrichR (FDR < 0.05) revealed that genes involved in different biological processes were enriched in each cluster (Fig. 1c). Cluster 1 ($n = 115$ genes) was enriched for genes involved in the cell cycle. In contrast, cluster 2 ($n = 37$ genes) was enriched for genes involved in tissue remodeling and vasculogenesis, including the epithelial to mesenchymal transition (EMT), extracellular matrix organization, and angiogenesis. The genes in cluster 3 ($n = 182$ genes) were enriched for some pathways that overlapped with cluster 1, like mTORC1 signaling and E2F targets, but was distinguished by enrichment for pathways related to response to reactive oxygen species and hypoxia. Finally, the genes with decreased expression in VI tumors (cluster 4; $n = 140$ genes) were enriched for pathways indicating reduced malignant progression (p53 pathway, regulation of cell growth), increased cell-cell adhesion via TGF-β signaling (regulation of pathway-restricted SMAD protein phosphorylation), and increased immune surveillance (IL-2/STAT5 signaling) in VI- tumors. In the discovery cohort, the mean z-score of the genes in each of the four clusters was a significant predictor of VI (Supplementary Fig. 2b) and was also able to distinguish between NST and VI+ tumors (Supplementary Fig. 2c), supporting that the differentially expressed genes reflect VI-specific biology. We observed that two VI+ tumors clustered with the LMP classified tumors. These both represented rare edge cases, with one being the only lepidic predominant VI+ sample and the other being the only VI+ sample with >50% micropapillary growth pattern. Larger cohorts containing more of these types of samples will be required to determine if the current signature might be less accurate in detecting VI in these conditions.

Given that VI and LI are routinely reported together as LVI[8,24], we compared the VI signature with gene expression differences associated with LI in stage I LUAD. We added LI as a covariate to our model to determine whether the gene expression differences seen with VI are due to the covariance between LI and VI. We identified 133 genes associated with VI (VI+ vs. LMP, FDR < 0.01), which included 130 genes from the original 474-gene VI signature (Supplementary Fig. 3a). At the same significance threshold (FDR < 0.01), we recovered only 6 genes associated with LI (LI+ vs. LI-) using the same model, with 2/6 belonging to the VI signature (Supplementary Fig. 3b), suggesting the biological signal for LI is much weaker. To further interrogate the relationship between LI and VI-associated gene expression, we explored whether the various VI gene clusters from the 474-gene signature are enriched among the genes most differentially expressed in LI+ tumors using

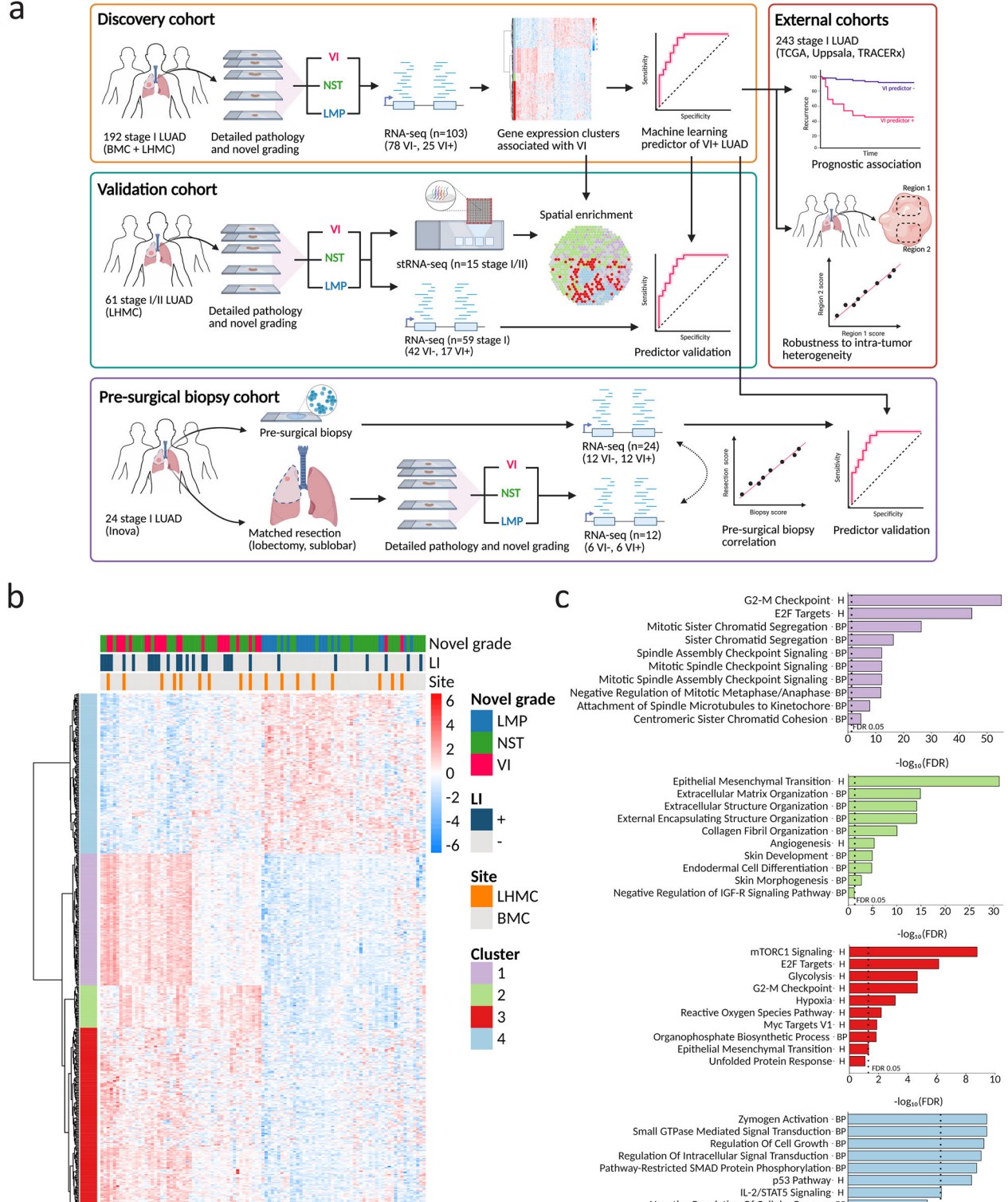

**Fig. 1 | Study overview and distinct gene expression changes associated with VI in stage I LUAD. a** Overview of cohorts, sequencing technologies, and analyses utilized in the study. Created in BioRender. Steiner, D. (https://BioRender.com/l1bofp9) is licensed under CC BY 4.0. VI vascular invasion, NST no special type, LMP low malignant potential, BMC Boston Medical Center, LHMC Lahey Hospital and Medical Center. All tumors were selected to be stage I at the time of collection, except for one tumor profiled by stRNA-seq that was upstaged to stage IIA under the 8th TNM edition. **b** Co-expression heatmap of 474 genes differentially expressed between VI and LMP (FDR < 0.01) in a discovery cohort of stage I LUAD tumors ($n = 103$) grouped into $k = 4$ clusters. Heatmap units are log counts per million (CPM) scaled by transcript. LI lymphatic invasion. **c** Top 10 enrichment terms of each gene co-expression cluster. Pathways were ranked within each cluster by FDR values obtained analyzing each cluster's gene list with EnrichR and the MSigDB Hallmark 2020 (H) and GO Biological Process 2021 (BP) gene set collections. Source data are provided as a Source data file.

gene-set enrichment analysis (GSEA). Interestingly, we found that the genes in VI cluster 4 are significantly enriched among the genes decreased in LI[+] tumors (Supplementary Fig. 3f). But unlike the genes in VI cluster 1 (cell-cycle) and VI cluster 3 (hypoxia), the genes in VI cluster 2 (EMT/angiogenesis), were not significantly enriched among the genes most increased in LI[+] tumors (Supplementary Fig. 3c–e). Finally, genes previously found to be increased in VI[+] endometrial cancer[33] or LVI[+] breast cancer[34] were each significantly enriched among genes most increased in VI[+] stage I LUAD tumors using GSEA (Supplementary Fig. 3g, h). This hints at common molecular changes associated with VI across epithelial tumors that should be explored in future work.

## VI gene clusters are associated with specific LUAD histopathology features in stRNA-seq data

Although the genes differentially expressed in tumors with VI were enriched for biological processes implicated in tumor intravasation, the RNA for sequencing was isolated from histologic sections selected to be representative of the predominant histologic pattern and were not selected to include invaded blood vessel(s). Given that VI foci represent such a small percentage of tumor volume (diameter ≤1 mm), we hypothesized that the VI signature we identified reflected molecular changes extending beyond the site of intravasation. To assess the spatial architecture of the expression of the VI signature and its association with LUAD histopathology features, we profiled 15 (post-QC) resected stage I and stage II LUAD samples ($n = 8$ VI[-], $n = 7$ VI[+]) from 13 patient tumors by high-resolution stRNA-seq using the 10× Genomics Visium platform, including 3 sections containing invaded vessels directly in the 6.5 mm[2] Visium capture area (Table S3). Of the VI[-] tumors, 7/8 were NST. All six non-mucinous LUAD growth patterns (solid, cribriform, micropapillary, acinar, papillary, lepidic) were represented across the stRNA-seq capture areas, along with three invasion types (VI, VPI, and STAS) (Fig. 2a). Adjacent normal appearing lung was present in 12/16 capture areas. The histopathology associated with VI[+] tumors in the stRNA-seq data was generally representative of trends observed in our larger clinical cohort, although the micropapillary pattern tended to be underrepresented in VI[+] tumors (p.adj = $8.62 \times 10^{-14}$) and the papillary pattern was overrepresented (p.adj = $1.01 \times 10^{-11}$) (Supplementary Fig. 4a). A solid growth pattern and the presence of desmoplastic stroma were both associated with angioinvasion (p.adj = $6.38 \times 10^{-31}$; p.adj = $2.26 \times 10^{-18}$). While VI co-occurred with other invasion types frequently in our clinical cohorts, we did not find more than one invasion type per capture area in the stRNA-seq dataset. In summary, our stRNA-seq dataset captures many of the pathology features that are known to occur in stage I LUAD and co-occur with VI.

After stRNA-seq quality control and filtering, we obtained expression estimates from 43,421 spots (50 μm diameter) with a median depth of 8548 unique molecular identifiers (UMIs)/spot (Supplementary Fig. 4b) that detected a median 3996 genes/spot (Supplementary Fig. 4c). The low depth of sample 6 was likely due to tissue detachment that occurred during the stRNA-seq workflow, and this sample was removed prior to downstream analysis. Strong concordance was observed between pseudo-bulked stRNA-seq gene expression counts and bulk RNA-seq counts of matched tumor tissues across all genes (spearman $R = 0.85$) (Supplementary Fig. 4d) and across VI signature genes (spearman $R = 0.78$) (Supplementary Fig. 4e). We next scored individual spots to create an enrichment score representing the activity of the genes from each of the four clusters in our bulk VI signature. Across all capture areas, spatially weighted correlation analysis of enrichment scores revealed that individual VI clusters were spatially distinct from one another (Supplementary Fig. 4f). These data suggest the distinct patterns of gene co-expression represented by the four clusters are also spatially distinct, indicating that the biological processes that these gene expression patterns

represent are spatially variable and are unlikely to represent redundant biology.

To investigate if the VI gene clusters are more strongly expressed in regions with different histopathologic feature annotations in the stRNA-seq data, we modeled the enrichment score of each cluster as a function of histopathologic annotation using a linear mixed model with the section as a random effect. Unexpectedly, despite the VI gene clusters being derived in bulk RNA-seq, we observed a strong increase in expression directly in VI foci of both cluster 1 (p.adj = $9.02 \times 10^{-28}$) and cluster 3 (p.adj = $8.92 \times 10^{-73}$). Cluster 1 and cluster 3 were also increased in regions annotated as high-grade patterns (solid (p.adj = $8.87 \times 10^{-15}$; $7.90 \times 10^{-75}$) micropapillary (p.adj = $3.94 \times 10^{-07}$; $1.40 \times 10^{-23}$) and cribriform (p.adj = $4.23 \times 10^{-08}$; $2.39 \times 10^{-58}$)) (Fig. 2b). Cluster 2 was most noticeably expressed in desmoplastic stroma (p.adj = $1.22^{-122}$) with a greater than 2.95-fold higher mean enrichment score than in stroma (p.adj = $1.87 \times 10^{-25}$). Conversely, expression of cluster 4 was strongly decreased in VI foci (p.adj = $5.79 \times 10^{-79}$), followed by desmoplastic stroma (p.adj = $6.53 \times 10^{-50}$) and showed strong positive enrichment in adjacent normal-appearing lung (p.adj = $2.57 \times 10^{-04}$), stroma (p.adj = $8.90 \times 10^{-12}$), and papillary ($1.68 \times 10^{-13}$). When enrichment scores were visualized directly on tissue sections, the independent spatial patterning of each cluster was apparent (Fig. 2c). Given that clusters 1 and 3 were significantly enriched directly in invasive foci in addition to high-grade patterns, we were interested to see whether the expression of these genes was also higher in VI[+] tumors independently of aggressive pattern. When we analyzed only spots annotated with high-grade LUAD patterns, we found that cluster 1 ($p = 0.047$), 2 ($p = 0.004$) and 3 ($p = 0.003$) were all higher and cluster 4 ($p = 6.9 \times 10^{-5}$) was lower in spots annotated with high-grade patterns belonging to VI[+] tumors, suggesting that our VI signature is capturing an angioinvasive phenotype that is independent of aggressive pattern and the invasive focus (Fig. 2d). In total, these findings suggest that gene expression clusters of angioinvasive LUAD identified by bulk RNA-seq are differentially expressed between distinct histopathologic features in situ, show increased expression in the same tumor patterns if it is a VI[+] tumor, and are not solely expressed in invaded blood vessels.

## The VI signature is composed of both tumor-specific and tumor-microenvironment changes reflective of angioinvasion

Our supervised analysis above relied upon detailed pathologic annotation of the tissue containing Visium spots, but spots were only labeled if there was a clear consensus on morphology, especially for the histologic pattern. In some cases, tumor areas may not have been annotated if they did not clearly contain canonical features of one of the six histologic patterns[35]. Additionally, for poorly differentiated patterns such as solid, the annotation may include admixed stromal or immune cells. With the aim to further delineate the spatial biology of the VI clusters, we estimated the cell type composition of each stRNA-seq spot with CytoSPACE by using a published single-cell RNAseq atlas of 295,813 annotated cells from 124 stage I LUAD samples (primary tumor and normal adjacent lung) and segmenting nuclei per spot with StarDist[36–39]. LUAD pathology annotations showed distinct cell type composition (e.g., B cells with lymphoid aggregates) (Supplementary Fig. 5a). When we modeled predicted cell type proportion per spot as a function of histopathologic annotation using a linear mixed model with the section as a random effect, we observed a significant enrichment of predicted B cells, Plasma cells, and other immune cell types with lymphoid aggregates (Fig. 3a). Normal bronchus had a significantly higher proportion of predicted ciliated cells, as expected. Spots annotated as VI foci had significantly higher proportions of predicted tumor cells, while desmoplastic stromal regions had significantly higher proportions of peribronchial fibroblasts. We found a trend toward a higher proportion of peribronchial fibroblasts belonging to VI[+] stRNA-seq samples (Fig. 3b), consistent with the tissue

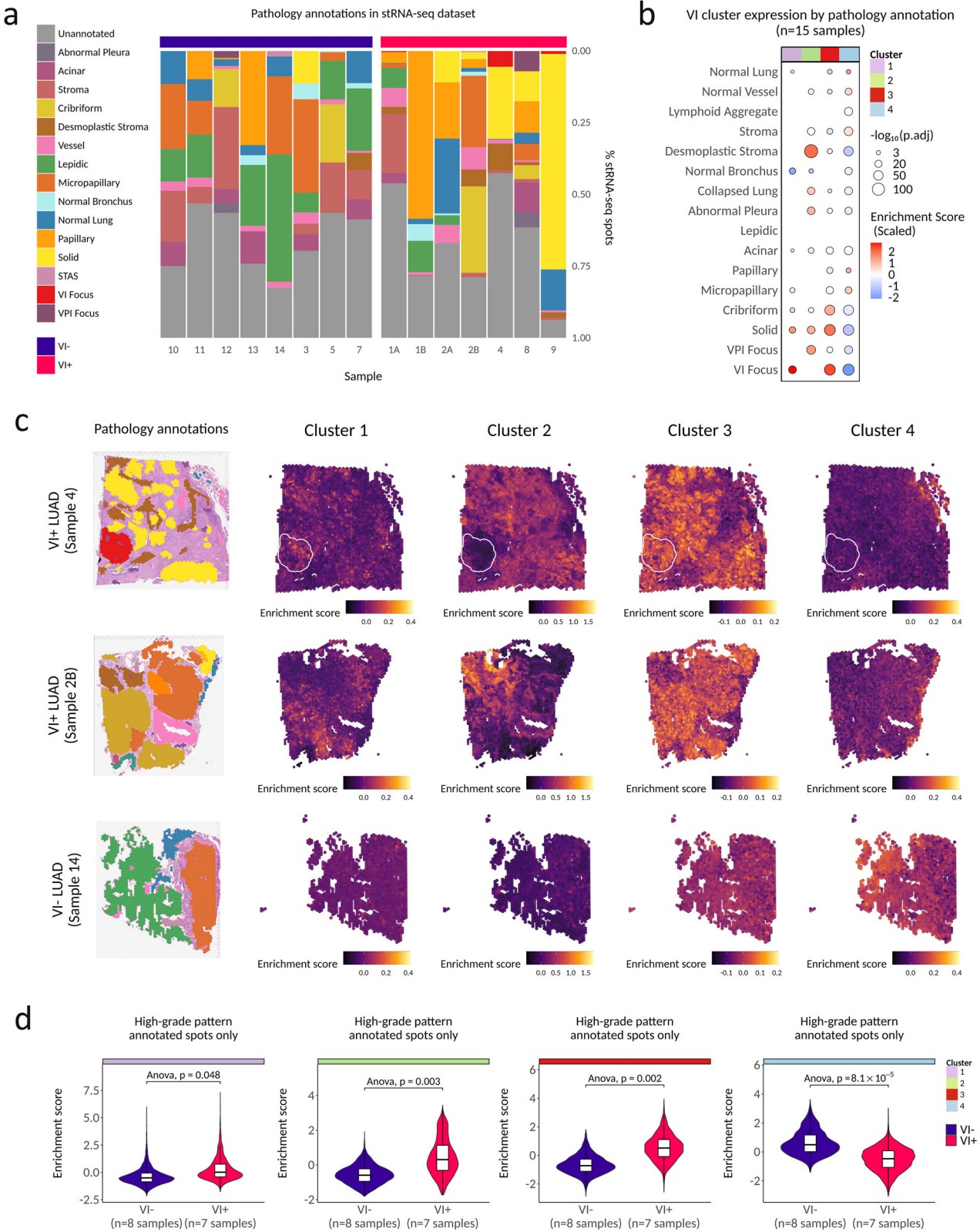

remodeling biology of VI cluster 2 described above. Peribronchial, but not adventitial or alveolar fibroblasts, showed a significantly higher enrichment for a published NSCLC myofibroblast-cancer-associated fibroblast (CAF) signature from Hanley et al.[40] (Fig. 3c). Hanley et al reported key markers of LUAD myofibroblasts as *MMP11*, *POSTN*, *CTHRC1*, *COL1A1*, and *COL3A1*, which are all VI cluster 2 genes. Consistently, VI cluster 2 was significantly more highly enriched in this

myofibroblast cluster than with alveolar and adventitial clusters in LUAD scRNAseq samples from Hanley et al ($n = 39$) (Fig. 3d). Deconvoluted VI⁺ stRNA-seq samples also had a trend toward a higher proportion of plasma cells (Fig. 3b). To validate these findings, we deconvoluted the bulk RNA-seq discovery cohort data using the same atlas from Salcher et al. Although the predicted plasma cell proportions were associated with pathologist-annotated plasma cell grade on

**Fig. 2 | Spatial transcriptomics of early-stage LUAD reveals an association of VI gene clusters with specific LUAD histopathology features. a** Pathology annotations present across spatial transcriptomics (stRNA-seq) capture areas that passed QC ($n = 15$). **b** Association between expression of VI-associated gene clusters (measured as per-spot VI cluster enrichment scores) and pathology annotations across all samples ($n = 15$). Statistical significance was assessed using two-sided Wald tests from linear mixed effect models predicting per-spot VI cluster enrichment scores as a function of spot pathology annotation, with sample included as a random effect. $P$ values were adjusted for multiple comparisons across all cluster-pathology association tests using Holm-Bonferroni correction, and only terms with adjusted $P < 0.01$ are shown. For visualization, points represent the mean per-spot enrichment score within each pathology

region; point size reflects $-\log_{10}$(adjusted $P$ value), and point color indicates mean enrichment score scaled within each gene cluster. **c** Pathology annotations (legend in **a**) in stRNA-seq samples and expression maps of VI gene expression clusters in representative samples from VI$^+$ (white outlines highlight VI foci) and VI$^-$ LUAD. **d** Spot-wise scoring of VI$^-$ and VI$^+$ stRNA-seq samples for expression of VI gene expression clusters in spots annotated with high-grade LUAD patterns (solid, micropapillary, and cribriform). Boxplots display the median, interquartile range (25th–75th percentiles), and whiskers extending to 1.5× the interquartile range. Violin plots depict the distribution of spot-level scores. $P$ values are derived from a type II ANOVA of a linear mixed model predicting gene expression with sample as a random effect and high-grade pattern as a fixed effect. Source data are provided as a Source data file.

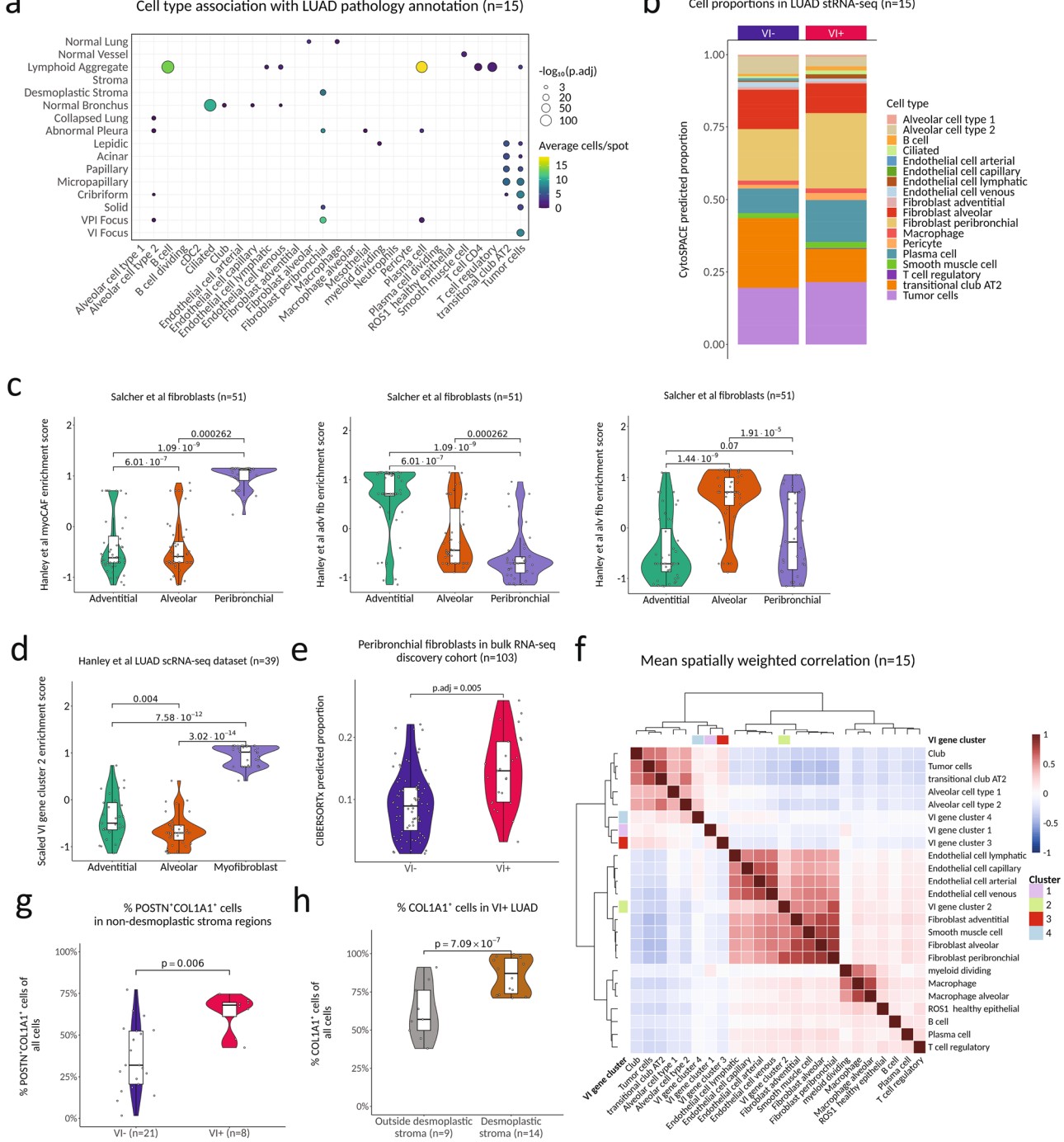

**Fig. 3 | The VI signature is composed of both tumor-specific and tumor-microenvironment changes reflective of angioinvasion. a** Association between predicted stRNA-seq per-spot cell type proportions and pathology annotations (*n* = 15 samples). STAS, spread through air spaces; VPI, visceral pleural invasion. Associations were tested using linear mixed-effects models predicting per-spot cell type proportion as a function of pathology annotation, with sample included as a random effect. Statistical significance was evaluated using two-sided Wald tests and adjusted for multiple comparisons across all cell type-pathology association tests using Holm-Bonferroni correction; only associations with adjusted *P* < 0.01 are shown. For visualization, points represent the mean predicted cell type proportion per spot within each pathology region: point size reflects −log₁₀(adjusted *P* value), and color indicates the average number of predicted cells per spot. Only results with p.adj < 0.01 are shown. **b** Deconvolution-predicted cell type proportions by VI status in the stRNA-seq data. Only cell types with predicted proportions >1% are shown. **c** Sample-level enrichment scores (averaged over single cells) of NSCLC myofibroblast-cancer-associated, alveolar, and adventitial fibroblast gene signatures from Hanley et al. 2023 in fibroblast subpopulations from the Salcher stage I LUAD scRNA-seq atlas (*n* = 51 samples). Boxplots display the median, interquartile range (25th–75th percentiles), and whiskers extending to 1.5× the interquartile range. Violin plots depict the distribution of sample-level scores, and points represent individual samples. *P* values were calculated using two-sided Wilcoxon rank-sum tests; no adjustment for multiple comparisons was applied. **d** Sample-level (*n* = 39) enrichment scores (averaged over single cells) of VI gene cluster 2 in

Hanley et al. fibroblast subpopulations, computed as in (**c**). Boxplots and violins are defined as in (**c**). *P* values were calculated as in (**c**). **e** Peribronchial fibroblast (myofibroblast) proportions by VI status in the bulk RNA-seq discovery cohort (*n* = 103 tumors). Boxplots and violins are defined as in (**c**). *P* values were calculated using two-sided Wilcoxon rank-sum test and adjusted for comparisons across all cell types using the Holm-Bonferroni correction. **f** Mean spatially weighted correlation of VI gene cluster enrichment scores and stage I LUAD cell type signatures from the Salcher et al. lung cancer atlas, averaged across all stRNA-seq samples. **g** Fraction of POSTN⁺COL1A1⁺ myofibroblasts measured by IHC out of all cells in pathologist annotated non-desmoplastic stroma regions stratified by VI status of the tumor (*n* = 29 regions, 21 VI⁻, 8 VI⁺). Boxplots are defined as in (**c**). Violin plots depict the distribution of cell fractions, and points represent individual samples. Statistical significance was assessed using a linear mixed-effects model with cell fraction as the outcome, VI status as a fixed effect, and sample as a random effect (*n* = 11 VI⁻, 9 VI⁺ tumors). Two-sided Wald t tests were used, and no adjustment for multiple comparisons was applied. **h** Fraction of COL1A1⁺ cells measured by IHC out of all cells in desmoplastic stroma regions (*n* = 14 regions) vs. outside desmoplastic stroma regions (*n* = 9 regions) in VI⁺ LUAD (*n* = 9 tumors). Boxplots are defined as in (**c**). Violin plots are defined as in (**g**). Statistical significance was assessed using a linear mixed-effects model with cell fraction as the outcome, region as a fixed effect, and sample as a random effect (*n* = 9 VI⁺ tumors). Statistical testing was performed as in (**g**). Source data are provided as a Source data file.

case-matched H&E images (*p* = 1.37 × 10⁻⁸) (Supplementary Fig. 5b), we did not observe an association with VI⁺ tumors (p.adj = 1), suggesting that plasma cell infiltration may be overrepresented in the stRNA-seq data. In contrast, in the same bulk RNA-seq data we found a significantly higher predicted proportion of peribronchial fibroblasts (myofibroblasts) (p.adj = 0.005) in VI⁺ tumors (Fig. 3e). To examine the associations between the expression of the VI gene clusters and predicted cell types in the stRNA-seq data, we first generated cell type specific gene signatures from the Salcher atlas (Supplementary Fig. 5c). We then used spatially weighted regression to identify correlation between cell type signatures and our VI gene clusters across the stRNA-seq data after removing low abundance cell types. Unsupervised clustering of the average spatially weighted correlation across all tumors revealed co-localization of cluster expression with distinct cell types (Fig. 3f). As expected from our previous observation of enrichment in desmoplastic stroma, VI cluster 2 was most spatially correlated with peribronchial fibroblasts, as well as other stromal cell types, including endothelial and smooth muscle cells. In contrast, VI clusters 1, 3, and 4 were all tightly spatially correlated with epithelial cells, suggesting the contribution of these gene clusters to the VI signature is dependent more on cell state than cell type. These associations were not as apparent when examining enrichment scores in the scRNA-seq atlas (Supplementary Fig. 5d), highlighting the added value of spatial deconvolution. Collectively, these results show that the VI gene clusters capture biological changes associated with angioinvasion from both tumor cells and the tumor microenvironment.

To further validate our stRNA-seq deconvolution findings about the relationship between VI cluster 2, desmoplastic stroma, and peribronchial fibroblasts, we performed *COL1A1* RNAscope ISH and COL1A1 IHC labeling on serial sections close to stRNA-seq sample 4, a VI⁺ tumor with prominent desmoplastic stroma. We selected *COL1A1* as it is a representative gene from VI cluster 2 that also showed significantly higher expression in myofibroblasts compared with adventitial or alveolar fibroblasts in Hanley et al (Supplementary Fig. 6a) After serial image registration to the H&E from the section used for stRNA-seq, we observed a significant correlation across spots between the fraction of ISH RNAscope *COL1A1*⁺ expressing cells per spot and the average VI cluster 2 gene expression per spot (*r* = 0.53, *p* < 2.2 × 10⁻¹⁶; Supplementary Fig. 6b). We also observed a similar significant correlation between the average VI cluster 2 gene expression per spot and the fraction of COL1A1⁺ cells per spot labeled by IHC (*r* = 0.47, *p* < 2.2 × 10⁻¹⁶; Supplementary Fig. 6c). We also performed IHC for THY-

1, a VI cluster 2 gene that was not reported by Hanley et al as a key myofibroblast marker. The fraction of THY-1+ expressing cells per spot by IHC was significantly correlated with the fraction of COL1A1+ expressing cells measured by IHC, which further supports the co-expression of VI gene cluster 2 at the protein level (*R* = 0.52, *p* < 2.2 × 10⁻¹⁶; Supplementary Fig. 6d). When we examined the correlation between COL1A1⁺ cell fractions measured by either IHC or ISH with the fraction of myofibroblasts predicted by the CytoSPACE stRNA-seq deconvolution, we found that both were significantly more correlated with the fraction of peribronchial/myofibroblast CAFs than either adventitial or alveolar fibroblasts (Supplementary Fig. 6e). This further demonstrates that cells expressing COL1A1, a key marker of VI cluster 2, show a myofibroblast-like phenotype. Finally, we also observed a significant association between the fraction of COL1A1⁺ IHC fraction and spots annotated as desmoplastic stroma (Supplementary Fig. 6f), further suggesting that VI cluster 2 is associated with tissue remodeling.

To further generalize these findings beyond the stRNA-seq sample, and to directly demonstrate the expression of COL1A1 in myofibroblasts, we analyzed an independent cohort (Table S4) of 20 resected stage I LUAD samples (11 VI⁻, 9 VI⁺) using duplex IHC for COL1A1 and POSTN. POSTN is a canonical myofibroblast marker[40,41] and a VI cluster 2 gene. In pathologist-annotated stroma regions (i.e., non-desmoplastic stroma), nearly all POSTN⁺ cells were also COL1A1⁺ (Supplementary Fig. 6g, h). The fraction of COL1A1⁺POSTN⁺ cells in these regions was significantly higher in VI⁺ than VI⁻ tumors (median 68% vs 32%, *p* = 0.006), consistent with VI-associated myofibroblast expansion (Fig. 3g). In analyzing desmoplastic stroma, we restricted our analysis to desmoplastic stroma from VI⁺ tumors, given the paucity of desmoplastic stroma in VI⁻ tumors. Within desmoplastic stroma, POSTN ⁺ cells are also nearly universally COL1A1⁺ (~100%, Supplementary Fig. 6i), confirming persistent co-expression within myofibroblasts. The fraction of all cells in desmoplastic stroma regions that are COL1A1⁺ was significantly higher than outside the desmoplastic stroma regions (median 87% vs 57%, *p* = 7.09 × 10⁻⁷) (Fig. 3h), without a corresponding rise in POSTN ⁺COL1A1⁺ cells (median 26.3% vs. 21.0%, *p* = 0.263) (Supplementary Fig. 6j). This is a result of an increase in COL1A1⁺POSTN⁻ stromal cells (median 61% vs 39%, *p* = 0.02) (Supplementary Fig. 6k). These findings support and extend the stRNA-seq results, demonstrating that the elevated expression of VI cluster 2 in VI⁺ tumors reflects both the VI-specific expansion of POSTN⁺COL1A1⁺ myofibroblasts in non-desmoplastic stroma, and the VI-associated

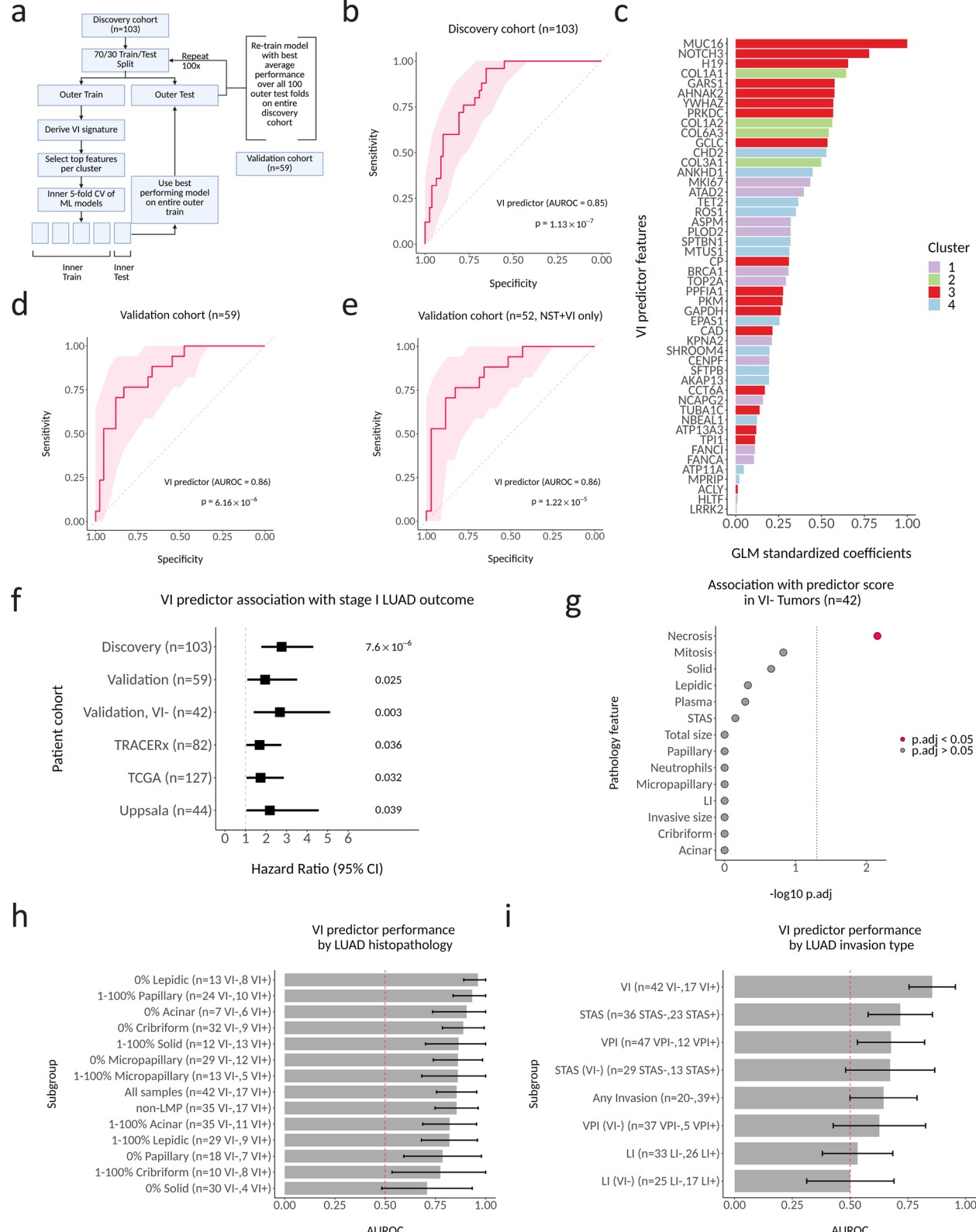

expansion of desmoplastic stroma in which the majority of COL1A1⁺ cells are POSTN⁻.

## A VI predictor derived from the signature validates in an independent stage I LUAD cohort

Our finding of biologically and spatially distinct gene expression clusters associated with VI⁺ LUAD suggests that all four clusters

provide orthogonal information and might be combined to form a predictor of angioinvasive stage I LUAD. To test this hypothesis, we proceeded with a standard machine learning cross validation pipeline using only the samples from the discovery cohort ($n = 103$; Fig. 4a). Training was performed in the discovery cohort ($n = 103$) where the differentially expressed genes were first identified, using a cross-validation approach (Fig. 4a). Predictors utilizing a binomial logit

**Fig. 4 | A VI predictor derived from the signature validates in an independent stage I LUAD cohort. a** Cross-validation approach for feature and model selection. Created in BioRender. Steiner, D. (https://BioRender.com/w7tillg) is licensed under CC BY 4.0. **b** Predictor performance for predicting VI in the discovery cohort ($n = 103$ tumors). Receiver operating characteristic (ROC) curve discriminating VI$^+$ and VI$^-$ tumors. AUROC, area under the ROC curve. Shaded regions indicate pointwise 95% CIs of sensitivity at fixed specificity, computed by stratified bootstrap resampling (2000 replicates). Statistical significance was assessed using a two-sided Wilcoxon rank-sum test comparing VI predictor scores between VI$^+$ and VI$^-$ tumors. **c** Feature importance (as indicated by the feature's standardized coefficient in the final generalized linear model, GLM) for the genes selected for the final VI predictor, colored by gene expression cluster. **d** Predictor performance in the validation cohort ($n = 59$ tumors). ROC curve and statistical analysis as in (b). **e** Predictor performance in the validation cohort in non-LMP tumors ($n = 52$ tumors). ROC curve and statistical analysis as in (b). **f** Predictor association with outcome across independent stage I LUAD cohorts: discovery (RFS, $n = 18$ events), validation (RFS, $n = 6$ events), VI$^-$ validation (RFS, $n = 5$ events), TRACERx (RFS, $n = 17$ events), TCGA (OS, $n = 14$ events), Uppsala (OS, $n = 11$ events). RFS recurrence-free survival, OS overall survival. Hazard ratios are shown per 1 SD increase in the predictor and were estimated using univariate Cox proportional hazards regression in each cohort. Error bars denote 95% CIs. *P* values are from two-sided Wald tests. No adjustment for multiple comparisons was performed. **g** Association of pathology features with VI predictor scores in VI$^-$ tumors within the validation cohort ($n = 42$). LI lymphatic invasion. Associations were assessed using univariate linear regression, with the VI predictor score as the dependent variable. Two-sided t tests were used for regression coefficients and *P* values Bonferroni-adjusted for multiple comparisons across pathology features. Values are displayed as -$\log_{10}$(adjusted *P*). The dotted vertical line indicates the Bonferroni-adjusted significance threshold (adjusted *P* = 0.05). **h** Predictor performance in the validation cohort stratified by histologic pattern ($n$ varies). Bars show the empirically estimated area under the ROC curve (AUROC) for each subgroup. Error bars denote the 95% CI computed using stratified bootstrap resampling ($n = 2000$). **i** Predictor performance when classifying other LUAD invasion types in the validation cohort. Bars show the empirically estimated AUROC. Error bars denote the 95% CI computed using stratified bootstrap resampling ($n = 2000$). Source data are provided as a Source data file.

generalized linear model (GLM) with ridge regression performed the best on the internal cross-validation test sets within our discovery cohort and we observed no dependence of gene set size on model performance (Supplementary Fig. 7a, b). When we trained a GLM with ridge regression on the full discovery cohort, a 48-gene predictor achieved an area under the receiver operating characteristic curve (AUROC) of 0.85 (95% CI 0.78–0.93, $p = 1.1 \times 10^{-7}$) to separate VI$^+$ and VI$^-$ LUAD (Fig. 4b). The predictor was independently associated with 7-year recurrence free survival (HR = 3.99, 95% CI 1.77–8.99, Cox regression $p = 8.29 \times 10^{-4}$) when including common clinicopathology variables, including TNM stage (Supplementary Fig. 7c). When ranked by feature importance in the model, the top 3 genes in the predictor were from cluster 3 and included *MUC16*, *NOTCH3*, and *H19*, but only cluster 2 genes were significantly enriched among the top ranked genes overall (two-sided Kolmogorov-Smirnov test $p = 0.02$; Fig. 4C). The top features for clusters 1, 2, and 4 were *MKI67*, *COL1A1* and *CHD2*, respectively.

The predictor retained a similar performance in predicting VI$^+$ LUAD when applied to our independent validation cohort (Table S5; $n = 42$ VI$^-$, $n = 17$ VI$^+$) with an AUROC of 0.86 (95% CI 0.76–0.96, $p = 6.16 \times 10^{-6}$) (Fig. 4d). It retained equivalent performance even after excluding LMP tumors (NST vs. VI only, AUROC 0.86, 95% CI 0.75–0.96, $p = 1.22 \times 10^{-5}$) (Fig. 4e). It was associated with histologic grade (Supplementary Fig. 7d) and 7-year recurrence free survival (HR = 1.98, 95% CI 1.09–3.60, Cox regression $p = 0.025$) (Fig. 4f). Additionally, it was also a significant predictor of RFS among VI$^-$ LUAD tumors (HR = 2.76, 95% CI 1.41–5.41, Cox regression $p = 0.003$), and in tumors without VI, necrosis was associated with higher VI predictor scores in a univariate linear model (p.adj = 0.003) (Fig. 4g). In our validation set, the VI predictor retained good performance across tumor subsets that varied by histological growth patterns, with the best performance in samples that contained no lepidic pattern (0% lepidic, $n = 22$, AUROC 0.96, 95% CI 0.89–1.0, $p = 1.18 \times 10^{-4}$) and the worst performance in samples that contained no solid pattern (0% solid, $n = 34$, AUROC 0.70, 95% CI 0.48-0.93, $p = 0.198$) (Fig. 4h). It also showed variable performance in predicting other invasion types, and notably was not significantly better than a random classifier at detecting LI$^+$ tumors (AUROC 0.53, 95% CI 0.38–0.69, $p = 0.677$) or LI$^+$VI$^-$ tumors (AUROC 0.50, 95% CI 0.31–0.69, $p = 1.0$) (Fig. 4i), supporting our hypothesis that vascular invasion is a different molecular process than lymphatic invasion. Although we did not have separate VI annotations for published LUAD RNA-seq datasets, in TRACERx, a longitudinal study of NSCLC, we observed a difference in predictor scores between LVI$^+$ and LVI$^-$ LUAD ($p = 0.0014$) (Supplementary Fig. 7e). Interestingly, VI predictor scores were higher in tumors from patients with detected preoperative ctDNA ($p = 5.9 \times 10^{-4}$;

Supplementary Fig. 7f). Higher VI predictor scores were significantly associated with decreased 5-year RFS in TRACERx stage I LUAD (Fig. 4f; HR = 1.69, 95% CI 1.04–2.75, Cox regression $p = 0.036$) but were not significantly independent of TNM stage IA and IB (HR = 1.60, 95% CI 0.97–2.65, Cox regression $p = 0.066$). However, the predictor was significantly associated with 5-year RFS independently of TNM stage I and II (Supplementary Fig. 7g; HR = 1.46, 95% CI 1.07–2.0, Cox regression $p = 0.018$). Additionally, elevated VI predictor scores remained significantly associated with decreased 5-year RFS when including surgery type (lobectomy or sublobar resection) and adjuvant therapy as covariates in stage I (HR = 1.66, 95% CI 1.02–2.71, multivariate Cox regression $p = 0.041$; Supplementary Fig. 7h). Finally, higher VI predictor scores associated with decreased overall survival in stage IA LUAD tumors from both The Cancer Genome Atlas (TCGA) ($n = 127$) (HR = 1.72, 95% CI 1.05–2.81, Cox regression $p = 0.032$) and Uppsala ($n = 44$) (HR = 2.11, 95% CI 1.05–4.27, Cox regression $p = 0.039$) cohorts[42,43] (Fig. 4f).

## The VI predictor is robust to intra-tumor heterogeneity and predicts VI when measured in pre-surgical biopsies from stage I LUAD patients

Our stRNA-seq analysis of the expression of VI-associated genes implies that each VI-associated cluster may be spatially distinct. This suggests that a VI predictor combining genes from each cluster might overcome ITH. Overcoming ITH is crucial for molecular biomarkers that sample only a portion of the tumor volume, such as those measured on tissue available from biopsies[44]. This is particularly important for detecting VI due to the small size of invaded vessels. When we divided stRNA-seq spots from VI$^+$ samples into distal VI$^+$ (spots >1 mm beyond the edge of invaded vessels) and proximal VI$^+$ (spots <1 mm from and including invaded vessels) (Fig. 5a), we found a significant difference in the enrichment score between VI$^-$ and distal VI$^+$ tumors for the predictor genes ($p = 0.028$) (Fig. 5b). This supports the hypothesis that the predictor may be able to detect VI away from the invasive focus. Next, we used RNA-seq from multi-region sampling of stage I LUAD ($n = 63$) in TRACERx to evaluate the variability of the VI predictor to random tissue sampling in a larger cohort[32]. Although a label for VI was not available for the TRACERx tumors, we observed strong correlation between predictor scores from two randomly selected paired regions ($n = 126$) across all tumors ($R = 0.87$, $p < 2.2 \times 10^{-16}$) (Fig. 5c). When we filtered the dataset to remove lowly expressed genes and ranked the remaining 15,800 genes in the dataset by correlation between paired regions, the 48 genes present in our predictor were significantly enriched among those with the least ITH ($p = 2.89 \times 10^{-6}$) (Fig. 5d). Furthermore, inter-tumor heterogeneity of the VI predictor score (as measured by the absolute difference of

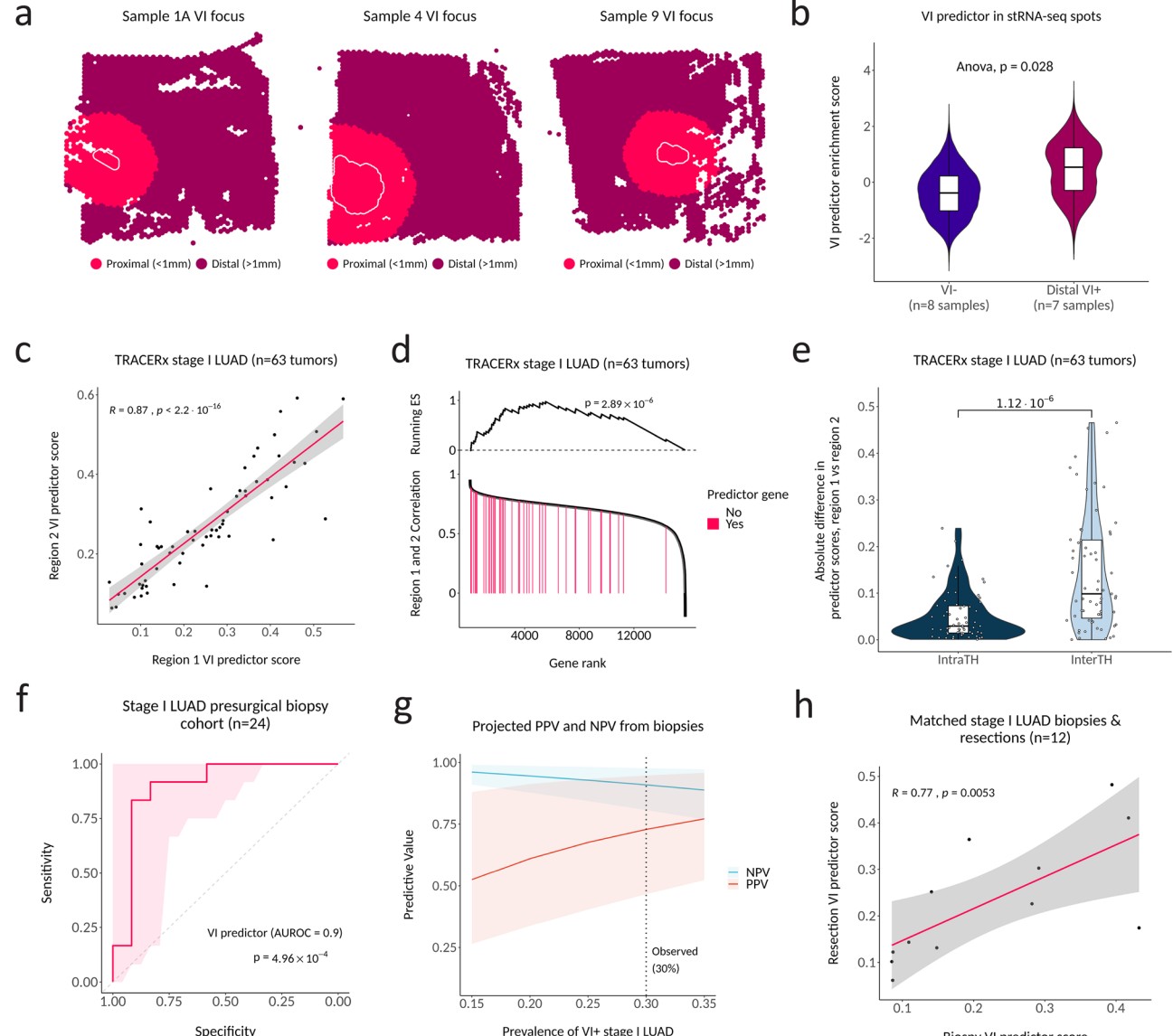

**Fig. 5 | The VI predictor is robust to intra-tumor heterogeneity and predicts VI when measured in pre-surgical biopsies from stage I LUAD patients. a** Examples of binning VI⁺ tumors into distal VI⁺ (spots >1 mm outside the invaded foci boundary) and proximal VI⁺ (spots <1 mm from and including the invaded foci). Any VI⁺ tumors that did not contain VI foci in the capture area were categorized as distal VI⁺. **b** Spot-wise expression of the VI predictor (enrichment of the VI-increased predictor genes minus enrichment of the VI-decreased predictor genes) between distal VI⁺ and VI⁻ tumors. Boxplots display the median, interquartile range (25th–75th percentiles), and whiskers extending to 1.5× the interquartile range of the per-spot VI predictor scores. Violin plots depict the distribution. *P* values are derived from a type II ANOVA of a linear mixed model predicting spot-wise VI predictor scores as a function of VI status with sample as a random effect. c. Intratumoral concordance of VI predictor scores across spatially distinct tumor regions in stage I LUAD from TRACERx (*n* = 63 tumors). For each tumor, two regions were randomly sampled. Points represent the VI-predictor score in tumor-matched region pairs. The solid line shows the linear regression fit, with the shaded region indicating the 95% CI. Spearman rank correlation coefficient and two-sided *P* value are shown (*P* < 2.2 × 10⁻¹⁶; machine precision limit). **d** Enrichment of the 48 VI predictor genes among all filtered genes ranked by expression correlation between two randomly selected regions of the same tumor. Pre-ranked Gene Set Enrichment Analysis (GSEA) was performed using a weighted Kolmogorov-Smirnov-like statistic; significance was evaluated using two-sided permutation. **e** Absolute difference in VI predictor scores between unmatched regions (inter-tumor heterogeneity) and

matched regions (intra-tumor heterogeneity). For inter-tumor heterogeneity, the absolute difference was calculated between two regions that were randomly sampled from two different tumors and repeated 63 times to match the number of intra-tumor comparisons. Boxplots are defined as in (**b**). Violin plots depict the distribution of absolute differences, and points represent individual comparisons. *P* value was calculated using a two-sided Wilcoxon rank-sum test. **f** Predictor performance in the stage I LUAD pre-surgical biopsy cohort (*n* = 24). ROC curve showing discrimination between VI⁺ and VI⁻ tumors. The diagnosis of VI was made on the matched resected tumors. The shaded region denotes the pointwise 95% confidence interval of sensitivity at fixed specificity, computed using stratified bootstrap resampling (2000 replicates). Statistical significance was assessed using a two-sided Wilcoxon rank-sum test. **g** Projected positive and negative predictive values (PPV and NPV) of the VI predictor across a range of VI prevalences in stage I LUAD. Lines indicate posterior median PPV and NPV at each prevalence. Shaded regions denote the 95% Bayesian credible intervals, centered on the posterior median and obtained by Monte Carlo propagation of uncertainty in sensitivity and specificity estimated from the biopsy cohort. **h** Correlation between VI predictor scores measured in pre-surgical biopsies and matched resected tumors (*n* = 12 pairs). Each point represents a matched biopsy–resection pair. The solid line indicates the linear regression fit, with the shaded region representing the 95% confidence interval. Spearman rank correlation coefficient and two-sided *P* value are shown. Source data are provided as a Source data file.

scores from regions of two randomly selected tumors) was significantly higher than ITH (as measured by the absolute difference of scores from two randomly selected regions within the same tumor) ($p = 1.12 \times 10^{-6}$) (Fig. 5e). Taken together, these results support our observations made from the stRNA-seq data that our VI signature represents a global shift in molecular aggressiveness toward an angioinvasive phenotype and that these changes can be leveraged to predict VI+ LUAD from any region of the tumor.

Finally, to demonstrate a proof of concept for how our VI predictor may be applied in routine clinical workflows, we performed RNA-seq on an independent cohort of formalin-fixed paraffin-embedded (FFPE) pre-surgical biopsies (Table S6; $n = 12$ VI−, $n = 12$ VI+), collected from stage I LUAD patients treated at a separate institution from our discovery or validation cohorts. The VI predictor measured in these presurgical biopsies predicted the detection of VI in the resected tumor with an AUROC of 0.90 (95% CI 0.75–1.0, $p = 4.96 \times 10^{-4}$) (Fig. 5f). The sensitivity and specificity when using the Youden point from the discovery cohort as a cutoff in the biopsies was 0.83 (95% CI 0.52–0.98) and 0.92 (95% CI 0.62–1.0), respectively. Following standard methods for estimating diagnostic accuracy[45], we derived posterior distributions for sensitivity and specificity from the biopsy cohort's classification counts. We then used Monte Carlo simulations and applied Bayes' theorem to project positive predictive value (PPV) and negative predictive value (NPV) across a realistic prevalence range (15–35%). The VI predictor's prevalence-adjusted NPV estimate from the biopsies was high and precise (median 91%, CI 81–98% at 30% prevalence) (Fig. 5g). Because our balanced biopsy cohort design under-samples true negatives relative to general clinical prevalence, these CIs are conservative and would gain further precision in an unselected cohort. Conversely, the prevalence-adjusted PPV estimates from this biopsy cohort are less precise (median 73%, CI 47–95% at 30% prevalence), reflecting both the small number of VI+ biopsy cases and the moderate prevalence of VI+ stage I LUAD. In addition, we observed a strong correlation between VI predictor scores determined from presurgical biopsies ($n = 12$) and matched resected tumors ($r = 0.77$, $p = 0.0053$) (Fig. 5h).

## Discussion

Vascular invasion (VI) is a hallmark of tumor progression and a strong independent predictor of recurrence in stage I LUAD, suggesting that VI might result in considerable alterations in gene expression. To interrogate this, we used RNA-seq and stRNA-seq to profile new patient cohorts of stage I LUAD tumors that had undergone detailed pathologic assessments. We identified a gene expression signature of VI in stage I LUAD that is replicated across datasets, patient cohorts, and sequencing technologies. In situ, components of the VI gene signature can be localized to histopathologic structures outside of the invaded vessel. A VI predictor developed from this signature discriminates between VI+ and VI− tumors in an independent validation cohort and is independent of histologic growth pattern. We also demonstrated that the VI predictor is robust to intratumor heterogeneity (ITH) and predicts VI when measured in clinical pre-surgical biopsy samples. This property has the potential to be exploited to develop a new preoperative biomarker for angioinvasive stage I LUAD that could enable a more tailored approach to surgical management.

Gene expression alterations associated with VI clustered into four distinct patterns of co-expression with unique biology. Gene clusters that had increased expression in VI+ tumors were enriched for cancer hallmarks, including cell cycle (Cluster 1), EMT/angiogenesis (Cluster 2), and hypoxic tumor metabolism (Cluster 3), indicating that angioinvasive tumors tend to be more proliferative, invasive, and hypoxia-adapted. Tumor hypoxia is well documented as a tumor phenotype with poor prognosis across multiple cancer types and has been shown to be moderately associated with solid histopathology in LUAD[46,47]. The elevated expression of hypoxia-responsive genes in VI+ tumors suggests that oxygen deprivation and related stress responses (e.g., HIF1a-driven pathways) are part of the aggressive biology facilitating blood vessel invasion. In contrast, genes that were decreased in VI+ tumors (Cluster 4) included genes involved in tumor-suppressive functions (cell adhesion, immune response), consistent with the loss of constraints on invasion in these cancers.

Lymphovascular invasion (LVI) is a commonly reported clinicopathologic entity, but our data demonstrated that its two components – VI and lymphatic invasion (LI) – have distinct biological profiles. We also observed a greater number of transcriptional changes associated with VI than LI, and distinctions between VI and LI-associated gene expression supported previous reports[8,24] that stage I LUAD VI and LI are distinct histologic categories with differing effects on patient outcome[6]. Specifically, the EMT/tissue-remodeling cluster (Cluster 2) was elevated in VI+ tumors but not in LI+ tumors. This disparity may reflect the more extensive tissue remodeling required to invade the thicker muscular vessel wall relative to thin-walled lymphatic vessels that can be breached with less matrix remodeling. Consistent with this, Cluster 2 genes in our spatial analysis were concentrated in regions of desmoplastic stroma at the invasive front (see discussion below), highlighting a close interplay between tumor cells and the surrounding matrix during VI. LI+ tumors did share increased cell-cycle and hypoxia signatures and reduced adhesion genes, indicating some overlap. While our results favor VI and LI as biologically distinct processes, we acknowledge that they could be interrelated steps in tumor progression; longitudinal studies will be needed to resolve this. Future efforts may also seek to separate additional biological signals or tumor subtypes that are associated with LUAD lymph node spread, given that tumor LI is not a reliable indicator of lymph node metastasis.

To determine whether the VI gene signature reflects merely the immediate vicinity of invaded vessels or a tumor-wide propensity for angioinvasion, we leveraged stRNA-seq. We found that Cluster 2 and 3 expression within tumor regions exhibiting aggressive growth patterns was significantly increased in VI+ compared to VI− tumors. Cluster 4 (tumor-suppressive genes) expression was localized to non-aggressive growth patterns and normal lung structures. In contrast, Cluster 2 expression (EMT/angiogenesis genes) was localized to desmoplastic stroma within the tumor. Desmoplastic stroma is a natural response of surrounding normal tissue to invasive tumor nests and involves widespread tissue remodeling[48]. Accordingly, our spatial in silico cell-type deconvolution showed a marked enrichment of cancer-associated fibroblasts (CAFs) in regions of the tumor with high Cluster 2 expression, which we validated by both ISH and IHC staining. Stage I LUAD tumors with VI had significantly higher proportions of activated myofibroblast CAFs than those without VI, a difference we observed in both the stRNA-seq data and our bulk RNA-seq cohort. Strikingly, the spatial co-localization of Cluster 2 with CAF-rich regions was the strongest among all clusters, whereas Clusters 1, 3, and 4 correlated instead with epithelial tumor-cell–rich regions. Subsequent IHC analysis in an independent LUAD cohort of resected tumors validated the stRNA-seq findings and clarified the cell type expression of Cluster 2. In both non-desmoplastic and desmoplastic stroma regions, nearly all POSTN+ myofibroblasts co-expressed COL1A1, a key Cluster 2 gene. The fraction of these POSTN+COL1A1+ myofibroblasts increases in non-desmoplastic stroma regions of VI+ tumors relative to VI− tumors. In addition, desmoplastic stroma, which comprises a greater fraction of the tissue area in VI+ tumors, contains COL1A1+POSTN− stromal cells, raising the possibility that additional stromal cell populations contribute to ECM remodeling. This hypothesis, and the relative contribution of these cells relative to the contributions of POSTN+COL1A1+ myofibroblasts, will require mechanistic validation, and it remains to be determined if the proportion of these COL1A1+POSTN− cells within desmoplastic stroma is modulated by vascular invasion. The

association between VI and desmoplastic stroma has been documented in thyroid carcinoma[49], and desmoplastic stroma has been associated with higher rates of distant metastasis or other prognostically unfavorable histologic features in other cohorts[50,51]. Collectively, these data suggest that aspects of VI-associated gene expression occur distally to the site of VI.

We developed a VI predictor that strongly discriminated between VI+ and VI− tumors in an independent validation cohort and was independent of tumor histologic pattern. The top weighted genes in this predictor belonged to Cluster 3 and included *MUC16*, *NOTCH3*, and *H19*. *MUC16*, which encodes CA125, enhances LUAD aggressiveness through downstream inactivation of *p53*[52]. Although serum CA125 is used as a diagnostic biomarker for patients at high risk of ovarian cancer, its association with lung cancer prognosis has not been demonstrated[53]. NOTCH pathway activity is associated with poor LUAD patient prognosis[54], and *NOTCH1* is essential for LUAD tumor cell survival under hypoxia[55]. Depleting *NOTCH3* in LUAD also leads to reduced tumor cell invasion and significantly reduces the expression of *COL1A1*[56], the top Cluster 2 gene in our VI predictor. The noncoding RNA *H19* locus is a tumor suppressor[57]. In lung cancer, it is induced by the oncogene *Myc*, which drives tumorigenesis by binding near to the *H19* imprinting control region[58]. *H19* also represses *CDH1*, which is responsible for maintaining cell-cell adhesion[59]. The top Cluster 1 gene, *MKI67*, is expressed by proliferating cells and is integral to carcinogenesis[60]. Finally, *CHD2*, the top Cluster 4 gene in our predictor, is a chromatin remodeler that, along with *CHD1* and *CHD5* belongs to a family of tumor suppressor genes implicated in various cancers[61–63], consistent with the decreased expression observed in VI+ LUAD. Collectively, many of the highly weighted genes in the VI predictor are implicated and may be accomplices in LUAD progression.

The VI predictor was also associated with RFS. Historically, prognostic gene signatures in lung cancer have suffered from poor reproducibility and lack of independence from known and more readily ascertained risk factors[21,64]. As an alternative to developing a molecular predictor of outcome that is a complex and context-specific phenotype, our approach was to develop a predictor of a specific invasive feature that determines the histopathologic grade of stage I LUAD. Similar strategies have had substantial clinical utility, for example, in guiding breast cancer management[65]. Interestingly, the VI predictor was associated with RFS even in tumors without detectable VI, where it was most associated with necrosis. This indicated that our transcriptomic predictor captures critical aspects of tumor aggressiveness that are not apparent from pathology alone. In our dataset, necrosis was the histologic feature most strongly associated with the predictor in VI− cases, suggesting a link between the molecular signature and severe tumor hypoxia. This is consistent with clinical evidence that tumor necrosis is an adverse prognostic factor in resected stage I NSCLC[66]. We propose that necrosis in these tumors is a surrogate for the same aggressive biology that leads to VI: a highly proliferative tumor outstrips its blood supply, inducing hypoxia-driven invasive programs, as implicated by the increased expression of hypoxia-associated genes in VI cluster 3. In support of this concept, experimental models have directly connected necrosis with VI and dissemination. An in vivo model of breast cancer demonstrated that *ANGPTL7* is a tumor-secreted factor that synchronously triggers the formation of a necrotic core, increases vascular permeability, induces the formation of VI foci, and leads to circulating tumor cells and metastasis[67]. In our study, ANGPTL7 was not part of the VI signature, but we found that the VI predictor score is positively associated with necrosis among the VI− tumors in our validation cohort and positively associated with detectable preoperative circulating tumor DNA in the TRACERx cohort, suggesting a similar relationship between necrosis and vascular invasion in early-stage LUAD. This parallel suggests that our gene signature is identifying VI− tumors with an unrecognized angioinvasive phenotype.

We anticipate that molecular prediction of VI using resected LUAD tumor tissue may improve the reproducibility of histopathological grading and thus prognostication. We show that the VI predictor has a strong association with patient prognosis and is independent from TNM staging; however, larger prospective studies will be necessary to determine if it can stratify clinical stage IA or IB by outcome. Although our focus is on VI as an intrinsic feature of tumor biology, the eventual clinical value of a VI predictor will also depend on extrinsic factors that influence recurrence risk, including surgical margins and the adequacy of nodal sampling. Our analysis of the multi-region sampling data from the longitudinal TRACERx cohort suggested that the VI predictor was also robust to ITH and may be suitable for use in tissue biopsies. The ability to predict VI from preoperative lung cancer biopsy tissue could inform pre-surgical treatment decisions and guide the extent of surgical resection[68]. We demonstrated in a pilot study that when measured in small FFPE pre-surgical biopsy specimens, the VI predictor has strong performance for predicting the histopathological detection of VI at resection. Despite our small, preselected biopsy cohort, we found that the signature's prevalence-adjusted NPV estimate was high and precise, suggesting that a negative VI call could be used to reliably recommend sublobar resection given the low likelihood that VI will be detected on final pathology. In contrast, the PPV estimate was less precise, so we are unsure that a positive VI call will be sufficient to recommend lobectomy. Larger multi-center prospective studies will be needed to refine these estimates and demonstrate the predictor's diagnostic accuracy and clinical utility. In addition, further work is required to determine the required fraction of tumor cells in the presurgical biopsies for accurate biomarker assessment.

Our biopsy results support the technical feasibility of predicting VI from presurgical tissue. After a pathologist reviews H&E slides of the pre-surgical biopsy, the remaining FFPE material can be used directly for RNA extraction and sequencing, enabling compatibility with existing clinical workflows. Although the VI predictor was derived and validated using RNA-seq, future work should also evaluate its performance when assessed using more widely available clinical platforms, such as qPCR or targeted probe panels. None of the cohorts we analyzed received neoadjuvant therapy, but preoperative detection of VI+ tumors may, in the future, also provide the opportunity to guide these treatments, which are increasingly being evaluated in earlier stages of resectable NSCLC[69]. Concordance of WHO grade between LUAD pre-surgical biopsies and resected specimens is poor[70,71]. Prospective studies should further assess the agreement of VI estimation from biopsy and resected tumor material using both histopathology grading and molecular prediction. Integration of the VI predictor with other biomarker modalities, such as radiomics[72] and pathomics, is likely to further improve risk stratification of stage I LUAD.

## Methods
### Clinical cohorts
**Discovery cohort.** A discovery cohort consisting of 192 resected tumors from 192 patients with 8th edition TNM stage 0/1 LUAD and not treated with neoadjuvant or adjuvant therapy was included in this study. Cases were from Boston Medical Center (BMC) and Lahey Hospital & Medical Center (LHMC). The BMC and LHMC Institutional Review Boards approved this study (BU/BMC IRB H-37859; Lahey Clinic IRB-518308). Relevant clinical information on all patients included can be found in Table S1. For all clinical cohorts, race was obtained from the electronic medical record and reflects patient self-reported categories. The 192 tumors represented a wide range of sample ages, so a small pilot batch of 12 samples was selected to determine which tumors might yield useful data (4/5 of the samples removed post-QC belonged to this pilot batch). We then selected an additional 96 samples that yielded at least 83 ng of library. RNA-seq was performed on these 108 tumors, with 103 passing quality control. Relevant clinical information on all patients can be found in Table S2.

**Validation cohort**. An independent validation cohort consisting of 61 resected tumors from 60 patients with 8th edition TNM stage I/II LUAD and not treated with neoadjuvant or adjuvant therapy was included in this study. All samples were selected to be stage IA/IB at the time of collection, except for one tumor that was upstaged to stage IIA under the 8th TNM edition. Cases were from LHMC. RNA-seq was performed on 61 tumors, with all passing quality control. StRNA-seq was performed on a subset of these 61 tumors and included 16 samples taken from 14 tumors (13 stage I tumors and 1 stage II tumor). Relevant clinical information on all stRNA-seq samples included can be found in Table S3. Only the 8th edition TNM stage I LUAD from the validation cohort was included in the bulk RNA-seq ($n = 60$) analysis. Relevant clinical information on all patients included can be found in Table S5.

**IHC cohort**. An independent cohort consisting of 20 resected tumors from 20 patients with 8th edition TNM stage I LUAD was included in this study. All samples were from patients not treated with neoadjuvant or adjuvant therapy and were receiving care at Inova Schar Cancer Institute. The Inova Institutional Review Board approved this study (IRB U23-06-5093). Relevant clinical information on all patients included in this cohort can be found in Table S4.

**Pre-surgical biopsy cohort**. RNA-seq was performed on an independent validation cohort consisting of 24 pre-surgical biopsies and 12 matched resected tumors from 24 patients with 8th edition TNM stage I LUAD. All samples were from patients not treated with neoadjuvant or adjuvant therapy and were receiving care at Inova Schar Cancer Institute. The Inova Institutional Review Board approved this study (IRB U23-06-5093). FFPE pre-surgical biopsy samples were derived from needle biopsy, forceps biopsy, or fine needle aspiration. The determination of novel grade was based upon a representative FFPE block of the resected tumor. Relevant clinical information on all patients included in this cohort can be found in Table S6.

## Pathology review

An experienced thoracic pathologist (E.B.) reviewed all pathology cases. An inter-reader reproducibility assessment was previously performed on these pathology cases and found substantial agreement ($\kappa = 0.71$) among 5 pathologists for the diagnosis of vascular invasion (VI)[13]. VI was defined as luminal invasion of a vein or muscular artery either within or adjacent to the tumor. Tumor proportions of lepidic, acinar, papillary, micropapillary, and solid patterns were assessed in 5% increments with distinction of simple tubular acinar from complex and cribriform acinar patterns. Adenocarcinoma in situ (AIS) was assigned to purely lepidic tumors ≤3 cm, whereas minimally invasive adenocarcinoma (MIA) was diagnosed when non-lepidic foci measured ≤0.5 cm as per WHO criteria. WHO-2021 grade was defined as G1, lepidic predominant with <20% high-grade patterns; G2, acinar or papillary predominant with <20% high-grade patterns; and G3, ≥20% high-grade patterns (solid, micropapillary, and/or complex glands)[73,74]. LMP adenocarcinoma was assigned as previously described[27]. LMP tumors were non-mucinous adenocarcinoma measuring ≤3 cm in total size, with ≥15% lepidic growth, and without nonpredominant high-grade patterns (≥10% cribriform, ≥5% micropapillary, ≥5% solid), >1 mitosis per 2 mm², vascular, lymphatic, or visceral pleural invasion, STAS, or necrosis. AIS/MIA and LMP were analyzed together due to their identical outcome (100% 10-year disease-specific survival). One LMP recurred after wedge resection with a positive surgical margin. The tumor recurred at the staple line and was treated with SBRT, resulting in prolonged survival (over 10 years) without further recurrence or metastasis. NST designation was given for all other tumors not classified as VI or LMP. Stage assignments were retrospectively made using the 8th edition of the American Joint Committee on Cancer (AJCC).

## Statistical analysis

All statistical analysis was performed with R version 4.2.1. Tables were created with the tableone package. P values were corrected for multiple hypothesis testing where appropriate using either Holm-Bonferroni adjustment (abbreviated as p.adj) or false-discovery rate correction (abbreviated as FDR), depending on the default method of the R package used, unless otherwise specified.

## RNA-Seq library preparation, sequencing, and data processing

Total RNA was extracted from FFPE tissue with the Qiagen AllPrep DNA/RNA Universal Kit (discovery, validation cohorts) or Qiagen miRNeasy FFPE kit (biopsy cohort), and exome-targeted sequencing libraries were prepared with the Illumina TruSeq RNA Exome Library Prep Kit (discovery, validation cohorts) or Illumina RNA Prep with Enrichment Kit (biopsy cohort). Sample sequencing was performed on the Illumina NextSeq 500 to generate paired-end 50-nucleotide reads (discovery cohort), Illumina HiSeq 2500 to generate paired-end 75-nucleotide reads (validation cohort), or Illumina NextSeq 2000 to generate paired-end 100-nucleotide reads (biopsy cohort). Basespace (Illumina) was used for demultiplexing and FASTQ file generation. A Nextflow v21.10.6 RNA-Seq pipeline aligned reads using hg38 and STAR[75] v2.6.0c. Counts were calculated using RSEM[76] v1.3.1 using Ensembl v108 annotation. Quality metrics were calculated with STAR and RSeQC[77]. EdgeR was used to compute normalized data (library sizes normalized using the trimmed mean of M-values). Genes with count per million (CPM) > 1 in at least 10% of samples were retained for further analysis. Samples in the discovery and validation cohorts were excluded if the transcript integrity number (TIN)[78] calculated by RSeQC was >2 standard deviations below the mean TIN. Combat-seq[79] was used to correct a batch effect between collection sites (BMC and LHMC) in the discovery cohort. It was also used to correct a batch effect between sequencing batches (batch 1 and 2) within the validation cohort.

## Derivation of VI signature and biological annotation

Genes whose expression is associated with the presence of VI were derived from 103 tumors in the discovery cohort using a negative binomial generalized linear model via edgeR[80]. Our previously published novel grading system (LMP, NST, and VI)[8,13] was the independent variable, and we compared VI and LMP groups using glmLRT(). Differentially expressed genes were identified using a false-discovery rate (FDR) threshold of 0.01. Four gene expression clusters were identified using the Ward2 method of hierarchical clustering of genes and samples based on Euclidean distance[81]. When we explored other values for $k$ ($k = 2$–10 clusters) using consensus clustering, we found that clustering stability showed substantial improvement (large change in delta area under the CDF) up to but not beyond $k = 4$. Additionally, $k > 4$ did not return distinct significant biological enrichment terms for all clusters, suggesting a lack of biologically distinct gene co-expression with $k > 4$. The top ten biological processes and pathways enriched in each of the four VI clusters were identified using Enrichr[82] with queries to the following databases: MSigDB Hallmark 2020 and GO Biological Process 2021. To compare VI and LI-associated gene expression, a VI signature was first rederived after adding LI as a covariate to the edgeR model described above, and differentially expressed genes were again identified using an FDR threshold of 0.01 when comparing VI and LMP groups. To identify genes associated with LI, contrasts were set to LI using makeContrasts() and differentially expressed genes identifying using an FDR threshold of 0.01. Genes associated with LI were ranked by $-\log_{10}(p \text{ value}) * $ the sign of the logFC. The VI gene cluster enrichment against this ranked list was calculated using the GSEA() function from the clusterProfiler[83] R package, and $p$-values were adjusted with p.adjust() for Bonferroni correction.

## Spatial transcriptomics library preparation, sequencing, and data processing

Sixteen samples from 14 patient tumors within the validation cohort were selected for 10× Genomics Visium for FFPE spatial whole-transcriptome profiling. All samples were selected to be stage IA/IB at the time of collection, except for one that was upstaged to stage IIA under the 8th TNM edition (this sample was excluded from the bulk RNA-seq validation cohort when evaluating the performance of VI predictor scores). Samples were chosen based on (1) the presence of pathological features of interest (e.g., VI foci, representative LUAD histologic patterns) and (2) more than 50% of RNA fragments being greater than 200 nucleotides (DV200) after extraction with the AllPrep DNA/RNA Universal Kit (Qiagen) during bulk RNA-seq library preparation. New 5 µm tissue sections were cut by a microtome, placed in a 42 °C RNase-free water bath, and manually transferred onto 6.5 × 6.5 mm tissue capture areas on Visium Spatial Gene Expression Slides (PN-1000185, 10× Genomics). Tissues were deparaffinized, H&E stained, imaged with a Leica Aperio AT2, and de-crosslinked according to the manufacturer's recommended protocol, with the following user modifications made to deparaffinization to prevent tissue detachment: (1) the 15 min incubation step during deparaffinization was removed, (2) after immersion in 96% ethanol, one 85%, one 70% and one 50% ethanol immersion for 3 min each was added. Probe hybridization was performed with the Visium Human Transcriptome Probe Kit (PN-1000364), followed by probe ligation, extension, and elution for downstream qPCR cycle determination and 10× library construction with the Visium FFPE Reagent Kit (PN-1000362). Sequencing was done on an Illumina NextSeq 2000. The 10× SpaceRanger pipeline was used to demultiplex and generate FASTQ files. FASTQ files were input into the SpaceRanger FFPE count algorithm along with matching H&E.tiff images for alignment to the reference probe set and generation of count matrices for each Visium capture area.

## Spatial transcriptomics pathology annotations

H&E images of sections captured for stRNA-seq analysis were annotated for pathological features of interest by an experienced thoracic pathologist (EB) using Loupe Browser v6 (10x Genomics).

## Spatial transcriptomics data analysis

stRNA-seq data were processed using the Seurat R package, unless otherwise specified[84]. Sample 6 was removed due to tissue loss that occurred during wash steps. Low-quality spots and spots not covered by tissue were filtered if <250 genes were detected per spot. For the global sample analysis, samples were merged and normalized using SCTransform and scaled[85]. For individual stRNA-seq samples, spots were log-normalized. Spots were scored with gene signatures using the AddModuleScore function from the Seurat R package. Spatial differences in VI cluster expression between pathology regions were assessed with a generalized binomial linear mixed effect model of VI cluster enrichment score as a function of spot pathology annotation with sample as a random effect. Pathology regions with <200 spots annotated were removed prior to analysis, and all regions were downsampled to 200 spots each. Normal lung was used as the reference factor level. In order to bin spots into distal VI⁺ and proximal VI⁺, image annotation screening on the invaded vessel pathology annotations was performed with the plotSurfaceIAS() function from the SPATA2[86] R package with a distance and bin width of 1 mm.

## Spot deconvolution with scRNA-seq data

Nuclear segmentation of high-resolution H&E images of tissue used for stRNA-seq analysis was performed with the StarDist[38,39] algorithm using default settings with a prob_thresh set to 0.3. Per-spot cell number estimates were generated within Squidpy[87]. Spot deconvolution was performed with the CytoSPACE[36] algorithm using the per-spot cell numbers as input and the Salcher stage I LUAD atlas[37] as the

reference scRNA-seq dataset. The Salcher atlas was downloaded from https://zenodo.org/records/6411868 and was filtered to stage I LUAD samples only. Spatial differences in cell type predictions between pathology regions were assessed with a generalized binomial linear mixed effect model of cell type proportion as a function of pathology annotation with sample as a random effect, after averaging per-spot cell type proportions within a given region annotated as the same pathology. Pathology regions with <200 spots annotated were removed prior to analysis, and all regions were downsampled to 200 spots each. Normal lung was used as the reference factor level.

## Deconvolution of bulk RNA-seq data

Deconvolution of the bulk RNA-seq discovery cohort was performed with CIBERSORTx[88] using the Salcher stage I LUAD atlas as input to the signature genes matrix generation step. The signature genes matrix was created by taking the average expression for each cell type per sample in the atlas.

## Spatially weighted correlation analysis

Spatial transcriptomic spots cannot be assumed to be spatially independent; therefore, spatially weighted regression analysis is used to calculate the correlation between gene signatures. Unlike simple linear regression, which produces a single coefficient per variable, spatially weighted regression allows for coefficients to change locally with tissue coordinates. Data points closer to the coordinates at a given location within a set window (defined by the bandwidth) are given more weight within the model. This window can then slide over the whole tissue to calculate local correlation coefficients per-spot between variables. First, we used Seurat's AddModuleScore to generate per-spot expression scores for our VI gene cluster signatures and cell type signatures from the Salcher atlas. Cell types that did not have at least one predicted cell by deconvolution across at least 20% of stRNA-seq capture areas were considered too sparse and were filtered out. Cell type expression signatures were generated for each of the remaining cell types annotated in the atlas using Seurat's FindAllMarkers function with default settings. We selected the top 50 most significantly differentially expressed genes for each cell type. Spatially weighted regression was performed for each sample using the gwss function from the GWmodel[89] package 2.3.1 with default settings and a bandwidth set to five. The mean Spearman's rho of each correlation pair across all tissue spots was then taken for each sample and finally averaged across all samples.

## In situ hybridization and immunohistochemistry

Chromogenic immunohistochemistry (IHC) and RNA in situ hybridization (ISH) were performed on a Ventana Discovery ULTRA (Roche Diagnostics). IHC staining for Thy1/CD90 was performed with a rabbit monoclonal antibody (D3V8A, #13801, Cell Signaling) at a 1:100 dilution after heat-induced epitope retrieval in alkaline buffer. IHC for COL1A1 was performed with a rabbit monoclonal antibody (E8F4L, #72026, Cell Signaling) at 1:200 dilution after heat-induced epitope retrieval in alkaline buffer. Primary antibody detection was performed using Discovery Hapten Detection with Discovery Anti-Rabbit HQ (#760-4815, Ventana) and visualized with the DAB chromogen (#760-124, Ventana). RNA ISH for COL1A1 was performed using RNAscope using 2.5 VS probe-HS-COL1A1 (#401899, Advanced Cell Diagnostics) using both protease and heat-induced epitope retrieval as per the manufacturer's guidelines. Multiplex IHC was performed using the Ventana Discovery ULTRA platform using a rabbit polyclonal antibody against POSTN (HPA012306, Sigma Aldrich) at a 1:100 dilution and the antibody against COL1A1 at 1:200 dilution after heat-induced epitope retrieval for 24 min at 95 °C using alkaline buffer (CC1, #950-124, Ventana). Primary antibody detection was performed using Discovery Hapten Detection with Discovery Anti-Rabbit HQ (#760-4815, Ventana), and visualized with purple (#760-229, Ventana) and yellow

(#760-239, Ventana) chromogens for COL1A1 and POSTN, respectively. Primary antibody inactivation for the multiplex reactions was carried out using temperature-induced denaturation for 8 min at 95 °C with CC2 buffer (#950-123, Ventana).

### Spatial alignment of ISH/IHC data to stRNA-seq data

Individual ISH or IHC images were registered separately to the matching adjacent original stRNA-seq H&E image via affine transformation with annotated landmarks using the STalign[90] python tool. The same affine transformation was used to align the stRNA-seq spot coordinate grid to the ISH or IHC image. Affine-aligned and filtered stRNA-seq spots were then imported into QuPath v0.5.1[91]. Only spots that were present over tissue in both images and passed QC in the stRNA-seq pipeline were included. QuPath positive cell detection was used to quantify the number of ISH and IHC positive cells at defined staining thresholds to maximize signal-to-noise ratio. For correlation analyses between ISH/IHC and stRNA-seq, we calculated the mean IHC or ISH positive cell fraction, smoothed by k-nearest neighbors, with $k = 7$ (representing a single spot and its immediate neighbor spots). Because the images were not exact serial sections, this approach was applied to smooth minor differences in spot alignment.

### Dual IHC of POSTN/COL1A1 data analysis

QuPath v0.5.1 positive cell detection was used to quantify the single positive and double positive cells across all samples. Color deconvolution vectors were defined from representative single-stained IHC slides for POSTN and COL1A1, and identical intensity thresholds were applied to all images across the dataset. Pathologist-annotated regions of non-desmoplastic stroma were used for the comparison of VI+ and VI− tumors. For both non-desmoplastic stroma and desmoplastic stroma analyses, tumor nests and lymphoid aggregates within these regions were manually excluded from total cell counts.

### VI predictor derivation

A nested cross-validation pipeline (Fig. 4A) was used to evaluate the performance of different machine learning models using only the discovery cohort samples and to prevent overfitting model hyperparameters. To accomplish this, samples from the discovery cohort were first divided into outer train and test sets using a 70/30 split 100 times.

**Feature selection.** Genes with VI-associated expression were derived within each train split as described above for the analysis of the entire discovery cohort: this included both selecting genes with FDR < 0.01 and dividing the genes into four clusters using hierarchical clustering. The size of the gene set used for generating a VI predictor in each fold of the cross-validation was set to 48. To automate feature selection and to ensure that diverse VI-associated gene expression patterns were evaluated, genes were selected within each fold of the cross-validation such that the proportion of genes from each of the four clusters was the same as the proportion of genes in that cluster among all the genes with FDR < 0.01 in that cross-validation fold. The gene set size of 48 was arbitrarily selected to allow for a 10-fold reduction of the size of the original signature in the full discovery cohort. We observed no dependence of gene set size on model performance from 2.5% up to 40% of the signature, which included median gene set sizes from 12 genes to 192 genes.

**Prediction model selection.** The outer train split was then divided into inner train and inner test folds using 5-fold cross-validation. This was to allow for evaluating predefined model hyperparameters (hyperparameter search) without learning information about the test data (the outer test fold) that would be used to inform final model selection. Model selection was based on comparing methods using this 5-fold cross-validation within the inner fold of the train split. This was done using the AutoML interface from the h2o.ai[92] package v.3.40.0.4

and compared models including generalized linear model (GLM), gradient boosting machine (GBM), extreme gradient boosting (XGBoost), and distributed random forest (DRF). The best performing model and hyperparameter combination in the inner test fold was selected to develop a predictor on the outer train split and then applied to the held out outer test set. This was repeated over 100 cross-validation iterations using the discovery cohort. The final prediction model was selected as the model with the highest mean AUROC across all cross-validation iterations. This prediction model method and its associated hyperparameters, together with the feature selection step, were used with the gene expression data from the entire discovery cohort to develop a final 48-gene VI predictor using a binomial logit GLM utilizing ridge regression.

### VI predictor validation

Genes in the validation cohort were subsetted to those in the filtered discovery cohort. The mean and variance of each gene's log-transformed CPM in the validation cohort were adjusted to match the discovery cohort using reference ComBat[93] with the discovery cohort as the reference batch. VI predictor scores were generated using the final 48-gene predictor, and performance was evaluated by AUROC. For evaluating the VI predictor scores by histologic subgroup, samples in the validation cohort were filtered based upon the percentage of annotated histopathologic growth patterns.

### Survival analysis

Univariate and multivariate Cox proportional hazards regression was performed using the survival R package.

### Additional datasets for survival analysis

Pre-processed RNA-seq and matching clinical data were downloaded using a Genomic Data Commons (GDC) query for TCGA LUAD using the TCGAbiolinks[94] R package and filtered to 127 stage IA samples with data on overall survival. Pre-processed RNA-seq and matching clinical data from the Uppsala NSCLC cohort[43] were downloaded using the Gene Expression Omnibus (GSE81089) and filtered to 44 stage IA LUAD samples with overall survival data. Because the novel histopathology grading system is based upon the 8th edition of the TNM stage I samples, and our predictor was derived in TNM 8th edition stage I samples, we focused our analysis on this subset. Therefore, since tumor size was not available for either cohort, we excluded stage IB samples because these included tumors up to 5 cm in previous TNM editions and may have been recommended for adjuvant therapy. Genes in both datasets were subsetted to those in the filtered discovery cohort, and the mean and variance of log-transformed CPM were adjusted to match the discovery cohort using reference ComBat with the discovery cohort as the reference batch.

### TRACERx intra-tumor heterogeneity analysis

Processed RNA-seq data and clinical annotations from the TRACERx NSCLC cohort of multi-region tumor sampling data were downloaded from https://doi.org/10.5281/zenodo.7683605 and https://doi.org/10.5281/zenodo.7603386[32]. Samples were filtered to TNM 7th edition stage I LUAD tumors ≤4 cm with data from at least two regions. Genes were subsetted to those in the filtered discovery cohort, and the mean and variance of log-transformed CPM were adjusted to match the discovery data using reference ComBat with the discovery cohort as the reference batch. VI predictions were generated as described above. To determine intratumor heterogeneity in predictor scores, we calculated the absolute difference in VI predictor score between regions of the same tumor, when all tumors ($n = 63$) were randomly downsampled to two regions each ($n = 126$ regions total). For inter-tumor heterogeneity, the absolute difference was calculated between two regions that were randomly sampled from two different tumors. This was repeated 63 times to make the number of observations equivalent

to the intratumor heterogeneity analysis. Spearman correlation between predictor scores from regions of the same tumor was also calculated. Predictor gene enrichment among all genes ranked by region-region correlation was determined using GSEA.

**Bayesian estimation of predictive values from biopsy cohort**

To estimate the diagnostic performance of the VI predictor in the biopsy cohort, we applied the decision threshold (Youden point) derived from the discovery cohort as a pre-specified cutoff to the pre-surgical biopsy samples. Based on this threshold, we tabulated the number of true positives (TP), false negatives (FN), true negatives (TN), and false positives (FP) among 12 VI+ and 12 VI− biopsy specimens. To model in diagnostic accuracy, we adopted a Bayesian framework in which sensitivity and specificity were treated as random variables with uniform Beta(1,1) priors. Posterior distributions for sensitivity and specificity were therefore Beta(1 + TP, 1 + FN) and Beta(1 + TN, 1 + FP), respectively. We generated 10,000 random draws from each posterior to obtain paired samples of sensitivity and specificity that reflect estimation uncertainty. For each simulated pair, positive and negative predictive values (PPV and NPV) are calculated at varying assumed prevalences of VI+ stage I LUAD (15, 20, 25, 30, and 35%) using Bayes' theorem. For each prevalence level, the median and 2.5th–97.5th percentiles of the simulated distributions were reported as the point estimate and 95% credible interval, respectively.

**Reporting summary**

Further information on research design is available in the Nature Portfolio Reporting Summary linked to this article.

## Data availability

All bulk RNA sequencing data and spatial transcriptomics data generated in this study has been deposited in the NCBI Gene Expression Omnibus (GEO) under series GSE273528 and is publicly available. Additional publicly available data used in this study are available as follows: Uppsala RNA-seq and associated clinical data are available under GEO series GSE81089. TCGA RNA-seq and associated clinical data are available on the National Cancer Institute Genomic Data Commons Data Portal under project TCGA-LUAD. TRACERx RNA-seq and associated clinical data are available on Zenodo under records 7683605 and 7603386. The single-cell lung cancer atlas (LuCA) scRNA-seq data is available on Zenodo under record 6411868. All other data supporting the findings of this study are available within the article and its supplementary information files. Source data are provided with this paper.

## Code availability

The analysis scripts to reproduce the findings reported in this study are archived on Zenodo (https://doi.org/10.5281/zenodo.18395867)[95].

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

## Acknowledgements

This work was supported by R01CA275015, U01CA196408, the John P. Wajda Trust, and the Ellison Foundation.

## Author contributions

D.S. processed raw sequencing data, performed all bioinformatic and statistical analyses, and wrote the manuscript. D.S. and L.S. performed spatial transcriptomics experiments and were supported by S.A.M. L.S. performed ISH and IHC experiments and was supported by E.J.B. D.S., J.Z., K.R.-C., E.J.B., J.B., and M.E.L. conceived the idea for and designed the study. E.J.B., K.S., and S.M. assembled the cohorts and pathological data. E.J.B. performed the pathology review. T.S. led RNA isolation and was supported by K.R.-C. H.L. and X.X. performed RNA QC evaluation and library preparation and were supported by G.L. A.L. performed library preparation and Illumina sequencing and was supported by Y.O.A. J.B. and M.E.L. jointly supervised bioinformatic and statistical analyses and helped edit the manuscript.

## Competing interests

A patent application relating to the VI predictor described in this manuscript has been filed (U.S. Provisional 63/570,598, filed March 27, 2024; PCT PCT/US25/21084, filed March 24, 2025; ISR issued) and is titled *"Methods and compositions relating to angioinvasive lung adenocarcinoma"*. The patent applicants include the authors' affiliated institutions, and the listed inventors include M.E.L., J.B., D.S., E.J.B., J.Z., S.A.M., K.R.-C., and T.S. All other authors declare no competing interests.
