## [Transparent Peer Review file · Nature Communications]

Vascular invasion-associated gene expression is detectable in pre-surgical biopsies of stage I lung adenocarcinoma

Corresponding Author: Dr Marc Lenburg

Version 0:

Reviewer comments:

Reviewer #1

(Remarks to the Author)

Steiner et al report a new set of genes that correlate specifically with vascular invasion in their cohort of patients with stage 1 LUAD. They have found four clusters of genes associated with vascular invasion. They have then explored some other characteristics of these genes, such as their site of expression within the tumors in relation to cellular content and their robustness in regards to intra-tumor heterogeneity, using stRNAseq on 15 samples. They have then used a second 60 patient cohort to validate their findings.

The main issues with this study are as follows:

1-The clinical utility of this set of genes as biomarkers for specific management of patients with stage 1 LUAD: As mentioned by the authors, the main management decision for this group of patients is the decision regarding lobectomy vs segmentectomy and two recent trials have failed to show inferiority of segmentectomy at this decision fork. This decision is made before surgical resection and possibly only based on a biopsy and imaging rather than sampling from the resected tumor. Therefore, a critical missing part from this study is validation of this gene signature in biopsies, rather than resected tumors.

The other clinically relevant point is if and how this gene expression biomarker adds to the universally used TNM staging, rather than the WHO grading of pathology. For instance, one expects to find a statistical analysis showing that adding vascular invasion to TNM staging has had an impact on the patients' survival and that this factor has shown an independent predictive value in a multivariable analysis, as shown in Ma et al, 2023. It is of note that even in the latter study, which is one of the main reference points for this paper, vascular invasion was only one of the factors that showed predictive value and other factors such as tumor size, clean margins, and suboptimal nodal sampling had similar or lower p values for predicting survival

2-Clinical feasibility in a "real world" setting: The proposed 48-gene set is derived from bulk-seq in 100 samples from the firsts cohort and then validated (presumably) using bulk-seq data on 60 tumor samples from the second cohort. Regarding the limitations of bulk seq performance and analysis in clinical settings, how would these 48 gene perform, if measured by more widely available gene probes/qPCR?

3-Pathophysiological significance: Even though the authors go through extensive description of gene expression patterns based on stRNA-seq, the paper does not contain simple validations such as staining for a group of the high-ranking genes in clinical samples to place them within known cell types and possibly imply a function. The best evidence supporting a biomarker is biological/functional validity that will come out of mechanistic experiments, and this kind of evidence is an implied prerequisite for a high impact publication.

The following is a critique of the specific sections:

The main finding in the first part of the study is that among the three pathological patterns defined by the authors, vascular invasion has the highest hazard ratio for RFS. While this analysis is valid internally and a multi-variable analysis shows its independent predictive value, the TNM stage of the tumors is missing from this table.

The study on the correlation with the histopathological feature is entirely based on stRNA-seq on 15 tumor samples that are mixed stage 1 and 2. Authors have tried to study various issues such as intratumor heterogeneity and structural correlates

based on these samples. While interesting, the value and general applicability of statistical reasoning based on findings within these samples are questionable.

The same critique applies to the findings on association of gene clusters with specific cell types. As an acceptable validation step for these correlative hypotheses, one would expect a simple staining showing a specific gene with a known function within a specific cell type, possibly together with a mechanistic experiment that captures at least one aspect of that gene function. In the absence of such data, statements such as “We found a trend toward a higher proportion of peribronchial fibroblasts belonging to VI+ samples (Fig. 3b), consistent with the tissue remodeling biology of VI cluster 2 described above. VI+ samples also had a trend toward a higher proportion of plasma cells and B cells.” remain enigmatic and of questionable value.

In the derivation study: was a multivariable analysis, similar to the one performed for VI, done for the 48-gene set? Wouldn't such analysis be required as the main message of this paper, ie showing if altered expression (levels defined for each gene) of a 48 gene set has an independent predictive value for tumor recurrence or survival.

The point about the distinction between vascular and lymphatic invasion is interesting and gene expression patterns are hinting at possible biological differences. However, to the reviewer, none of these findings prove or disprove whether lymphatic invasion is a stage in the continuum, or a biologically different and unique process with clinical implications. In the validation set, the 48 gene signature is applied to 60 samples, only 17 of which had VI. Would a power calculation for this analysis be appropriate? I cannot find the data for the sentence “It was associated with histologic grade and 7-year recurrence free survival (HR=1.98, 95% CI 1.09–3.60, Cox regression p=0.024) (Extended Data Fig. 6b and Fig. 4f).” In discussion, some of the pathophysiological explanations are not convincing or need more explanation. For instance, why would a group of “cell cycle” or “tumor hypoxia” genes specifically contribute to vascular invasion. Why does cluster 2 (EMT/Angiogenesis) correlate with the presence of CAFs? What is the value of necrosis? How would a gene signature of vascular invasion have prognostic value in VI- tumors?

On stylistic issues, the results section is filled with technical details, and this overrepresentation of details detracts from the main message of the paper and its proposed value as a biomarker of invasiveness. Also, some of the premises of the study are not solid. For instance, authors mention that in the Hattori et al study (2024) “...post-hoc analyses are beginning to show subsets of patients for which lobectomy might decrease the rate of recurrence.” The mentioned study showed that lobar resection was associated with higher locoregional recurrence among the full cohort, but survival was age and sex-dependent. Authors mention that gene signature for vascular invasion has not been studied. A number of studies in the literature including <https://doi.org/10.1016/j.ajpath.2010.10.040>, <https://doi.org/10.1371/journal.pone.0216847>, and <https://doi.org/10.1038/s41416-019-0486-6>, had visited this entity in cancer.

(Remarks on code availability)

Reviewer #2

(Remarks to the Author)

This manuscript reports a lung adenocarcinoma transcriptional signature that is associated with the pathological detection of vascular invasion in resected stage 1 tumors.

The association was confirmed in an independent cohort of 60 patients with AUROC of 0.86. The signature was associated with recurrence free survival and was detectable in regions without invasion in tumors that were annotated as harboring vascular invasion.

The study design is sound and the observations are robust, although not conceptually novel.

1. The determination of vascular invasion was by a single thoracic pathologist. Were these cases previously read by other clinical pathologists and what was the agreement of determination of vascular invasion in cases read by more than one pathologist? The authors should address inter-reader variability concerns with determination of this pathological finding.

2. The introduction emphasizes the potential clinical significance of detection of vascular invasion and suggests that the finding may support use of adjuvant therapy or lobectomy. This is confusing and should be clarified. Further, it is possible that the signature impact on recurrence free survival may be independent of performance of lobectomy or administration of adjuvant therapy. Addressing the potential clinical utility aspects of the proposed classifier and how it can be assessed will be helpful.

3. Figure 3 shows imputed cell type associations with the vascular invasion signature using spatial transcriptomic and bulk sequencing data. It will be important to validate key gene expression signatures in specific cell types using immunohistochemistry.

(Remarks on code availability)

Reviewer #3

(Remarks to the Author)

In their manuscript “Identification of a gene expression signature of vascular invasion and recurrence in stage I lung adenocarcinoma via bulk and spatial transcriptomics”, Steiner et al. developed a transcriptomic signature of microscopic vascular invasion (VI) in stage I lung adenocarcinoma based on a cohort of 163 tumors. The VI-associated expression patterns clustered into four subclasses linked to unique biological processes. The authors utilized spatial transcriptomics to describe the spatial organization of the four gene clusters. The authors additionally developed a predictor of VI in an

independent cohort and evaluated its performance in multiple regions of the same tumor to show the predictor is largely unaffected by intra-tumor heterogeneity. The novel transcriptional signature for vascular invasion in early-stage lung adenocarcinoma presented in this work may enable more precise diagnosis and treatment of patients.

My specific comments which may help to improve the manuscript are (as line numbers are not available, I am referencing the figure panels to identify the relevant section of the manuscript):

Ext. Data Fig. 1a-f: It would be helpful to add outcomes that include both VI and LI classification given that LI is used as a covariate for some of the differential expression analysis.

Ext. Data Fig. 1j: In the legend, "only" is cut-off.

Ext. Data Fig. 1j-k: Is the site of recurrence related to time to recurrence?

Fig. 1b: It may be helpful to have VI/NST/LMP as a single column annotation as they are mutually exclusive and based on the same grading system.

Fig. 1b: The differential expression results and cluster assignments should be provided as supplementary tables or in a data repository.

Fig. 1b: There are two VI samples that are clustering with LMP samples. Are there any biological or technical factors that may explain this grouping?

Fig. 1b: The NST samples are clustering with both VI and LST samples. Can you comment on how to best interpret the two VI classifications?

Fig. 1b: Cluster 1 and, to a lesser degree, cluster 3 appear to be separating the VI+ samples into two subsets. Is there any merit to treating those as independent groups for downstream analysis?

Ext. Data Fig. 3a: It may be helpful to add a brief explanation of the biological significance of adding LI as a covariate.

Ext. Data Fig. 3b: In the main text, it is not clear which specific comparison was performed to get these genes.

Ext. Data Fig. 4d: Are all genes in all samples shown? Is the correlation similar for just the VI gene signature? Is the correlation consistent for all samples?

Fig. 3a: It would be helpful to have the cell types order be consistent with Fig. 3b.

Fig. 3b: Are all the cell types listed in the legend present? For this visualization, it may be beneficial to collapse all the cell types with negligible proportions.

Fig. 3b: It would be useful to additionally show the cell type frequencies across the pathology annotations.

Fig. 3b: Are there other cell types in addition to peribronchial fibroblasts with significantly altered abundances between the two groups?

Fig. 3c: It would be interesting to see the stRNA-seq proportions in addition to the bulk RNA-seq ones.

Fig. 3d: If you check the enrichment of the VI cluster signatures across all the cell types (analogous to Ext. Data Fig. 5b), does that yield different associations?

Fig. 4a: The gene selection process includes dividing the genes into four clusters. Can you clarify the reasoning for the step?

Fig. 4b: Does increasing or decreasing the gene set size improve the performance of the predictor?

Fig. 5a: The figure legend states "VI+ tumors that did not contain VI foci in the capture area were considered as proximal VI+". Would the tumors without VI foci be considered distal?

(Remarks on code availability)

The URL provided in the Reviewer Assessment form (https://github.com/dst53/VI_manuscript) is different from the URL provided in the Code and Software Submission Checklist and the Reporting Summary (https://github.com/dst53/Steiner_NatComm). Neither one is accessible.

Version 1:

Reviewer comments:

Reviewer #1

(Remarks to the Author)

In response to my comments authors have performed a study on 24 selected FFPE samples and IHC and ISH for col1a1 on several slides from one sample. The following are my comments on these and other modifications:

1-clinical utility:

The addition of the FFPE biopsies to the end of the paper increases the clinical applicability of the paper. However, unfortunately it falls short of the main purpose of my critique which was to show the actual clinical utility of this approach. Testing the results of the bulk seq on 12 VI+ and 12VI- preselected samples does not mean that this approach will work on FFPE biopsies. This groups of samples do not have the variation seen in a regular sample pool and the AUROC derived from this set is of questionable applicability for clinical practice. Also, the placement of this section as a proof of concept defies the main purpose of this critique; the fact that the main utility of this finding, if validated, would be its diagnostic utility for FFPE samples.

-The addition to the TNM staging:

The statistical analyses showing that VI remains an independent predictor of RFS with TNM stage added is satisfactory. (fig1i extended) However,

-I cannot find the clinical characteristics of this 183 tumors that are “a subset” of previously published data (Yambayev et al). I can not find 1823 samples in that paper.

-Also: what percentage of the data is missing for each variable?

-About the age and pack year in these samples. Since these are continuous variables why are there no CIs?

-Extrinsic factors: Even though author's response about focusing on VI as an intrinsic factor (vs extrinsic factors such as clear margins and suboptimal sampling) is valid, the extrinsic features will still have importance in determining the predictive power of a biomarker in real world and should at least be mentioned in the discussion.

2-Clinical feasibility:

The authors in their response mention that in a real-world scenario the remainder of an FFPE after pathologist review will probably be used for bulk-seq, stressing the importance of the FFPE sample validation. Yet perhaps the weakest point of this manuscript at present is the analysis of the FFPE samples (24 preselected VI+ and VI- samples). Increasing the number of the FFPE samples and testing this approach on a true representative sample pool (rather than as an add-on proof of concept piece) will show the basic utility of this study.

The comment about the use of RNA-seq in current clinical practice: the sentence “whole transcriptome RNA-seq test...fine-needle aspiration FFPE material used ... is now standard of care for preoperative diagnosis...” should be moderated. The use of molecular probes for indeterminate thyroid nodules refers to a small group of samples assessed by FNA, and mRNA sequencing is only one of the three approaches used for these samples: mutational analysis, mRNA sequencing, and miRNA expression analysis. Authors are presenting Bulk-seq as a method for general classification of all early stage NSCLC samples, which is a very different premise.

Also, on the disparities between RNA-seq and qRT-PCR please refer to:

-Int. J. Mol. Sci. 2021, 22(17), 9349; <https://doi.org/10.3390/ijms22179349>

Despite the extensive discussion on the advantages of sequencing by some authors and commercial entities, I think we all know that the access to RNAseq is much more limited and technical caveats are much more prevalent, compared to hybridization probes or qPCR. These methods are currently used for biomarker analysis in many clinical settings and for various cancers, including breast and colon cancer. Therefore, validation by qPCR or RNA probes for some genes, which is routinely done for RNAseq studies is reasonable and adds to the possible clinical utility.

3-pathophysiological significance:

This is one of the main critiques regarding the appropriateness of this paper for Nature communications. Despite the extensive work and the multitude of statistical analyses, the conclusions in the manuscript are based on correlative data and deficient in mechanistic insights, which is a prerequisite for publication in journals like Nature Comm. The involvement of CAFs in vascular invasion is known. However, the main mechanisms mentioned for this contribution are increased angiogenesis, expression of ECM remodeling enzymes, immune response, and metabolic programming of tumor cells. (<https://doi.org/10.3389/fimmu.2025.1582532>)

Authors have picked Col1a1 for validation. This gene is expressed as part of a fibrotic tissue response to injury/invasion. Mechanistic questions remain. For instance: Does col1a1 contribute to the process of invasion? If so, how? Or is it just a consequence? Does it identify “activated” CAFs? Is there any mechanism downstream of col1a1 that can be shown to enhance any aspect of vascular invasion? In a mechanistic paper these questions need to be answered by experiments, not just correlation and citing literature.

-Critique on association with a specific cell type: staining for one of the genes.

Authors have performed IHC and ISH staining for col1a1, on multiple sections from one sample. Question arises, why could this not be done on more than one sample, as is regularly done for the histopathological validation of a gene involved in a pathological process? Staining even one or two sections from 10-20 samples for col1A1 and presenting the histopathological images that show the expression of this gene in myofibroblasts within the conventional “vascular invasion” sites would be appropriate for this response. The multiple statistical analyses show that within sample 4 the expression of col1A1 coincides with the expression of the other genes in cluster 2 in that sample. This kind of analysis adds to the rigor but does not provide an adequate validation and link between this new finding and the currently used histopathological studies that are the basis of the statistical analysis in extended figure 1i.

-Multivariable analysis on the derivation study: Extended Figure 7c is adequate and a critical piece of data. The same

questions about the CI for age and pack year apply and need to be answered. If the latter questions are adequately answered, I suggest moving this data to the main figures.

-On the distinction between LI and VI : the changes are adequate. I would add that LI was also an independent predictor in extended fig.1i and would show the gene signature associated with it as a variable in extended figure 7c.

-The explanations for tumor hypoxia and cell cycle genes are adequate.

-There are still a multitude of technical details in result sections that take away from the main message in each paragraph. The literature references are adequate.

Additionally:

The manuscript still has several stylistic and language issues and would benefit from cleaning up.

Some examples:

-What is the meaning of the sentence “ tumors were graded.... Or VI, is associated with outcome than the World...”

-The text still has redundant and hard to understand details that do not clarify the significance: For instance: “based on our analysis from consensus clustering the delta area under the cumulative....” This sentence and sentences alike should be modified/simplified or deleted.

-Where can we find the list of the genes for the four clusters described in the second results paragraph?

-In discussion paragraph 7:

“High VI predictor scores were associated not only with... VI- tumors were shedding cells into the circulation...” This sentence is confusing and needs clarification.

(Remarks on code availability)

Reviewer #2

(Remarks to the Author)

The authors have responded to the main comments that are listed below.

1. The determination of vascular invasion was by a single thoracic pathologist. Were these cases previously read by other clinical pathologists and what was the agreement of determination of vascular invasion in cases read by more than one pathologist? The authors should address inter-reader variability concerns with determination of this pathological finding.

The authors' response is complete.

2. The introduction emphasizes the potential clinical significance of detection of vascular invasion and suggests that the finding may support use of adjuvant therapy or lobectomy. This is confusing and should be clarified. Further, it is possible that the signature impact on recurrence free survival may be independent of performance of lobectomy or administration of adjuvant therapy. Addressing the potential clinical utility aspects of the proposed classifier and how it can be assessed will be helpful.

The discussion addresses these points. To further address potential clinical utility, the authors added analysis of a cohort of presurgical FFPE lung biopsy specimens acquired from 24 subjects in Figure 5f. The AUROC is shown. To better understand the performance of this proposed biomarker, it will be important to show the sensitivity, specificity, PPV and NPV of the predictor. In addition, it is important to carefully describe the specimens from which the predictor was evaluated. Were these consecutively acquired specimens or were they selected? Were the specimens derived from cell block or forceps biopsy? What was the evidence for vascular invasion on the biopsy specimens used for the analysis, if from forceps biopsy?

3. Figure 3 shows imputed cell type associations with the vascular invasion signature using spatial transcriptomic and bulk sequencing data. It will be important to validate key gene expression signatures in specific cell types using immunohistochemistry.

In extended figure 6, The authors show analysis of COL1A1IHC and ISH in one sample. This does not address validation of key V1 biomarkers cell specific expression in lung adenocarcinoma specimens with and without vascular invasion. These data will be important to demonstrate the generalizability of the reported observations.

(Remarks on code availability)

Reviewer #3

(Remarks to the Author)

The authors addressed my comments and introduced satisfactory corrections.

(Remarks on code availability)

Version 2:

Reviewer comments:

Reviewer #2

(Remarks to the Author)

The authors' responses are satisfactory.

(Remarks on code availability)

RESPONSE TO REVIEWERS' COMMENTS

Dear reviewers,

Thank you for your insightful comments. We have made many changes to the manuscript under your guidance and now report an important advance in our study. To demonstrate real-world feasibility and clarify the clinical application of our method, we profiled 36 new stage I LUAD FFPE pre-surgical biopsy specimens (24 biopsies, 12 matched resections) from an independent institution. Our VI predictor measured in biopsies achieved an AUROC of 0.90 for predicting VI at resection and showed strong concordance with matched resected tumors. These results underscore the potential of our predictor to help tailor pre- and peri-operative care and extend our biomarker's applicability beyond resected tumors. Collectively, this additional data brings our manuscript from the initial discovery of a robust gene expression signature linked to VI to the threshold of clinical implementation. In response to the reviewers' request for orthogonal validation of key findings from the spatial transcriptomics studies, we have now performed validation of a key VI cluster 2 gene, COL1A1, by RNAscope ISH and IHC on serial sections adjacent to a VI+ stRNA-seq sample. We find that COL1A1 signal (both RNA and protein) correlates tightly with our cluster 2 enrichment scores and localizes to the expected desmoplastic stroma and myofibroblastic CAFs. These data validate our previous observations from the stRNA-seq data and clarify the link between VI cluster 2, CAFs, and tissue remodeling, suggesting that there is a strong tumor microenvironment component to our VI gene expression signature. Finally, we addressed all other reviewer comments in detail. We clarified our clustering methodology and included new data to support our discovery of four main gene expression clusters linked to vascular invasion in response to Reviewer #3. We have refined our discussion in response to Reviewer #1 to better contextualize the roles of cell-cycle, hypoxia, EMT/angiogenesis, CAFs, and necrosis in vascular invasion. We have also included new multivariate survival analyses to show that our VI predictor is independent of TNM stage, surgical type, and adjuvant therapy. We also note a minor change to the original reported n of cases that passed QC in the validation cohort (now n=59, previously n=60). This was reported in error and has now been fixed, with no changes to the reported performance results. We provide all changes to the manuscript as highlighted text in the updated file and elaborate on each reviewer comment below.

REVIEWER COMMENTS

Reviewer #1 (Remarks to the Author): expertise in lung cancer TME

Steiner et al report a new set of genes that correlate specifically with vascular invasion in their cohort of patients with stage 1 LUAD. They have found four clusters of genes associated with vascular invasion. They have then explored some other characteristics of these genes, such as their site of expression within the tumors in relation to cellular content and their robustness in regards to intra-tumor heterogeneity, using stRNAseq on 15 samples. They have then used a second 60 patient cohort to validate their findings.

The main issues with this study are as follows:

1-The clinical utility of this set of genes as biomarkers for specific management of patients with stage 1 LUAD: As mentioned by the authors, the main management decision for this group of patients is the decision regarding lobectomy vs segmentectomy and two recent trials have failed to show inferiority of segmentectomy at this decision fork. This decision is made before surgical resection and possibly only based on a biopsy and imaging rather than sampling from the resected tumor. Therefore, a critical missing part from this study is validation of this gene signature in biopsies, rather than resected tumors.

We thank the reviewer for calling attention to the potential clinical applicability of our gene signature, and we agree that the ability to identify vascular invasive tumors prior to the decision of lobectomy vs segmentectomy is a major potential application of our gene signature. To demonstrate the potential utility of the VI gene signature in this setting, we have now validated the gene signature by RNA-sequencing of pre-surgical biopsies and matched resected tumors obtained from an independent cohort of stage I LUAD patients. In total, we performed RNA-seq on 36 new samples, which included 24 pre-surgical tumor biopsies and 12 from matched subsequently resected tumors. In this cohort, our VI predictor measured in presurgical tumor biopsies showed strong performance in predicting the detection of vascular invasion in the resected tumor (n=12 VI-, n=12 VI+) with an AUROC of 0.90 ($p=4.96 \times 10^{-4}$). Furthermore, we observed strong correlation in VI predictor score between the pre-surgical tumor biopsies and the matched resected tumors ($r=0.77$, $p=0.0053$). These samples were evaluated retrospectively from residual clinical samples from patients who received their care at Inova Schar Cancer Institute (extending our biomarker validation to another institution) and contain no overlap with our previously reported discovery or validation cohorts. They are also formalin-fixed paraffin embedded (FFPE) pre-surgical tumor biopsies, which is important for the clinical feasibility of our biomarker and ensures its compatibility with existing clinical workflows. After a pathologist has examined H&E slides of the pre-surgical biopsy, remaining FFPE biopsy material can be sent for molecular testing to generate a prediction of VI risk score, enabling timely pre-surgical decision making. Collectively, these results suggest that our VI predictor may have strong clinical utility in guiding pre-surgical management decisions for stage I LUAD patients. We have now updated the manuscript to reflect these important new results, including the figures (see Fig. 5f and 5g), tables (see Table S4), abstract, methods, results, and discussion sections.

The other clinically relevant point is if and how this gene expression biomarker adds to the universally used TNM staging, rather than the WHO grading of pathology.

We thank the reviewer for raising this relevant point. We agree that it is important to evaluate the gene expression biomarker in the context of TNM staging. While our validation cohorts are presently underpowered to stratify patient RFS by both TNM stage IA/IB + VI predictor, we have added additional analysis in the TRACERx cohort showing that our VI predictor is an independent predictor of RFS across TNM stage I and stage II LUADs. We have now included this result as a new panel in Extended Data Fig. 7 (previously Extended Data Fig. 6). We acknowledge the limitation of being underpowered to show that the VI

predictor is useful for identifying poor prognosis clinical stage IA or good prognosis clinical stage IB and have added this to the discussion. We are currently running an ongoing prospective clinical trial that will further explore the added benefit of VI predictor staging in larger cohorts. We have now added the following text to the results section of the manuscript: “Higher VI predictor scores were significantly associated with decreased 5-year RFS in TRACERx (Fig. 4f; HR=1.69, 95% CI 1.04–2.75, Cox regression p=0.036). The predictor was also significantly associated with 5-year RFS independently of TNM stage I and II (Extended Data Fig. 7g; HR=1.46, 95% CI 1.07-2.0, Cox regression p=0.018).”

For instance, one expects to find a statistical analysis showing that adding vascular invasion to TNM staging has had an impact on the patients’ survival and that this factor has shown an independent predictive value in a multivariable analysis, as shown in Ma et al, 2023.

We thank the reviewer for pointing out this omission. We have now included TNM stage as an additional covariate in our multivariate Cox regression analysis and updated Extended Data Fig. 1i and the results accordingly.

It is of note that even in the latter study, which is one of the main reference points for this paper, vascular invasion was only one of the factors that showed predictive value and other factors such as tumor size, clean margins, and suboptimal nodal sampling had similar or lower p values for predicting survival.

We thank the reviewer for raising this important point. We agree that factors such as tumor size, clean margins, and nodal sampling are all important determinants of prognosis. For decision making about pre-surgical management, we focus on intrinsic features of the tumor (such as size – which is already an integral component of TNM staging, and vascular invasion). In contrast, resection margins and the extent of nodal sampling reflect aspects of the surgical resection that we hope will be influenced by the preoperatively determined intrinsic features of the tumor.

2-Clinical feasibility in a “real world” setting: The proposed 48-gene set is derived from bulk-seq in 100 samples from the firsts cohort and then validated (presumably) using bulk-seq data on 60 tumor samples from the second cohort. Regarding the limitations of bulk seq performance and analysis in clinical settings, how would these 48 gene perform, if measured by more widely available gene probes/qPCR?

We thank the reviewer for this question about clinical feasibility. Our new results show validation of our 48-gene set VI predictor in FFPE pre-surgical biopsy material by RNA-seq. We anticipate that after a pathologist reviews H&E slides of the pre-surgical biopsy, the remaining FFPE material can be used directly for RNA extraction and sequencing, enabling compatibility with existing clinical workflows. To that end, we are not aware of any limitations of bulk RNA-seq performance and analysis in clinical settings. In fact, RNA-seq has proven clinical use even on small FFPE cytology specimens. For example, Veracyte’s

Afirma Genomic Sequencing Classifier is a CLIA-certified, whole transcriptome RNA-seq test that is performed on the same fine-needle aspiration FFPE material used for cytology review and is now standard of care for preoperative diagnosis of indeterminate thyroid nodules in thousands of patients (PMID: 29799911). This demonstrates that RNA-seq on FFPE clinical specimens can integrate into routine practice. While we cannot answer how our 48-gene predictor would perform when measured by qPCR, large multicenter efforts (SEQC/MAQC-III) have shown that RNA-seq expression measurements correlate tightly with TaqMan qPCR (Pearson $r > 0.9$ for most genes) (PMID: 25150838). Furthermore, RNA-seq has additional advantages over qPCR that are desirable for clinical implementation of our VI predictor. RNA-seq based normalization is more robust and means that predictor performance will be less sensitive to variation in any single reference gene used by qPCR. RNA-seq also has built-in sample QC metrics, such as mapping rates, and the ability to perform orthogonal checks on sample identity. We are nevertheless eager to determine the cost effectiveness of measuring the VI predictor using other laboratory methods in future studies.

3-Pathophysiological significance: Even though the authors go through extensive description of gene expression patterns based on stRNA-seq, the paper does not contain simple validations such as staining for a group of the high-ranking genes in clinical samples to place them within known cell types and possibly imply a function. The best evidence supporting a biomarker is biological/functional validity that will come out of mechanistic experiments, and this kind of evidence is an implied prerequisite for a high impact publication.

We thank the reviewer for raising this important point. We have addressed the request for validation via staining in the reviewer's related comment below about validation by staining.

The following is a critique of the specific sections:

The main finding in the first part of the study is that among the three pathological patterns defined by the authors, vascular invasion has the highest hazard ratio for RFS. While this analysis is valid internally and a multi-variable analysis shows its independent predictive value, the TNM stage of the tumors is missing from this table.

We thank the reviewer for pointing out this omission. We have now added TNM stage 8th edition to both Table S1 and Table S3 and included TNM stage as a covariate in the multivariable analysis presented in Extended Data Fig. 1i, as described above.

The study on the correlation with the histopathological feature is entirely based on stRNA-seq on 15 tumor samples that are mixed stage 1 and 2. Authors have tried to study various issues such as intratumor heterogeneity and structural correlates based on these samples. While interesting, the value and general applicability of statistical reasoning based on findings within these samples are questionable.

We thank the reviewer for raising the concern about sample size in our stRNA-seq analysis. Although our initial dataset comprised 15 sections (8 VI-, 7 VI+), we used a linear mixed-effects model with sample ID as a random effect to control for inter-sample variability, yielding robust associations (e.g. Cluster 2 with desmoplastic stroma – a finding that we have since validated by IHC and ISH – see response below). To further demonstrate that these findings are not cohort-specific, we validated our results using 11 additional Visium sections (6 VI-, 5 VI+; no overlap with our discovery or validation sets). In these new samples, we similarly found that VI gene cluster 2 is strongly associated with desmoplastic stroma ($p_{\text{adj}}=1.14 \times 10^{-63}$). In addition, as before, we observed strong expression of Cluster 1 and Cluster 3 genes directly in VI foci ($p = 9.70 \times 10^{-7}$ and 2.63×10^{-56} respectively). We also validated that cluster 1 and cluster 3 have increased expression in regions annotated as high-grade patterns: solid ($p_{\text{adj}}= 6.84 \times 10^{-7}$; 2.26×10^{-68}), micropapillary ($p_{\text{adj}}=1.32 \times 10^{-3}$; 3.94×10^{-101}), and cribriform ($p_{\text{adj}}=1.56 \times 10^{-48}$; 9.50×10^{-100}). Finally, expression of cluster 4 was strongly decreased in VI foci ($p_{\text{adj}}=6.60 \times 10^{-38}$) and showed positive enrichment in lepidic pattern ($p_{\text{adj}}=8.68 \times 10^{-8}$), normal bronchus ($p_{\text{adj}}=1.10 \times 10^{-5}$) and adjacent normal-appearing lung ($p_{\text{adj}}=1.57 \times 10^{-3}$). Validating these results in a second independent stRNA-seq cohort suggests that our statistical approach studying the association of VI cluster expression by pathology annotation is valid and yields highly reproducible findings. Because the new data used for validation is part of separate ongoing study and cannot yet be made publicly available, we have opted not to include these validation results in the present manuscript. We include them in the figure below in our response to the reviewer only.

Reviewer Response Figure 1. Reproducing our stRNA-seq histopathology findings in an additional cohort a. Pathology annotations present across new spatial transcriptomics (stRNA-seq) capture areas that passed QC (n=11). **b.** Association between expression of VI-associated clusters and pathology across all 11 samples. Only results with Bonferroni

adjusted p values ($p_{\text{adj}} < 0.01$) are shown. P_{adj} values were derived from a generalized binomial linear mixed effect model of VI cluster enrichment score as a function of spot pathology annotation with sample as a random effect. For visualization purposes the plot shows the mean of per-spot enrichment scores within a given region annotated as the same pathology.

The same critique applies to the findings on association of gene clusters with specific cell types. As an acceptable validation step for these correlative hypotheses, one would expect a simple staining showing a specific gene with a known function within a specific cell type, possibly together with a mechanistic experiment that captures at least one aspect of that gene function. In the absence of such data, statements such as “ We found a trend toward a higher proportion of peribronchial fibroblasts belonging to VI+ samples (Fig. 3b), consistent with the tissue remodeling biology of VI cluster 2 described above. VI+ samples also had a trend toward a higher proportion of plasma cells and B cells.” remain enigmatic and of questionable value.

We thank the reviewer for providing this valuable feedback around validating our cell type association findings. We have now generated and included the requested data and added additional analyses. Because our main finding about cell types (Figure 3) was the association between VI gene cluster 2 and peribronchial fibroblasts (or myofibroblasts), desmoplastic stroma, and VI+ tumors, we have now performed in situ hybridization (ISH) and immunohistochemistry (IHC) for COL1A1, a key Cluster 2 gene. Below is a summary of the new data and how it clarifies the link between VI cluster 2, peribronchial fibroblasts, and tissue remodeling.

First, we previously demonstrated that the peribronchial fibroblasts from the Salcher LUAD atlas we used for our stRNAseq deconvolution analysis are referred to as myofibroblasts by the Hanley et al 2023 LUAD fibroblast scRNAseq dataset. This is important because we used the Salcher et al data for deconvolution of our stRNAseq data. The original submission of the manuscript demonstrates that these differently named cell populations are very likely equivalent. As part of the revised manuscript, we have further scored the Hanley et al data (an independent dataset consisting of 46 independent scRNAseq samples) for expression of the genes in VI gene cluster 2. We observed significantly higher expression of VI gene cluster 2 in Hanley’s myofibroblast cluster compared with alveolar and adventitial clusters. Hanley et al reported that the key markers of LUAD myofibroblasts are *MMP11*, *POSTN*, *CTHRC1*, *COL1A1*, and *COL3A1*, which are all VI cluster 2 genes.

Next, to validate our stRNA-seq deconvolution results directly, we performed *COL1A1* RNAscope ISH and *COL1A1* protein IHC labeling on serial sections close to stRNA-seq sample 4, a VI+ tumor with prominent desmoplastic stroma. We selected sample 4 due to the difficulty in obtaining serial sections close enough to the original stRNA-seq section in the other samples in our cohort. After computationally registering the adjacent sections to the original Visium image and creating a ‘virtual’ Visium spot grid on the adjacent sections, we observed a significant correlation across spots between the fraction of ISH RNAscope

COL1A1+ expressing cells per spot (quantified by QuPath) and the average VI cluster 2 gene expression per spot (from spatial transcriptomics) ($r=0.53$, $p < 2.2 \times 10^{-16}$). We also observed a similar significant correlation across spots between average VI cluster 2 gene expression and *COL1A1*+ protein expressing cell fraction measured by IHC ($r=0.47$, $p < 2.2 \times 10^{-16}$). We also did IHC staining for another VI cluster 2 gene, *THY-1*, that was not reported by Hanley et al as a key myofibroblast marker. The fraction of *THY-1*+ protein expressing cells measured by IHC was significantly correlated with the fraction of *COL1A1*+ protein expressing cells measured by IHC ($R=0.52$, $p < 2.2 \times 10^{-16}$), which further supports the coexpression of VI gene cluster 2 at the protein level. When we examined the correlation between *COL1A1*+ expressing cells measured by both IHC or ISH, and the proportion of myofibroblasts predicted by our stRNA-seq deconvolution algorithm, we found that both *COL1A1* RNA and protein were significantly more correlated with the fraction of peribronchial (ISH $r=0.43$, $p=7.95 \times 10^{-159}$; IHC $r = 0.36$, $p = 1.66 \times 10^{-107}$) than adventitial (ISH $r=-0.10$, $p=4.1 \times 10^{-9}$; IHC $r=0.11$, $p=2.47 \times 10^{-10}$) or alveolar fibroblasts (ISH $r=0.14$, $p=1.05 \times 10^{-17}$; IHC $r=-0.05$, $p=0.001$) (Steiger's p value for difference in *COL1A1* correlation between peribronchial vs adventitial = 7.85×10^{-17} for IHC and 4.18×10^{-78} for ISH; Steiger's p value for difference in *COL1A1* correlation between peribronchial and alveolar = 3.12×10^{-24} for IHC and 5.61×10^{-101} for ISH). This further demonstrates that cells expressing a key marker of our VI gene cluster 2 show a myofibroblast-like phenotype. Finally, we also observed a significant association between the fraction of *COL1A1*+ protein expressing cells measured by IHC and spots annotated by the pathologist as belonging to desmoplastic stroma ($p = 4.17 \times 10^{-16}$), further suggesting that VI cluster 2 is associated with desmoplastic stroma and tissue remodeling. We have now updated the figures (see Fig. 3 and Extended Data Fig. 6) results section, methods section detailing these additional findings and our analysis approach.

We have added the following text to the results section:

To further validate our stRNA-seq deconvolution findings about the relationship between VI cluster 2, desmoplastic stroma, and peribronchial fibroblasts, we performed *COL1A1* RNAscope ISH and *COL1A1* IHC labeling on serial sections close to stRNA-seq sample 4, a VI+ tumor with prominent desmoplastic stroma. We selected *COL1A1* as it is a representative gene from VI cluster 2 that also showed significantly higher expression in myofibroblasts compared with adventitial or alveolar fibroblasts in Hanley et al (Extended Data Fig. 6a) After serial image registration to the H&E from the section used for stRNA-seq, we observed a significant correlation across spots between the fraction of ISH RNAscope *COL1A1*+ expressing cells per spot and the average VI cluster 2 gene expression per spot ($r=0.53$, $p < 2.2 \times 10^{-16}$; Extended Data Fig. 6b). We also observed a similar significant correlation between the average VI cluster 2 gene expression per spot and the fraction of *COL1A1*+ cells per spot labeled by IHC ($r=0.47$, $p < 2.2 \times 10^{-16}$; Extended Data Fig. 6c). We also performed IHC for *THY-1*, a VI cluster 2 gene that was not reported by Hanley et al as a key myofibroblast marker. The fraction of *THY-1*+ expressing cells per spot by IHC was significantly correlated with the fraction of *COL1A1*+ expressing cells measured by IHC, which further supports the coexpression of VI gene cluster 2 at the protein level

($R=0.52$, $p < 2.2 \times 10^{-16}$; Extended Data Fig. 6d). When we examined the correlation between COL1A1+ cell fractions measured by either IHC or ISH with the fraction of myofibroblasts predicted by the CytoSPACE stRNA-seq deconvolution, we found that both were significantly more correlated with the fraction of peribronchial/myofibroblastic CAFs than either adventitial or alveolar fibroblasts (Extended Data Fig. 6e). This further demonstrates that cells expressing COL1A1, a key marker of VI cluster 2, show a myofibroblast-like phenotype. Finally, we also observed a significant association between the fraction of COL1A1+ IHC fraction and spots annotated as desmoplastic stroma (Extended Data Fig. 6f), further suggesting that VI cluster 2 is associated with tissue remodeling.

We have added the following text to the methods section:

In situ hybridization and immunohistochemistry. Chromogenic immunohistochemistry and RNA in situ hybridization were performed on a Ventana Discovery ULTRA (Roche Diagnostics). Immunohistochemical staining for Thy1/CD90 was performed with a rabbit monoclonal antibody (D3V8A, #13801, Cell Signaling) at 1:100 dilution after heat-induced epitope retrieval in alkaline buffer. IHC for COL1A1 was performed with a rabbit monoclonal antibody (E8F4L, #72026, Cell Signaling) at 1:200 dilution after heat-induced epitope retrieval in alkaline buffer. RNA ISH for COL1A1 was performed using RNAscope using 2.5 VS probe-HS-COL1A1 (#401899, Advanced Cell Diagnostics) using both protease and heat-induced epitope retrieval as per the manufacturer's guidelines.

Spatial alignment of ISH/IHC data to stRNA-seq data. Individual ISH or IHC images were registered separately to the matching adjacent original stRNA-seq H&E image via affine transformation with annotated landmarks using the STalign87 python tool. The same affine transformation was used to align the stRNA-seq spot coordinate grid to the ISH or IHC image. Affine-aligned and filtered stRNA-seq spots were then imported into QuPath v0.5.188. Only spots that were present over tissue in both images and passed QC in the stRNA-seq pipeline were included. QuPath positive cell detection was used to quantify the number of ISH and IHC positive cells at defined staining thresholds to maximize signal to noise ratio. For correlation analyses between ISH/IHC and stRNA-seq, we calculated the mean IHC or ISH positive cell fraction, smoothed by k-nearest neighbors, with $k=7$ (representing a single spot and its immediate neighbor spots). Because the images were not exact serial sections this approach was applied to smooth minor differences in spot alignment.

In the derivation study: was a multivariable analysis, similar to the one performed for VI, done for the 48-gene set? Wouldn't such analysis be required as the main message of this paper, ie showing if altered expression (levels defined for each gene) of a 48 gene set has an independent predictive value for tumor recurrence or survival.

We thank the reviewer for pointing out this omission. We have now included a multivariate survival analysis with the VI predictor in the discovery cohort as an additional figure panel in Extended Data Fig. 7 (previously Extended Data Fig. 6). We show here that the VI

predictor is independently predictive of RFS when including common clinicopathology variables, including TNM stage. We have added the following text to the results section of the manuscript: “The predictor was independently associated with 7-year recurrence free survival (HR= 3.31, 95% CI 1.38 – 7.96, Cox regression $p=1.74 \times 10^{-3}$) when including common clinicopathology variables, including TNM stage (Extended Data Fig. 7c).”

The point about the distinction between vascular and lymphatic invasion is interesting and gene expression patterns are hinting at possible biological differences. However, to the reviewer, none of these findings prove or disprove whether lymphatic invasion is a stage in the continuum, or a biologically different and unique process with clinical implications.

We thank the reviewer for this question about whether LI and VI represent a continuum or distinct process in LUAD. We have further analyzed our data and clarified in the manuscript that, although LI and VI are often combined as lymphovascular invasion (LVI) in clinical reports, our findings indicate that LI and VI exhibit distinct gene expression profiles and biological characteristics. For example, genes involved in epithelial–mesenchymal transition (EMT) and tissue remodeling (Cluster 2) were strongly upregulated in VI+ tumors but not in LI+ tumors, indicating a VI-specific invasion phenotype. Consistently, the VI predictor developed in our study did not identify LI+ cases in the absence of VI (AUC=0.51) – supporting that it predicts VI specifically rather than a general LVI state. Finally, the transcriptional signature associated with VI was far more robust than that of LI. After adjusting for LI status in our model, we identified 133 genes significantly associated with VI (FDR < 0.01), whereas only 6 genes met this threshold for LI. Taken together, these observations suggest that while LI and VI often co-occur, they are associated with distinct gene expression patterns and tumor biology in our cohort.

However, we do acknowledge the limitations in conclusively determining whether LI and VI lie along a biological continuum or represent entirely separate pathways. Thus, while our evidence supports a biological distinction (with VI conferring unique molecular and clinical features), we cannot exclude the possibility that LI and VI are related steps in tumor progression. We have revised our conclusions in the discussion to reflect this uncertainty. Future investigations, such as spatial transcriptomic analysis focused on lymphatic invasion sites or longitudinal studies (e.g., tracking tumors over time) could determine whether tumors progress from LI to VI or if they arise via distinct routes. Such studies will be needed to establish whether LI and VI are part of a continuum of metastatic spread or are fundamentally distinct biological processes.

In the validation set, the 48 gene signature is applied to 60 samples, only 17 of which had VI. Would a power calculation for this analysis be appropriate?

In the validation set the 48 gene signature achieved an AUROC of 0.86 with a 95% confidence interval of 0.76–0.96 and a p value of 6.16×10^{-6} . This robust performance demonstrates that, even with only 17 VI+ samples, the probability that the true AUROC is above 0.50 is effectively 100%. Given these results, a power calculation would be of

limited additional value. While we acknowledge that a larger sample size would narrow the uncertainty around the effect size, our current analysis is adequately powered to support the conclusion that the 48-gene signature is predictive of VI status. Future studies with larger cohorts can help to refine these estimates further. For additional clarity, we have now added the 95% confidence interval and p values in the text of the results section when reporting AUROC values.

I cannot find the data for the sentence “It was associated with histologic grade and 7-year recurrence free survival (HR=1.98, 95% CI 1.09–3.60, Cox regression p=0.024) (Extended Data Fig. 6b and Fig. 4f).”

We apologize for the confusion. We have now separated the figure references for these statements for clarity. The text now reads: “It was associated with histologic grade (Extended Data Fig. 7d) and 7-year recurrence free survival (HR=1.98, 95% CI 1.09–3.60, Cox regression p=0.024) (Fig. 4f).”

In discussion, some of the pathophysiological explanations are not convincing or need more explanation. For instance, why would a group of “cell cycle” or “tumor hypoxia” genes specifically contribute to vascular invasion. Why does cluster 2 (EMT/Angiogenesis) correlate with the presence of CAFs? What is the value of necrosis? How would a gene signature of vascular invasion have prognostic value in VI- tumors?

We thank the reviewer for these questions. We have now substantially revised the discussion section to clarify the biological interpretation our findings, drawing on evidence from our data and the literature.

On stylistic issues, the results section is filled with technical details, and this overrepresentation of details detracts from the main message of the paper and its proposed value as a biomarker of invasiveness.

We thank the reviewer for providing stylistic feedback on the results section. We have now made substantial changes to the results section, which includes reporting the results of many new analyses that were done to address reviewer comments. During this process, we were careful to avoid including unnecessary technical details and have moved relevant details about existing and new results to the methods section.

Also, some of the premises of the study are not solid. For instance, authors mention that in the Hattori et al study (2024) “...post-hoc analyses are beginning to show subsets of patients for which lobectomy might decrease the rate of recurrence.” The mentioned study showed that lobar resection was associated with higher locoregional recurrence among the full cohort, but survival was age and sex-dependent.

We appreciate the reviewer pointing out our lack of clarity here. We cited the Hattori (2024) study to highlight that post-hoc analysis are just beginning to be published on the

more recent randomized control trials but recognize this may be confusing as stated. We have simplified the language of paragraph 2 of the introduction to make this more clear and further improved clarity in response to reviewer #2's feedback. We have modified the following text in Paragraph 2 of the introduction:

Original text:

Patients with stage I VI+ LUAD may benefit from adjuvant therapy⁹⁻¹², but it is difficult to assess in resected tumor specimens. Elastic stains can be used to improve visualization of invaded blood vessels over hematoxylin and eosin (H&E), but comprehensive tumor histopathology review for small (<1mm) VI foci is difficult and prone to false negatives. Similarly, biopsy specimens do not provide enough tissue material to identify VI+ LUAD prior to surgery, preventing the use of VI status to guide surgical or neoadjuvant treatment approaches. Recent evidence from large clinical trials including JCOG0802/WJOG4607L and CALGB140503 suggests that lobectomy provides no clinical benefit over sublobar resection for patients with stage IA disease^{13,14}, although post-hoc analyses are beginning to show subsets of patients for which lobectomy might decrease the rate of recurrence¹⁵. Moreover, retrospective analysis shows that VI+ patients receiving sublobar resection have increased rates of recurrence, highlighting an emerging need to identify if these patients will benefit from precision surgical approaches¹⁶.

Revised text:

Patients with stage I VI+ LUAD may benefit from adjuvant therapy⁹⁻¹², but it is difficult to assess in resected tumor specimens. Elastic stains can be used to improve visualization of invaded blood vessels over hematoxylin and eosin (H&E), but comprehensive tumor histopathology review for small (<1mm) VI foci is difficult and prone to false negatives. A biomarker of VI would improve existing routine histopathologic review of high-risk cases at resection. Moreover, biopsy specimens do not provide enough tissue material for pathologists to identify VI+ LUAD prior to surgery, preventing the use of VI status to guide surgical or neoadjuvant treatment approaches. Retrospective analysis shows that VI+ patients receiving sublobar resection have increased rates of recurrence¹³. While recent evidence from large clinical trials including JCOG0802/WJOG4607L and CALGB140503 suggests that lobectomy provides no clinical benefit over sublobar resection for patients with stage IA disease^{14,15}, post-hoc analyses of pathologic factors have yet to be published¹⁶. Therefore, there is an emerging need to identify if stage I patients will benefit from precision surgery. In parallel, biomarker approaches that are informed by the tumor microenvironment could also improve patient selection for neoadjuvant or periadjuvant therapy in early-stage LUAD^{17,18}.

Authors mention that gene signature for vascular invasion has not been studied. A number of studies in the literature including <https://doi.org/10.1016/j.ajpath.2010.10.040>, <https://doi.org/10.1371/journal.po>

[ne.0216847](https://doi.org/10.1038/s41416-019-0486-6), and <https://doi.org/10.1038/s41416-019-0486-6>, had visited this entity in cancer.

We apologize for the lack of clarity. We have re-written the introduction text to more clearly state that “To our knowledge, VI-associated gene expression has not been studied in lung cancer.” To further highlight the novelty and robustness of our gene expression signature of VI, we have compared our signature with two of the three papers the reviewer referenced that examined RNA differences (the third investigated miRNA differences). The first paper was from Mannelqvist et al. 2011 (PMID: 21281818), who compared VI+ and VI- primary endometrial cancers by DNA microarray and identified a 35-gene signature VI signature (26 upregulated, 9 downregulated). The second paper came from Kurozumi et al. 2019 (PMID: 31114020), who derived a 99-gene signature (42 upregulated, 57 downregulated) between LVI+ and LVI- breast cancers by RNA-seq. We found limited overlap between the upregulated genes in these signatures and the upregulated genes in our signature. But when we performed gene-set enrichment analysis (GSEA) on each signature’s upregulated genes against a ranked list of genes ordered by association with VI (18,285 genes measured), we found that both the 26-gene Mannelqvist 2011 signature upregulated with VI+ endometrial cancer and the 42-gene Kurozumi 2019 signature upregulated with LVI+ breast cancer were significantly enriched. This suggests that some of the molecular aspects of VI may be shared among different epithelial cancers and that this is an area that should be further explored. We have added the following text to the results section of the manuscript titled ‘Gene expression changes in stage I LUAD with VI’: “Finally, genes previously found to be increased in VI+ endometrial cancer³⁰ or LVI+ breast cancer³¹ were each significantly enriched among genes most increased in VI+ stage I LUAD tumors using GSEA (Extended Data Fig. 3g-h). This hints at common molecular changes associated with VI across epithelial tumors that should be explored in future work.”

Reviewer #2 (Remarks to the Author): clinical expertise in lung cancer development and progression

This manuscript reports a lung adenocarcinoma transcriptional signature that is associated with the pathological detection of vascular invasion in resected stage 1 tumors. The association was confirmed in an independent cohort of 60 patients with AUROC of 0.86. The signature was associated with recurrence free survival and was detectable in regions without invasion in tumors that were annotated as harboring vascular invasion. The study design is sound and the observations are robust, although not conceptually novel.

1. The determination of vascular invasion was by a single thoracic pathologist. Were these cases previously read by other clinical pathologists and what was the agreement of determination of vascular invasion in cases read by more than one pathologist? The authors should address inter-reader variability concerns with determination of this pathological finding.

We thank the reviewer for raising the important point of inter-reader robustness for VI among pathologists. We previously published on inter-reader reproducibility for VI (PMID 38204657 and reported substantial agreement ($\kappa = 0.71$) among 5 pathologists for the diagnosis of vascular invasion. The cases that were used for assessing inter-reader reproducibility for VI included histopathology samples that were profiled as part of our discovery cohort. We have now added the following statement to the methods to reference this prior inter-reader variability assessment: “An inter-reader reproducibility assessment was previously performed on these pathology cases and found substantial agreement ($\kappa = 0.71$) among 5 pathologists for the diagnosis of VI (PMID 38204657).”

2. The introduction emphasizes the potential clinical significance of detection of vascular invasion and suggests that the finding may support use of adjuvant therapy or lobectomy. This is confusing and should be clarified.

We thank the reviewer for suggesting clarification of the clinical significance of VI detection. We have re-written paragraph #2 of the introduction for clarity to distinguish between two use cases for a VI biomarker: 1) predicting VI using resection tissue to guide adjuvant therapy and 2) predicting VI pre-surgically using biopsies to guide precision surgery or neoadjuvant therapy. To support the use of the VI biomarker in presurgical management decisions, and as detailed in our response to Reviewer #1 (see above), we have now validated the gene signature by RNA-sequencing of pre-surgical biopsies and matched resected tumors obtained from an independent cohort of stage I LUAD patients. We have modified the following text in paragraph 2 of the introduction of the manuscript:

Patients with stage I VI+ LUAD may benefit from adjuvant therapy⁹⁻¹², but it is difficult to assess in resected tumor specimens. Elastic stains can be used to improve visualization of invaded blood vessels over hematoxylin and eosin (H&E), but comprehensive tumor histopathology review for small (<1mm) VI foci is difficult and prone to false negatives. A biomarker of VI would improve existing routine histopathologic review of high-risk cases at resection. Moreover, biopsy specimens do not provide enough tissue material for pathologists to identify VI+ LUAD prior to surgery, preventing the use of VI status to guide surgical or neoadjuvant treatment approaches. Retrospective analysis shows that VI+ patients receiving sublobar resection have increased rates of recurrence¹³. While recent evidence from large clinical trials including JCOG0802/WJOG4607L and CALGB140503 suggests that lobectomy provides no clinical benefit over sublobar resection for patients with stage IA disease^{14,15}, post-hoc analyses of pathologic factors specifically associated with recurrence in patients who underwent sublobar resection have yet to be published¹⁶. Therefore, there is an emerging need to identify if stage I patients will benefit from precision surgery. In parallel, biomarker approaches could also improve patient selection for neoadjuvant or adjuvant therapy in early-stage LUAD^{17,18}.

Further, it is possible that the signature impact on recurrence free survival may be independent of performance of lobectomy or administration of adjuvant therapy.

Addressing the potential clinical utility aspects of the proposed classifier and how it can be assessed will be helpful.

We thank the reviewer for this insightful comment. Yes, in the TRACERx LUAD dataset we find that when we add surgery type (lobectomy or sublobar resection) and administration of adjuvant therapy as covariates in our Cox regression model, the VI predictor scores remain significantly associated with decreased 5-year RFS. We have now included this result in Extended Data Fig. 7h and reported this in the results section. Our hypothesis is that any difference in RFS between VI+ and VI- patients in the group receiving sublobar resection would be attenuated in the group receiving lobar resection. These groups are not large enough in TRACERx to conduct this analysis, but we are planning a well-powered clinical study to address this question in the future.

It is important to note at the time of dataset collection adjuvant therapy was not routinely given to stage I LUAD patients, and only 4/82 patients in this analysis received adjuvant therapy. Furthermore, adjuvant therapy was not administered to any patients in our discovery or validation cohorts. We have now clarified this by changing the following text in our methods section from “A discovery cohort consisting of 192 resected tumors from 192 patients with 8th edition TNM stage 0/1 LUAD and not treated with neoadjuvant therapy were included in this study” to “A discovery cohort consisting of 192 resected tumors from 192 patients with 8th edition TNM stage 0/1 LUAD and not treated with neoadjuvant or adjuvant therapy were included in this study.” We also changed the text, “An independent validation cohort consisting of 61 resected tumors from 60 patients with 8th edition TNM stage I/II LUAD and not treated with neoadjuvant therapy were included in this study” to “An independent validation cohort consisting of 61 resected tumors from 60 patients with 8th edition TNM stage I/II LUAD and not treated with neoadjuvant or adjuvant therapy were included in this study.” We have added the following text to the results section of the manuscript: “Additionally, VI predictor scores remained significantly associated with decreased 5-year RFS when including surgery type (lobectomy or sublobar resection) and adjuvant therapy as covariates (HR = 1.66, 95% CI 1.02-2.71, multivariate Cox regression $p=0.041$; Extended Data Fig. 7h).”

3. Figure 3 shows imputed cell type associations with the vascular invasion signature using spatial transcriptomic and bulk sequencing data. It will be important to validate key gene expression signatures in specific cell types using immunohistochemistry.

We thank the reviewer for raising the important point about validation via IHC. We have performed the requested analysis as described above in response to Reviewer #1’s similar comment.

Reviewer #3 (Remarks to the Author): expertise in lung ST and scRNA-seq

In their manuscript “Identification of a gene expression signature of vascular invasion and recurrence in stage I lung adenocarcinoma via bulk and spatial transcriptomics”, Steiner et al. developed a transcriptomic signature of microscopic vascular invasion (VI) in stage I lung adenocarcinoma based on a cohort of 163 tumors. The VI-associated expression patterns clustered into four subclasses linked to unique biological processes. The authors utilized spatial transcriptomics to describe the spatial organization of the four gene clusters. The authors additionally developed a predictor of VI in an independent cohort and evaluated its performance in multiple regions of the same tumor to show the predictor is largely unaffected by intra-tumor heterogeneity. The novel transcriptional signature for vascular invasion in early-stage lung adenocarcinoma presented in this work may enable more precise diagnosis and treatment of patients.

My specific comments which may help to improve the manuscript are (as line numbers are not available, I am referencing the figure panels to identify the relevant section of the manuscript):

Ext. Data Fig. 1a-f: It would be helpful to add outcomes that include both VI and LI classification given that LI is used as a covariate for some of the differential expression analysis.

We thank the reviewer for this helpful feedback. We have now added an additional panel to Extended Data Fig. 1 that shows patients stratified by lymphovascular invasion (VI+ and/or LI+) in the same Kaplan-Meier plot and updated the corresponding figure reference (Extended Data Fig. 1g) in the results section.

Ext. Data Fig. 1j: In the legend, “only” is cut-off.

We thank the reviewer for pointing this out. We have now fixed the plot legend for the original Extended Data Fig. 1j (now Fig. 1k).

Ext. Data Fig. 1j-k: Is the site of recurrence related to time to recurrence?

This is an interesting question. In our discovery cohort there is no clear association between the site of recurrence and the time to recurrence. When we plot the cumulative incidence functions, we observe substantial overlap in the 95% confidence intervals for loco-regional only and distant recurrences, indicating that any differences in timing are minimal. We have now included this plot as an additional panel in Extended Data Fig. 1 and updated the figure legend and reference to the figure (Extended Data Fig. 1m) in the results section accordingly.

Fig. 1b: It may be helpful to have VI/NST/LMP as a single column annotation as they are mutually exclusive and based on the same grading system.

We thank the reviewer for this suggestion. We have updated the VI/NST/LMP column annotation in the Fig. 1b and Extended Data Fig. 3a-b heatmaps to better reflect their mutual exclusivity.

Fig. 1b: The differential expression results and cluster assignments should be provided as supplementary tables or in a data repository.

Thank you for the helpful suggestion. We have now included this information in the source data file.

Fig. 1b: There are two VI samples that are Ving with LMP samples. Are there any biological or technical factors that may explain this grouping?

This is an astute observation. Yes, we have provided additional details to the results section of the manuscript about these two samples: “We observed that two VI⁺ tumors clustered with the LMP classified tumors. These both represented rare edge cases, with one being the only lepidic predominant VI⁺ sample and the other being the only VI⁺ sample with >50% micropapillary growth pattern. Larger cohorts containing more of these types of samples will be required to determine if the current classifier might be less accurate in detecting VI in these conditions.”

Fig. 1b: The NST samples are clustering with both VI and LST samples. Can you comment on how to best interpret the two VI classifications?

We apologize for the confusion. The “No Special Type” (NST) histopathology grade refers to tumors that do not contain the required characteristics to be considered low malignant potential (LMP) and do not contain vascular invasion. NST is thus a ‘catch all’ for ‘intermediate’ aggressiveness. When we derive our 48-gene VI predictor, we train it via supervised learning to discriminate VI tumors from both NST and LMP tumors.

Fig. 1b: Cluster 1 and, to a lesser degree, cluster 3 appear to be separating the VI⁺ samples into two subsets. Is there any merit to treating those as independent groups for downstream analysis?

We thank the reviewer for this suggestion. We had performed additional analysis to support the observation that there may be two subsets of VI⁺ tumors present in the discovery cohort. However, when we divided the VI⁺ samples into the two subsets referenced by the reviewer, we did not find a significant difference in RFS between them (log-rank p value = 0.16). The primary differences between these two VI⁺ samples were in histopathologic pattern. The VI⁺ tumors with higher co-expression of cluster 1 genes had higher mean % solid (39 vs. 8) and mean % cribriform (18 vs. 7). These findings further highlight the importance of building our VI predictor to be independent of the LUAD high-grade patterns, which we demonstrated in our stRNA-seq data.

Ext. Data Fig. 3a: It may be helpful to add a brief explanation of the biological significance of adding LI as a covariate.

We thank the reviewer for this suggestion. Adding LI as a covariate helps isolate gene expression changes specific to VI by accounting for any overlap between VI and LI. We have now added the following text to the results section of the manuscript (that also addresses the reviewer's point below). Previously the text stated, "When we added LI as a covariate to our model, we identified 133 genes associated with VI (FDR < 0.01), which included 130 genes from the original 474-gene VI signature (Extended Data Fig. 3a)." It now reads: "We added LI as a covariate to our model to determine whether the gene expression differences seen with VI are due to the covariance between LI and VI. We identified 133 genes associated with VI (VI+ vs. LMP, FDR < 0.01), which included 130 genes from the original 474-gene VI signature (Extended Data Fig. 3a). At the same significance threshold (FDR < 0.01), we recovered only 6 genes associated with LI (LI+ vs. LI-) using the same model, with 2/6 belonging to the VI signature (Extended Data Fig. 3b), suggesting the biological signal for LI is much weaker."

Ext. Data Fig. 3b: In the main text, it is not clear which specific comparison was performed to get these genes.

We apologize for the lack of clarity. We hope the changes outlined in the preceding response also clarify this point.

Ext. Data Fig. 4d: Are all genes in all samples shown? Is the correlation similar for just the VI gene signature? Is the correlation consistent for all samples?

Yes, Extended Data Fig. 4d shows the expression of 17,544 genes measured by both the stRNA-seq and bulk RNAseq assays across all samples (n=15). The correlation is consistent for all samples, with the range in spearman correlation coefficients for individual samples between 0.80 and 0.87. We have added the sample-level correlation in the source data file. When we consider only the correlation for the VI signature genes present in both assays, we observe a similarly high correlation (0.78) across all samples. We have added this correlation plot as Extended Data Fig. 4e. We have modified the following sentence from the results section. Previously, it read: "Strong concordance was observed between pseudo-bulked stRNA-seq gene expression counts and bulk RNA-Seq counts of matched tumor tissues (spearman R=0.85) (Extended Daga Fig. 4d)." It now states: "Strong concordance was observed between pseudo-bulked stRNA-seq gene expression counts and bulk RNA-Seq counts of matched tumor tissues across all genes (spearman R=0.85) (Extended Data Fig. 4d) and across VI signature genes (spearman R=0.78) (Extended Data Fig. 4e)."

Fig. 3a: It would be helpful to have the cell types order be consistent with Fig. 3b.

We thank the reviewer for this suggestion. We have updated the cell type order in Fig.3a accordingly.

Fig. 3b: Are all the cell types listed in the legend present? For this visualization, it may be beneficial to collapse all the cell types with negligible proportions.

We thank the reviewer for this suggestion. While all the cell types listed in the legend were present, the reviewer is correct to point out that some had negligible proportions. We have now revised Fig. 3b to only plot cell types that had proportions >1%.

Fig. 3b: It would be useful to additionally show the cell type frequencies across the pathology annotations.

We thank the reviewer for this suggestion. We agree that this would be an informative plot. We have now replaced the original Extended Data Fig. 5a panel and updated the figure legend in favor of this new plot showing cell type proportions across LUAD pathology annotations. Previously the results section read: “Visualization of cell types within the stRNA-seq data showed estimated cell types localizing with expected pathology structures (e.g., B cells with lymphoid aggregates) (Extended Data Fig. 5a).” It now reads: “LUAD pathology annotations showed distinct cell type composition (e.g., B cells with lymphoid aggregates) (Extended Data Fig. 5a).”

Fig. 3b: Are there other cell types in addition to peribronchial fibroblasts with significantly altered abundances between the two groups?

We highlighted peribronchial fibroblasts and macrophages as cell types with different abundances between VI- and VI+ samples, as these showed statistically significant differences in the bulk RNAseq data after correction for multiple hypothesis testing. We have highlighted that these are the only cell types that differ significantly between the VI- and VI+ samples in the Results.

Fig. 3c: It would be interesting to see the stRNA-seq proportions in addition to the bulk RNA-seq ones.

Although we reported trends for higher proportion of peribronchial fibroblasts, B cells, and plasma cells with VI+ tumors in our stRNA-seq data shown in Fig 3B, these were not statistically significant due to low N. Thus, we focused only on changes in proportions between VI+ and VI- tumors that we could validate by deconvolution in our bulk RNA-seq discovery cohort, after accounting for multiple hypothesis testing.

Fig. 3d: If you check the enrichment of the VI cluster signatures across all the cell types (analogous to Ext. Data Fig. 5b), does that yield different associations?

We thank the reviewer for this suggestion. We have now examined the enrichment of the VI cluster signatures across all the cell types in Fig 3d and included this as Extended Data Fig. 5d for comparison. Importantly, the tumor cells present in the single-cell atlas are not the same as those in our stRNA-seq data and may reflect different etiologies. We note similarities from this analysis with Fig. 3d; for instance, VI gene cluster 4 shows enrichment in club cells and shows some spatial correlation with club cells in our LUAD stRNAseq data. VI gene cluster 2 shows higher enrichment in peribronchial fibroblasts than alveolar or adventitial fibroblasts, as expected. However, this analysis also highlights the limitation in not considering spatial location. For instance, spatial correlation in our stRNA-seq data is better able to detect the relationship between VI gene cluster 1 or 3 and tumor cells within the tissue. We have added the following text to the results section of the manuscript: “These associations were not as apparent when examining enrichment scores in the scRNA-seq atlas (Extended Data Fig. 5d), highlighting the added value of spatial deconvolution.”

Fig. 4a: The gene selection process includes dividing the genes into four clusters. Can you clarify the reasoning for the step?

We agree this is a useful point to clarify. We selected $k=4$ clusters in our original signature (Fig 1B.) based on several factors. First, we evaluated the biological enrichment terms returned by EnrichR as reported in Fig 1C. When selecting $k>4$ clusters, we did not find distinct and significant biological enrichment terms between all clusters, suggesting lack of biologically distinct gene co-expression with $k>4$. For a more quantitative metric, we explored other values for k ($k = 2-10$) using consensus clustering and found that adding clusters greatly improved cluster stability up to but not beyond $k=4$. We have now included the consensus clustering results as an additional panel in Extended Data Fig. 2. We also added the following text to the results: “Based on our analysis of $k=2-10$ from consensus clustering, the delta area under the cumulative density function indicated substantial improvement in clustering stability up to but not beyond $k=4$ (Extended Data Fig. 2a).”

We also added the following text to the methods in the “Derivation of VI signature and biological annotation” section: “ $K=4$ gene expression clusters were identified using the Ward2 method of hierarchical clustering of genes and samples based on Euclidean distance⁷². When we explored other values for k ($k = 2-10$) using consensus clustering, we found that clustering stability showed substantial improvement (large change in delta area under the CDF) up to but not beyond $k=4$. Additionally, $k>4$ did not return distinct significant biological enrichment terms between all clusters, suggesting a lack of biologically distinct gene co-expression with $k>4$.” Finally, when we perform the gene selection process as part of our cross-validation pipeline in Fig. 4a, we preserve this division into four clusters in the outer train loop to ensure that we are consistently training a VI predictor that contains all aspects of VI+ tumor biology.

Fig. 4b: Does increasing or decreasing the gene set size improve the performance of the predictor?

We initially picked a gene set size representing 10% of our 474 gene VI signature for ease of model evaluation, as opposed to trying to fine-tune the most optimal biomarker. However, within our cross-validation pipeline in our discovery cohort we varied our gene set size from 2.5% of the total 474 gene VI signature up to 40% of the signature and observed consistent performance using our GLM model. We have now included an additional panel in Extended Data Fig. 7 (previously Extended Data Fig. 6) that demonstrates this lack of dependence of gene set size on VI predictor performance. We also added the following text to the results: “Predictors utilizing a binomial logit generalized linear model (GLM) with ridge regression performed the best on the internal cross-validation test sets within our discovery cohort and we observed no dependence of gene set size on model performance (Extended Data Fig. 7a and 7b). We also added the following text to the methods in the “VI predictor derivation” section: “We observed no dependence of gene set size on model performance up to 40% of the signature.”

Fig. 5a: The figure legend states “VI+ tumors that did not contain VI foci in the capture area were considered as proximal VI+”. Would the tumors without VI foci be considered distal?

We thank the reviewer for catching this error. The Fig. 5a legend should read: “Examples of binning VI+ tumors into distal VI+ (spots > 1mm outside invaded foci boundary) and proximal VI+ (spots < 1mm from and including the invaded foci). Any VI+ tumors that did not contain VI foci in the capture area were considered as distal VI+”. We have updated the text in figure legend.

Reviewer #3 (Remarks on code availability):

The URL provided in the Reviewer Assessment form (https://github.com/dst53/VI_manuscript) is different from the URL provided in the Code and Software Submission Checklist and the Reporting Summary (https://github.com/dst53/Steiner_NatComm). Neither one is accessible.

We thank the reviewer for bringing this to our attention. The correct URL is https://github.com/dst53/Steiner_NatComm and the code should now be accessible.

RESPONSE TO REVIEWERS' COMMENTS

Reviewer #1 (Remarks to the Author):

In response to my comments authors have performed a study on 24 selected FFPE samples and IHC and ISH for col1a1 on several slides from one sample. The following are my comments on these and other modifications:

1-clinical utility:

The addition of the FFPE biopsies to the end of the paper increases the clinical applicability of the paper. However, unfortunately it falls short of the main purpose of my critique which was to show the actual clinical utility of this approach. Testing the results of the bulk seq on 12 VI+ and 12VI- preselected samples does not mean that this approach will work on FFPE biopsies. This groups of samples do not have the variation seen in a regular sample pool and the AUROC derived from this set is of questionable applicability for clinical practice. Also, the placement of this section as a proof of concept defies the main purpose of this critique; the fact that the main utility of this finding, if validated, would be its diagnostic utility for FFPE samples.

R1C1. We thank the reviewer for these comments. We agree with the reviewer about the need to demonstrate clinical utility prior to clinical implementation, while acknowledging that true clinical utility – showing that the use of the test changes patient management and improves outcomes – requires a prospective, randomized controlled trial with multi-year endpoints that is beyond the scope of this revision. With this in mind, we now introduce additional findings related to the diagnostic performance of our classifier in our pre-surgical biopsy cohort that further defines the potential clinical application of our biomarker. Below is a summary of these findings:

We first specified a classification threshold for the biomarker score based on the Youden point in the discovery cohort and used this threshold to classify the presurgical biopsy samples. We now report the VI predictor has a sensitivity of 0.83 (exact binomial 95% confidence interval (CI) 0.52 - 0.98) for predicting VI+ tumors at resection and a specificity of 0.92 (95% CI 0.62 to 1.0). As the reviewer points out, our biopsy cohort has a balanced design with equal numbers of VI+ and VI- tumors. We intentionally over-sampled VI+ tumors relative to their real-world prevalence (~25-30%) to ensure equivalent precision for estimating both sensitivity and specificity in this pilot cohort.

Next, in keeping with standard practice for estimating diagnostic accuracy (PMID: 35403239), we used independent and uniform Beta priors for sensitivity and specificity, updated them to posterior distributions based on our biopsy cohort's true-positive/false-negative and true-negative/false-positive classification counts, simulated 10,000 specificity/sensitivity pairs and then applied Bayes' theorem to estimate positive predictive value (PPV) and negative predictive value (NPV) over a realistic range of VI prevalence (15-35%).

Despite our small, preselected biopsy cohort, the signature's prevalence-adjusted negative predictive value (NPV) estimate is high and precise (median 93%, credible interval (CI) 84-98% at 25% prevalence, dropping to median 91%, CI 81-98% at 30% prevalence), suggesting that a negative VI predictor classification may be used to recommend sublobar resection given the low probability that vascular invasion will be detected on final pathology of the resected specimen. Notably, because our balanced biopsy cohort design under-samples true negatives relative to observed real-world prevalence, these CIs are conservative and would tighten in an unselected cohort.

Conversely, the prevalence-adjusted positive predictive value estimates from this biopsy cohort are less precise (median 73%, CI 47%-95% at 30% prevalence), reflecting both the small number of VI+ biopsy cases and the moderate prevalence of VI+ stage I LUAD in the real-world population. As a result, it is unclear if a positive VI call will be sufficient to recommend lobectomy. Larger prospective studies and incorporation of orthogonal assays (e.g. radiomics) will be needed to narrow PPV confidence bounds and demonstrate the signature's clinical utility for guiding more extensive surgery.

With the current n=24 FFPE biopsy specimens we have demonstrated the classifier's potential to be measured in routine biopsy tissue and its high NPV. Larger, prospective cohorts will be needed to determine its PPV and determine true clinical utility. We have now included the projected NPV and PPV result as a new panel in Figure 5 and updated the methods, results and discussion with this new analysis, including the limitations described above.

-The addition to the TNM staging:

The statistical analyses showing that VI remains an independent predictor of RFS with TNM stage added is satisfactory. (fig1i extended) However,

-I cannot find the clinical characteristics of this 183 tumors that are "a subset" of previously published data (Yambayev et al). I can not find 1823 samples in that paper.

R1C2. We thank the reviewer for pointing out this omission. We have now added an additional supplementary table (now the new Table S1, with subsequent tables renamed) that summarizes the clinical characteristics of the resected stage I LUAD tumors (n=192) that were a subset of the previously published data and used for the analysis presented in Extended Data Fig. 1.

-Also: what percentage of the data is missing for each variable?

R1C3. We thank the reviewer for this inquiry about Extended Data Fig. 1i. From the 192 tumors, we excluded 16 tumors that had missing values. The remaining tumors (n=176) had no missing data for any of the covariates presented in the multivariate analysis in Extended Data Fig. 1i. We have now revised the Extended Data Fig. 1i legend with this information to improve clarity. The legend now reads: "i. Association of VI with RFS (n=176 tumors) when controlling for common clinical variables, collection site (LHMC – Lahey Hospital & Medical Center, BMC – Boston Medical Center) and the other invasion types. Patients with

any missing values (n=10) were excluded: including n=3 excluded for missing race, and n=7 for missing pack years. Additionally, patients with mucinous tumors (n=6) were excluded.”

-About the age and pack year in these samples. Since these are continuous variables why are there no CIs?

R1C4. We thank the reviewer for this astute observation. There are indeed confidence intervals for age and pack years, but they were not visible on the plot because of their tightness. The hazard ratio for age was 0.996 (95% CI 0.956-1.04) and the hazard ratio for pack years was 1.010 (95% CI 0.994-1.02). We have now edited the Extended Data Fig. 1i plot to reduce the size of the points so that the CIs for age and pack years are visible.

-Extrinsic factors: Even though author’s response about focusing on VI as an intrinsic factor (vs extrinsic factors such as clear margins and suboptimal sampling) is valid, the extrinsic features will still have importance in determining the predictive power of a biomarker in real world and should at least be mentioned in the discussion.

R1C5. We thank the reviewer for raising this important point. We have now acknowledged in the discussion that extrinsic factors will remain important determinants of outcome and should be incorporated when evaluating the real-world performance of the VI predictor in prospective cohorts. We have added the following sentence to the second to last paragraph in the discussion: “Although our focus is on VI as an intrinsic feature of tumor biology, the eventual clinical value of a VI predictor will also depend on extrinsic factors that influence recurrence risk, including surgical margins and the adequacy of nodal sampling.”

2-Clinical feasibility:

The authors in their response mention that in a real-world scenario the remainder of an FFPE after pathologist review will probably be used for bulk-seq, stressing the importance of the FFPE sample validation. Yet perhaps the weakest point of this manuscript at present is the analysis of the FFPE samples (24 preselected VI+ and VI- samples). Increasing the number of the FFPE samples and testing this approach on a true representative sample pool (rather than as an add-on proof of concept piece) will show the basic utility of this study.

R1C6. We kindly ask the reviewer to refer to the response above (R1C1) about clinical utility, which addresses this critique.

The comment about the use of RNA-seq in current clinical practice: the sentence “whole transcriptome RNA-seq test...fine-needle aspiration FFPE material used ... is now standard of care for preoperative diagnosis...” should be moderated. The use of molecular probes for indeterminate thyroid nodules refers to a small group of samples assessed by FNA, and mRNA sequencing is only one of the three approaches used for these samples: mutational analysis, mRNA sequencing, and miRNA expression

analysis. Authors are presenting Bulk-seq as a method for general classification of all early stage NSCLC samples, which is a very different premise.

R1C7. We thank the reviewer for this clarification. We agree that our initial response overstated the role of RNA-seq in thyroid nodule assessment. While RNA-seq is indeed one of the molecular approaches used in this setting, it is not the sole standard-of-care method. Our point was that RNA-seq is used clinically on small FFPE-derived samples, supporting its feasibility as an approach for assessing our VI predictor. Furthermore, we did not intend to present bulk RNA-seq as a general diagnostic for all early-stage NSCLC cases. As detailed above in our response related to the potential clinical application of the predictor, we expect that our VI predictor will be most useful in a subgroup of patients, for example as a rule-out test, that is identifying patients unlikely to harbor VI, where less aggressive management may be appropriate. This approach parallels how other RNA-seq predictors are deployed clinically, such as Veracyte's Percepta Genomic Sequencing Classifier (PMID: 33849470), which is used to down-classify risk of malignancy in indeterminate pulmonary nodules.

Also, on the disparities between RNA-seq and qRT-PCR please refer to:

-Int. J. Mol. Sci. 2021, 22(17), 9349; <https://doi.org/10.3390/ijms22179349>

Despite the extensive discussion on the advantages of sequencing by some authors and commercial entities, I think we all know that the access to RNAseq is much more limited and technical caveats are much more prevalent, compared to hybridization probes or qPCR. These methods are currently used for biomarker analysis in many clinical settings and for various cancers, including breast and colon cancer. Therefore, validation by qPCR or RNA probes for some genes, which is routinely done for RNAseq studies is reasonable and adds to the possible clinical utility.

R1C8. We thank the reviewer for highlighting this important consideration. We agree that qPCR and hybridization probe-based assays are widely used in current clinical biomarker practice and that the potential to assess our predictor by these methods might add to its clinical efficiency. In this study we focused on demonstrating the biological validity and clinical associations of the RNA-seq-derived predictor, including the feasibility of assessing it via RNA-seq in FFPE biopsy tissue. We view testing alternative measurement platforms such as qPCR or targeted probe panels as an important future step, but one that is outside the scope of our initial discovery and validation work. We have added the following sentence to the last paragraph of the discussion to acknowledge this as a limitation and suggest it as a future direction: "Although the VI predictor was derived and validated using RNA-seq, future work should also evaluate its performance when assessed using more widely available clinical platforms, such as qPCR or targeted probe panels."

3-phthophysiological significance:

This is one of the main critiques regarding the appropriateness of this paper for Nature communications. Despite the extensive work and the multitude of statistical analyses, the conclusions in the manuscript are based on correlative data and deficient in mechanistic insights, which is a prerequisite for publication in journals like Nature

Comm. The involvement of CAFs in vascular invasion is known. However, the main mechanisms mentioned for this contribution are increased angiogenesis, expression of ECM remodeling enzymes, immune response, and metabolic programming of tumor cells. (<https://doi.org/10.3389/fimmu.2025.1582532>)

Authors have picked Col1a1 for validation. This gene is expressed as part of a fibrotic tissue response to injury/invasion. Mechanistic questions remain. For instance: Does col1a1 contribute to the process of invasion? If so, how? Or is it just a consequence? Does it identify “activated” CAFs? Is there any mechanism downstream of col1a1 that can be shown to enhance any aspect of vascular invasion? In a mechanistic paper these questions need to be answered by experiments, not just correlation and citing literature.

R1C9. We appreciate the reviewer’s comments on the mechanistic aspects of CAF involvement in vascular invasion. Our study was designed to identify the molecular and cellular correlates of VI in stage I LUAD and to develop a transcriptomic predictor capable of pre-surgical prediction of VI.

We share the reviewer’s enthusiasm that our work might nevertheless provide insights about the molecular underpinnings of vascular invasion that could be tested via subsequent functional studies. Specifically, our initial stRNA-seq and IHC analyses demonstrate that VI gene cluster 2 genes, including COL1A1, are expressed in stromal myofibroblasts within desmoplastic stroma adjacent to invaded vessels. We have now expanded our IHC cohort (see R1C10 below for details) and find that in both stroma (i.e., non-desmoplastic stroma) and desmoplastic stroma regions, nearly all POSTN+ myofibroblasts co-expressed COL1A1, indicating that COL1A1 is broadly expressed in myofibroblasts. The fraction of these POSTN+COL1A1+ myofibroblasts increases in non-desmoplastic stroma regions of VI+ tumors relative to VI- tumors. In addition, within desmoplastic stroma regions, we have identified COL1A1+POSTN- cells. These results support a model in which stromal remodeling accompanies the angioinvasive phenotype, consistent with prior studies implicating CAF-mediated ECM remodeling in invasive cancer biology (PMID: 21107292). We have now added the following statement to the discussion section: “In addition, desmoplastic stroma, which comprises a greater fraction of the tissue area in VI+ tumors, contains COL1A1+POSTN- stromal cells, raising the possibility that additional stromal cell populations contribute to ECM remodeling. This hypothesis, and the relative contribution of these cells relative to the contributions of POSTN+COL1A1+ myofibroblasts will require mechanistic validation, and it remains to be determined if the proportion of these COL1A1+POSTN- cells within desmoplastic stroma is modulated by vascular invasion.”

-Critique on association with a specific cell type: staining for one of the genes. Authors have performed IHC and ISH staining for col1a1, on multiple sections from one sample. Question arises, why could this not be done on more than one sample, as is regularly done for the histopathological validation of a gene involved in a pathological process? Staining even one or two sections from 10-20 samples for col1A1 and

presenting the histopathological images that show the expression of this gene in myfibroblasts within the conventional “vascular invasion” sites would be appropriate for this response. The multiple statistical analyses show that within sample 4 the expression of col1A1 coincides with the expression of the other genes in cluster 2 in that sample. This kind of analysis adds to the rigor but does not provide an adequate validation and link between this new finding and the currently used histopathological studies that are the basis of the statistical analysis in extended figure 1i.

R1C10. We thank the reviewer for these comments. We initially focused on one sample for the direct validation of our stRNA-seq deconvolution, because it was the only sample with remaining serial sections for IHC and ISH staining that were close enough to the section used for stRNA-seq to be registered. We now report IHC results for an additional 20 resected stage I LUAD specimens (11 VI-, 9 VI+) from an independent cohort. All samples were from patients not treated with neoadjuvant or adjuvant therapy and were receiving care at Inova Schar Cancer Institute. Our scRNA-seq and stRNA-seq analyses indicated that VI gene cluster 2 was enriched in myfibroblasts, prompting us to examine the spatial distribution of these cells using dual IHC for COL1A1 and POSTN. POSTN was used as the key IHC marker of myfibroblasts in Hanley et al (PMID: 36720863) and is also a VI gene cluster 2 gene.

After analyzing these new IHC samples, we found that nearly all POSTN+ cells (myfibroblasts) in pathologist-annotated stroma regions (i.e., non-desmoplastic stroma) are also COL1A1+ regardless of VI status (i.e. the fraction of POSTN+ COL1A1+ double positive cells out of all POSTN+ cells), a pattern that also persisted within desmoplastic stroma regions as described below. This supports our observation that COL1A1 is expressed in myfibroblasts and helps explain the co-expression of COL1A1 and POSTN within VI cluster 2 in our RNA-seq dataset. Furthermore, the fraction of POSTN+ COL1A1+ cells out of total cells in stroma regions is significantly higher in VI+ than VI- stage I LUAD (median 68% vs. 32%, $p = 0.006$), indicating VI-specific expansion of myfibroblasts. Collectively, these findings demonstrate that COL1A1, a component of VI gene cluster 2 with strong importance in the VI predictor, is expressed in almost all myfibroblasts and that COL1A1-expressing myfibroblasts are more abundant in VI+ than VI- tumors.

We next revisited desmoplastic stroma regions specifically to validate our stRNA-seq findings in Extended Data Fig. 6, where we demonstrated that the COL1A1 IHC+ cell fraction was higher in desmoplastic stroma annotated spots in a VI+ tumor. We strengthened our previous finding that the fraction of all cells in desmoplastic stroma regions that are COL1A1+ is significantly higher than outside desmoplastic stroma regions (median 87% vs. 57%, $p=7.09 \times 10^{-7}$) without a corresponding rise in POSTN+ COL1A1+ cells (median 26.3% vs. 21.0%, $p=0.263$). This is a result of an increase in COL1A1+POSTN- stromal cells (median 61% vs. 39%, $p=0.02$). These findings refine our previous interpretation by showing that desmoplastic stroma, which is more abundant in VI+ tumors, contains both POSTN+COL1A1+ myfibroblasts and an additional population of COL1A1+POSTN- stromal cells. Together with POSTN+COL1A1+ myfibroblasts that are expanded in VI+ non-desmoplastic stroma, these desmoplastic and non-desmoplastic

populations collectively reflect the fibroblast-driven ECM remodeling program captured by VI gene cluster 2. We have now updated the manuscript to reflect these new results, including the figures, methods, results, and discussion sections.

We have added the following text to the results section:

To further generalize these findings beyond the stRNA-seq sample, and to directly demonstrate the expression of COL1A1 in myofibroblasts, we analyzed an independent cohort (Table S4) of 20 resected stage I LUAD samples (11 VI⁻, 9 VI⁺) using duplex IHC for COL1A1 and POSTN. POSTN is a canonical myofibroblast marker (PMID: 36720863, PMID: 39255773) and a VI cluster 2 gene. In pathologist-annotated stroma regions (i.e., non-desmoplastic stroma), nearly all POSTN⁺ cells were also COL1A1⁺ (Extended Data Fig. 6g-h). The fraction of COL1A1⁺POSTN⁺ cells in these regions was significantly higher in VI⁺ than VI⁻ tumors (median 68% vs 32%, $p=0.006$), consistent with VI associated myofibroblast expansion (Fig. 3g). In analyzing desmoplastic stroma, we restricted our analysis to desmoplastic stroma from VI⁺ tumors given the paucity of desmoplastic stroma in VI⁻ tumors. Within desmoplastic stroma, POSTN⁺ cells are also nearly universally COL1A1⁺ (~100%, Extended Data Fig. 6i), confirming persistent co-expression within myofibroblasts. The fraction of all cells in desmoplastic stroma regions that are COL1A1⁺ was significantly higher than outside the desmoplastic stroma regions (median 87% vs 57%, $p=7.09 \times 10^{-7}$) (Fig. 3h), without a corresponding rise in POSTN⁺COL1A1⁺ cells (median 26.3% vs. 21.0%, $p=0.263$) (Extended Data Fig. 6j). This is a result of an increase in COL1A1⁺POSTN⁻ stromal cells (median 61% vs. 39%, $p=0.02$) (Extended Data Fig. 6k). These findings support and extend the stRNA-seq results, demonstrating that the elevated expression of VI cluster 2 in VI⁺ tumors reflects both the VI-specific expansion of POSTN⁺COL1A1⁺ myofibroblasts in non-desmoplastic stroma, and the VI-associated expansion of desmoplastic stroma in which the majority of COL1A1⁺ cells are POSTN⁻.

We have added the following text to the discussion:

Subsequent IHC analysis in an independent LUAD cohort of resected tumors validated the stRNA-seq findings and clarified the cell type expression of Cluster 2. In both non-desmoplastic and desmoplastic stroma regions, nearly all POSTN⁺ myofibroblasts co-expressed COL1A1, a key Cluster 2 gene. The fraction of these POSTN⁺COL1A1⁺ myofibroblasts increases in non-desmoplastic stroma regions of VI⁺ tumors relative to VI⁻ tumors. In addition, desmoplastic stroma, which comprises a greater fraction of the tissue area in VI⁺ tumors, contains COL1A1⁺POSTN⁻ stromal cells, raising the possibility that additional stromal cell populations contribute to ECM remodeling. This hypothesis, and the relative contribution of these cells relative to the contributions of POSTN⁺COL1A1⁺ myofibroblasts will require mechanistic validation, and it remains to be determined if the proportion of these COL1A1⁺POSTN⁻ cells within desmoplastic stroma is modulated by vascular invasion.

-Multivariable analysis on the derivation study: Extended Figure 7c is adequate and a critical piece of data. The same questions about the CI for age and pack year apply and

need to be answered. If the latter questions are adequately answered, I suggest moving this data to the main figures.

R1C11. We thank the reviewer for these comments. As with Extended Data Fig. 1i, the confidence intervals for age and pack years are present in Extended Data Fig. 7c but were not visible on the plot due to their tightness. The hazard ratio for age was 0.952 (95% CI 0.881 – 1.03) and the hazard ratio for pack years was 0.977 (95% CI 0.950 – 1.00). We have now edited the original Extended Data Fig. 7c plot to reduce the size of the points so that the CIs for age and pack years are visible. We agree that this analysis provides important context for the VI predictor's independence but given the limited number of recurrence events relative to the number of covariates, we view it as confirmatory analysis rather than as a fully powered prognostic model and have therefore retained it in the Extended Data. We have clarified this in the figure legend.

-On the distinction between LI and VI : the changes are adequate. I would add that LI was also an independent predictor in extended fig.1i and would show the gene signature associated with it as a variable in extended figure 7c.

R1C12. We thank the reviewer for suggesting inclusion of the LI gene signature in this analysis. While we identified a small set of LI-associated genes (n=6 at FDR < 0.01), this was insufficient to construct a stable predictor suitable for inclusion in the multivariate model. Instead, we have added a sentence in the results section noting that LI is also an independent predictor of recurrence in Extended Data Fig. 1i. The revised text now reads: “Regardless, VI remained a significant predictor of recurrence when including clinical covariates and other invasion types (HR = 11.04, 95% CI 3.80–32.1, multivariate Cox regression $p=1.01 \times 10^{-5}$; Extended Data Fig. 1i). Of the different invasion types, only LI was also an independent predictor of recurrence (HR = 2.73, 95% CI 1.15 – 6.44, multivariate Cox regression $p=0.02$).”

-The explanations for tumor hypoxia and cell cycle genes are adequate.

-There are still a multitude of technical details in result sections that take away from the main message in each paragraph.

R1C13. We thank the reviewer for pointing this out. During the process of including the new results described in the responses above, we have also further refined the results section to remove unnecessary technical details.

The literature references are adequate.

Additionally:

The manuscript still has several stylistic and language issues and would benefit from cleaning up.

Some examples:

-What is the meaning of the sentence “ tumors were graded.... Or VI, is associated with outcome than the World...”

R1C14. We thank the reviewer for pointing this out. We have now corrected the sentence to: “Tumors were graded using our novel grading system whose reproducibility has been validated by us and others^{8,13,27,28}. This novel grading system classifies tumors into low malignant potential (LMP), no special type (NST), or VI and is more associated with outcome than the World Health Organization (WHO) grading system (Extended Data Fig. 1a-b).”

-The text still has redundant and hard to understand details that do not clarify the significance: For instance:

“ based on our analysis from consensus clustering the delta area under the cumulative....” This sentence and sentences alike should be modified/simplified or deleted.

R1C15. We thank the reviewer for pointing this out. We have now simplified this sentence and moved the technical details to the methods section. This sentence now reads: “We selected the optimal number of clusters as four via consensus clustering (Extended Data Fig. 2a).”

-Where can we find the list of the genes for the four clusters described in the second results paragraph?

R1C16. The genes and their respective cluster assignments are provided in the uploaded source data file in the tab labeled Figure 1B-2.

-In discussion paragraph 7:

“High VI predictor scores were associated not only with... VI- tumors were shedding cells into the circulation...”

This sentence is confusing and needs clarification.

R1C17. We thank the reviewer for pointing this out. We have revised the sentence to clarify that necrosis is the key feature associated with VI predictor score in VI- tumors in our validation cohort, whereas in the TRACERx cohort (which lacks VI labels) the VI predictor score is associated with ctDNA positivity. The sentence now reads: “In our study, ANGPTL7 was not part of the VI signature but we found that the VI predictor score is positively associated with necrosis among the VI tumors in our validation cohort and positively associated with detectable preoperative circulating tumor DNA in the TRACERx cohort, suggesting a similar relationship between necrosis and vascular invasion in early-stage LUAD.”

Reviewer #2 (Remarks to the Author):

The authors have responded to the main comments that are listed below.

1. The determination of vascular invasion was by a single thoracic pathologist. Were these cases previously read by other clinical pathologists and what was the agreement of determination of vascular invasion in cases read by more than one pathologist? The authors should address inter-reader variability concerns with determination of this pathological finding.

The authors' response is complete.

2. The introduction emphasizes the potential clinical significance of detection of vascular invasion and suggests that the finding may support use of adjuvant therapy or lobectomy. This is confusing and should be clarified. Further, it is possible that the signature impact on recurrence free survival may be independent of performance of lobectomy or administration of adjuvant therapy. Addressing the potential clinical utility aspects of the proposed classifier and how it can be assessed will be helpful.

The discussion addresses these points. To further address potential clinical utility, the authors added analysis of a cohort of presurgical FFPE lung biopsy specimens acquired from 24 subjects in Figure 5f. The AUROC is shown. To better understand the performance of this proposed biomarker, it will be important to show the sensitivity, specificity, PPV and NPV of the predictor.

R2C1. We agree with the reviewer that these are important metrics to report. In response to this request, we first specified a classification threshold for the biomarker score based on the Youden point in the discovery cohort and used this threshold to classify the presurgical biopsy samples. We now report the VI predictor has a sensitivity of 0.83 (exact binomial 95% confidence interval (CI) 0.52 - 0.98) for predicting VI+ tumors at resection and a specificity of 0.92 (95% CI 0.62 to 1.0). Our biopsy cohort has a balanced design with equal numbers of VI+ and VI- tumors. We intentionally over-sampled VI+ tumors relative to their clinical prevalence (~25-30%) to ensure equivalent precision for estimating both sensitivity and specificity in this balanced pilot cohort.

Next, in keeping with standard practice for estimating diagnostic accuracy (PMID: 35403239), we used independent and uniform Beta priors for sensitivity and specificity, updated them to posterior distributions based on our biopsy cohort's true-positive/false-negative and true-negative/false-positive classification counts, simulated 10,000 specificity/sensitivity pairs and then applied Bayes' theorem to estimate positive predictive value (PPV) and negative predictive value (NPV) over a realistic range of VI prevalence (15-35%).

Despite our small, preselected biopsy cohort, the signature's prevalence-adjusted negative predictive value (NPV) estimate is high and precise (median 93%, credible interval (CI) 84-98% at 25% prevalence, dropping to median 91%, CI 81-98% at 30% prevalence), suggesting that a negative VI predictor classification may be used to recommend sublobar resection given the low probability that vascular invasion will be detected on final

pathology of the resected specimen. Notably, because our balanced biopsy cohort design under-samples true negatives relative to previously observed real world prevalence, these CIs are conservative and would tighten in an unselected cohort.

Conversely, the prevalence-adjusted positive predictive value estimates from this biopsy cohort are less precise (median 73%, CI 47%-95% at 30% prevalence), reflecting both the small number of VI+ biopsy cases and the moderate prevalence of VI+ stage I LUAD in the real-world population. As a result, it is unclear if a positive VI call will be sufficient to recommend lobectomy. Larger prospective studies and incorporation of orthogonal assays (e.g. radiomics) will be needed to narrow PPV confidence bounds and demonstrate the signature's clinical utility. We have now included the projected NPV and PPV result as a new panel in Figure 5 and updated the methods, results and discussion with this new analysis, including the limitations described above.

In addition, it is important to carefully describe the specimens from which the predictor was evaluated. Were these consecutively acquired specimens or were they selected? Were the specimens derived from cell block or forceps biopsy? What was the evidence for vascular invasion on the biopsy specimens used for the analysis, if from forceps biopsy?

R2C2. We thank the reviewer for these questions. The specimens in our pre-surgical biopsy cohort were selected retrospectively. We selected patients with pathologic stage I LUAD that did not receive neoadjuvant or adjuvant therapy and that had available preoperative biopsy material with tumor nuclei content above 50%. FFPE specimens were derived from needle biopsy, forceps biopsy, or fine needle aspiration. We have now included this info in Table S6. The determination of VI was based on a representative FFPE block from the subsequently resected tumors. We enriched our biopsy cohort to include equal numbers of VI+ and VI- samples so that both sensitivity and specificity could be estimated with similar precision in this pilot set, acknowledging that this 50/50 split over-samples VI+ cases relative to their clinical prevalence in stage I LUAD (~20-30% VI+ cases). We have also now added the following sentence to the discussion acknowledging future needed work: "In addition, further work is required to determine the required fraction of tumor cells in the presurgical biopsies for accurate biomarker assessment."

3. Figure 3 shows imputed cell type associations with the vascular invasion signature using spatial transcriptomic and bulk sequencing data. It will be important to validate key gene expression signatures in specific cell types using immunohistochemistry.

In extended figure 6, The authors show analysis of COL1A1IHC and ISH in one sample. This does not address validation of key V1 biomarkers cell specific expression in lung adenocarcinoma specimens with and without vascular invasion. These data will be important to demonstrate the generalizability of the reported observations.

R2C3. We thank the reviewer for these comments. We initially focused on one sample for the direct validation of our stRNA-seq deconvolution, because it was the only sample with remaining serial sections for IHC and ISH staining that were close enough to the section used for stRNA-seq to be registered. We now report IHC results for an additional 20 resected stage I LUAD specimens (11 VI-, 9 VI+) from an independent cohort. All samples were from patients not treated with neoadjuvant or adjuvant therapy and were receiving care at Inova Schar Cancer Institute. Our scRNA-seq and stRNA-seq analyses indicated that VI gene cluster 2 was enriched in myofibroblasts, prompting us to examine the spatial distribution of these cells using dual IHC for COL1A1 and POSTN. POSTN was used as the key IHC marker of myofibroblasts in Hanley et al (PMID: 36720863) and is also a VI gene cluster 2 gene.

After analyzing these new IHC samples, we found that nearly all POSTN+ cells (myofibroblasts) in pathologist-annotated stroma regions (i.e., non-desmoplastic stroma) are also COL1A1+ regardless of VI status (i.e. the fraction of POSTN+ COL1A1+ double positive cells out of all POSTN+ cells), a pattern that also persisted within desmoplastic stroma regions as described below. This supports our observation that COL1A1 is expressed in myofibroblasts and helps explain the co-expression of COL1A1 and POSTN within VI cluster 2 in our RNA-seq dataset. Furthermore, the fraction of POSTN+ COL1A1+ cells out of total cells in stroma regions is significantly higher in VI+ than VI- stage I LUAD (median 68% vs. 32%, $p = 0.006$), indicating VI-specific expansion of myofibroblasts. Collectively, these findings demonstrate that COL1A1, a component of VI gene cluster 2 with strong importance in the VI predictor, is expressed in almost all myofibroblasts and that COL1A1-expressing myofibroblasts are more abundant in VI+ than VI- tumors.

We next revisited desmoplastic stroma regions specifically to validate our stRNA-seq findings in Extended Data Fig. 6, where we demonstrated that the COL1A1 IHC+ cell fraction was higher in desmoplastic stroma annotated spots in a VI+ tumor. We strengthened our previous finding that the fraction of all cells in desmoplastic stroma regions that are COL1A1+ is significantly higher than outside desmoplastic stroma regions (median 87% vs. 57%, $p=7.09 \times 10^{-7}$) without a corresponding rise in POSTN+ COL1A1+ cells (median 26.3% vs. 21.0%, $p=0.263$). This is a result of an increase in COL1A1+POSTN- stromal cells (median 61% vs. 39%, $p=0.02$). These findings refine our previous interpretation by showing that desmoplastic stroma, which is more abundant in VI+ tumors, contains both POSTN+COL1A1+ myofibroblasts and an additional population of COL1A1+POSTN- stromal cells. Together with POSTN+COL1A1+ myofibroblasts that are expanded in VI+ non-desmoplastic stroma, these desmoplastic and non-desmoplastic populations collectively reflect the fibroblast-driven ECM remodeling program captured by VI gene cluster 2. We have now updated the manuscript to reflect these new results, including the figures, methods, results, and discussion sections.

We have added the following text to the results section:

To further generalize these findings beyond the stRNA-seq sample, and to directly demonstrate the expression of COL1A1 in myofibroblasts, we analyzed an independent cohort (Table S4) of 20 resected stage I LUAD samples (11 VI-, 9 VI+) using duplex IHC for

COL1A1 and POSTN. POSTN is a canonical myofibroblast marker (PMID: 36720863, PMID: 39255773) and a VI cluster 2 gene. In pathologist-annotated stroma regions (i.e., non-desmoplastic stroma), nearly all POSTN⁺ cells were also COL1A1⁺ (Extended Data Fig. 6g-h). The fraction of COL1A1⁺POSTN⁺ cells in these regions was significantly higher in VI⁺ than VI⁻ tumors (median 68% vs 32%, p=0.006), consistent with VI associated myofibroblast expansion (Fig. 3g). In analyzing desmoplastic stroma, we restricted our analysis to desmoplastic stroma from VI⁺ tumors given the paucity of desmoplastic stroma in VI⁻ tumors. Within desmoplastic stroma, POSTN⁺ cells are also nearly universally COL1A1⁺ (~100%, Extended Data Fig. 6i), confirming persistent co-expression within myofibroblasts. The fraction of all cells in desmoplastic stroma regions that are COL1A1⁺ was significantly higher than outside the desmoplastic stroma regions (median 87% vs 57%, p=7.09x10⁻⁷) (Fig. 3h), without a corresponding rise in POSTN⁺COL1A1⁺ cells (median 26.3% vs. 21.0%, p=0.263) (Extended Data Fig. 6j). This is a result of an increase in COL1A1⁺POSTN⁻ stromal cells (median 61% vs. 39%, p=0.02) (Extended Data Fig. 6k). These findings support and extend the stRNA-seq results, demonstrating that the elevated expression of VI cluster 2 in VI⁺ tumors reflects both the VI-specific expansion of POSTN⁺COL1A1⁺ myofibroblasts in non-desmoplastic stroma, and the VI-associated expansion of desmoplastic stroma in which the majority of COL1A1⁺ cells are POSTN⁻.

We have added the following text to the discussion:

Subsequent IHC analysis in an independent LUAD cohort of resected tumors validated the stRNA-seq findings and clarified the cell type expression of Cluster 2. In both non-desmoplastic and desmoplastic stroma regions, nearly all POSTN⁺ myofibroblasts co-expressed COL1A1, a key Cluster 2 gene. The fraction of these POSTN⁺COL1A1⁺ myofibroblasts increases in non-desmoplastic stroma regions of VI⁺ tumors relative to VI⁻ tumors. In addition, desmoplastic stroma, which comprises a greater fraction of the tissue area in VI⁺ tumors, contains COL1A1⁺POSTN⁻ stromal cells, raising the possibility that additional stromal cell populations contribute to ECM remodeling. This hypothesis, and the relative contribution of these cells relative to the contributions of POSTN⁺COL1A1⁺ myofibroblasts will require mechanistic validation, and it remains to be determined if the proportion of these COL1A1⁺POSTN⁻ cells within desmoplastic stroma is modulated by vascular invasion.

Reviewer #3 (Remarks to the Author):

The authors addressed my comments and introduced satisfactory corrections.

RESPONSE TO REVIEWERS' COMMENTS

Reviewer #2 (Remarks to the Author):

The authors' responses are satisfactory.

Reviewer #1 rebuttal
R1C1.

1. To address comments regarding diagnostic accuracy, the authors present results from 24 selected biopsy specimens acquired from a collaborator. The results are supportive but are insufficient to support the statement that the classifier should be measured in routine biopsy tissue. Validation of diagnostic accuracy in a multi-center clinical trial is required to support this claim.

We thank the reviewer for this comment. We agree with the reviewer's concern regarding diagnostic accuracy and clinical validation of the classifier in routine biopsy tissue. As detailed in our responses below, we have revised the Abstract, Introduction, Results, and Discussion to clarify that the biopsy cohort represents proof-of-concept feasibility rather than evidence of routine clinical applicability, and to explicitly state that multi-center clinical trials will be required to establish diagnostic accuracy and clinical utility.

2. Abstract. The results do not support the claim that "VI-associated gene expression changescan be used to predict the presence of VI from routinely collected pre-surgical tumor biopsy specimens" The results do support the need for further studies to validate the diagnostic utility and to demonstrate the clinical utility of this assay.

We thank the reviewer for this comment. We have now revised the previously second to last sentence of the abstract (now the last sentence) to emphasize that VI-associated gene expression changes are detectable in pre-surgical biopsies and support further validation. The sentence now reads: "These findings indicate that VI-associated transcriptional changes extend across the tumor and are detectable in limited biopsy material, supporting further validation for preoperative risk stratification in stage I LUAD."

3. Introduction The final sentence needs to be revised for clarity and accuracy. Specifically, the term "robustness" should be clarified and the phrase "validated the VI predictor in real-world pre-surgical tumor biopsies" needs to be rewritten for accuracy- the data presented are not acquired from a "real-world" setting.

We thank the reviewer for these comments. We have now revised the final sentence of the introduction to clarify what is meant by robustness (i.e., stability across intra-tumor heterogeneity and sampling variability) and to remove the term "real-world," which could be misinterpreted. The revised text now accurately reflects the controlled, pilot nature of the biopsy cohort. The sentence now reads: "We derive a novel molecular signature

associated with VI⁺ LUAD using bulk RNA-seq, describe its association with histopathology in situ using stRNA-seq, develop and validate a machine learning based predictor of VI, evaluate its stability across intra-tumor heterogeneity (ITH) and demonstrate its proof-of-concept feasibility in a pilot cohort of pre-surgical tumor biopsies.”

4. Results. Similarly, the phrase “real-world population” should be deleted from the final paragraph.

We thank the reviewer for this comment. We have removed the phrase “real-world population” from the final paragraph of the Results section. The sentence now reads “Conversely, the prevalence-adjusted PPV estimates from this biopsy cohort are less precise (median 73%, CI 47%-95% at 30% prevalence), reflecting both the small number of VI⁺ biopsy cases and the moderate prevalence of VI⁺ stage I LUAD.” We have also removed the only other use of “real world” in the final paragraph of the discussion. The sentence “Our biopsy results support real world feasibility of predicting VI from presurgical tissue.” has been rewritten to “Our biopsy results support the technical feasibility of predicting VI from presurgical tissue.”

5. Discussion. The first paragraph on page 19 needs to be revised to place the results in appropriate context in light of the limitations noted above.

We appreciate the reviewer’s suggestion to further contextualize the biopsy findings considering their limitations. While several revisions had already been made to this section in prior rounds, we have now further revised the first paragraph on page 19 of the discussion to more explicitly emphasize the pilot nature of the biopsy cohort, remove language that could be interpreted as implying clinical decision-making, and clearly distinguish technical feasibility from diagnostic or clinical utility.

R1C3. Thank you for clarifying the confusion about samples listed in Extended Data Figure 1i and for specifying that data for 16 of the 192 tumors is missing. The authors should examine and report the potential impact of the missing data on the multivariate analysis. Simply excluding the missing data may introduce bias.

We thank the reviewer for raising this point. As noted in the figure legend for Extended Data Fig. 1i, missing covariate data affected a small subset of tumors (16/192) primarily due to missing race or smoking pack-year information, as well as mucinous tumors that were excluded from WHO pathology grading. To assess the potential impact of excluding these cases, we performed a sensitivity analysis by refitting the multivariate Cox model in the full cohort (n=192 tumors) using a reduced covariate set that avoided exclusion due to missing data. The association between vascular invasion and recurrence remained strong and statistically significant (HR = 7.07, 95% CI 2.97–16.8, multivariate Cox regression $p=9.82 \times 10^{-6}$), indicating that exclusion of cases with missing covariates does not affect the conclusions. We have now clarified this in the figure legend.